# Optimizer Choice Matters For The Emergence of Neural Collapse

**Jim Zhao,**[*] **Tin Sum Cheng**
University of Basel
{jim.zhao, tinsum.cheng}@unibas.ch

**Wojciech Masarczyk**
Warsaw University of Technology
IDEAS Research Institute
wojciech.masarczyk@gmail.com

**Aurelien Lucchi**
University of Basel
aurelien.lucchi@unibas.ch

## Abstract

Neural Collapse (NC) refers to the emergence of highly symmetric geometric structures in the representations of deep neural networks during the terminal phase of training. Despite its prevalence, the theoretical understanding of NC remains limited. Existing analyses largely ignore the role of the optimizer, thereby suggesting that NC is universal across optimization methods. In this work, we challenge this assumption and demonstrate that the choice of optimizer plays a critical role in the emergence of NC. The phenomenon is typically quantified through NC metrics, which, however, are difficult to track and analyze theoretically. To overcome this limitation, we introduce a novel diagnostic metric, NC0, whose convergence to zero is a necessary condition for NC. Using NC0, we provide theoretical evidence that NC cannot emerge under decoupled weight decay in adaptive optimizers, as implemented in AdamW. Concretely, we prove that SGD, SignGD with coupled weight decay (a special case of Adam), and SignGD with decoupled weight decay (a special case of AdamW) exhibit qualitatively different NC0 dynamics. Also, we show the accelerating effect of momentum on NC (beyond convergence of train loss) when trained with SGD, being the first result concerning momentum in the context of NC. Finally, we conduct extensive empirical experiments consisting of 3,900 training runs across various datasets, architectures, optimizers, and hyperparameters, confirming our theoretical results. This work provides the first theoretical explanation for optimizer-dependent emergence of NC and highlights the overlooked role of weight-decay coupling in shaping the implicit biases of optimizers.

## 1 Introduction

Neural networks have driven many of the recent breakthroughs in artificial intelligence, yet the mechanisms underlying their success remain only partially understood. A key empirical clue is neural collapse (NC) – first documented by Papyan et al. (2020) – in which the last-layer feature vectors and classifier weights self-organise into a highly symmetric configuration during the terminal phase of training (TPT). While the reasons for the emergence of NC are still not fully understood, its impact on the behavior of a model is evident. For instance, Liu et al. (2023) induce NC to improve generalization in class-imbalanced training and Galanti et al. (2021) show that the emergence of NC improves transfer learning as well. Furthermore, the presence of NC has been connected to better out-of-distribution detection (Liu & Qin, 2023).

Theoretical explanations for NC have primarily relied on simplified models and assumptions (Mixon et al., 2022; Zhu et al., 2021) that have largely ignored the role of the optimizer, thereby suggesting that NC is universal across optimization methods. In this work, we challenge this assumption and

---

[*]First two authors share equal contribution.

demonstrate that the choice of optimizer plays a critical role in the emergence of NC. Concretely, we show that training with AdamW (Loshchilov & Hutter, 2019) does not lead to an NC solution, whereas training with SGD or Adam (Kingma & Ba, 2014) does. Through extensive experiments, we trace this back to how weight decay is applied in both optimizer and identify the coupling of weight decay as a necessity for the emergence of NC.

One major challenge in studying NC lies in the original metrics, which are difficult to track and analyze theoretically. These metrics were designed to quantify the progressive geometric alignment associated with NC and are expected to converge to zero in the idealized setting where NC holds as training time approaches infinity. However, under realistic training regimes, such as finite training epochs and learning rate decay, these metrics typically plateau at small but nonzero values. As a result, there is no rigorous criterion for determining whether NC has truly occurred.

This limitation motivates us to introduce a novel diagnostic metric, NC0, whose convergence to zero is necessary (though not sufficient) for NC. Unlike previous metrics, NC0 enables a more definitive assessment: if NC0 diverges during training, we can conclude that NC can not occur—even in cases where other NC metrics misleadingly converge to small positive values, creating an illusion of collapse. We discuss the peculiarity of interpreting NC metrics in practice later in Section 4.1. Furthermore, NC0 allows us to go beyond loss landscape analysis and theoretically derive convergence rates with which NC0 converges to zero.

**Contribution**   In this paper, we conduct extensive experiments – spanning over 3,900 training runs – to investigate the role of coupled weight decay in the emergence of NC. We identify coupled weight decay as a key driver of NC in realistic settings, extending recent theoretical insights (Pan & Cao, 2024; Jacot et al., 2024) that were limited to quasi-optimal solutions in simplified models. In particular, we show that the form of weight decay used in adaptive optimizers such as Adam (Kingma & Ba, 2014) and AdamW (Loshchilov & Hutter, 2019) critically affects whether NC emerges. Strikingly, while networks trained with Adam often exhibit NC, AdamW – despite its algorithmic similarity –fails to produce NC, with the corresponding metrics failing to converge to zero over time (Figure 1). This subtle yet consequential distinction has been largely overlooked in prior work. An overview of our theoretical contributions can be found in Table 1

In summary, we make the following contributions:

1. Across a wide range of experiments, we find that coupled weight decay is a necessary condition for NC to emerge in adaptive optimizers, such as Adam and Signum.

2. Furthermore, we show the accelerating effect of momentum on NC (beyond convergence of train loss) when trained with SGD, being the first result concerning momentum in the context of NC.

3. We support our empirical findings with the following theoretical statements on the new NC0 metric:
   - with SGD (with both coupled or decoupled weight decay), NC0 converges to zero at an exponential rate proportional to the weight decay;
   - with sign gradient descent (SignGD) with decoupled weight decay, a special case of AdamW, NC0 converges to some positive constant;
   - with SignGD with coupled weight decay, a special case of Adam, NC0 exhibits a non-monotonic trajectory, increasing before eventually decreasing. Using learning rate decreasing to zero, we show that NC0 also vanishes.

**Organization**   This paper is organized as follows. In Section 2, we recapitulate the four properties to characterize NC and introduce a novel NC property NC0. In Section 3 we present our main experimental results with theoretical support. Finally, Section 4 provides insights and discussions on the implications of our results.

**Notation**   We use $[K] = \{1, 2, \ldots, K\}$ to denote the index set for any integer $K \in \mathbb{N}$. For a matrix $\mathbf{W}$, we let $\mathrm{Vec}(\mathbf{W})$ denote the vectorization of $\mathbf{W}$ obtained by stacking its columns. The Frobenius inner product between two matrices $\mathbf{W}, \mathbf{W}'$ is denoted by $\langle \mathbf{W}, \mathbf{W}' \rangle = \mathrm{Tr}(\mathbf{W}^\top \mathbf{W}')$. With slight abuse of notation, we write $\|\mathbf{W}\| = \|\mathbf{W}\|_F$ for the Frobenius norm when $\mathbf{W}$ is a matrix, and $\|\mathbf{v}\| = \|\mathbf{v}\|_2$ for the Euclidean norm when $\mathbf{v}$ is a vector. In other words, $\|\mathbf{W}\| = \|\mathrm{Vec}(\mathbf{W})\|$.

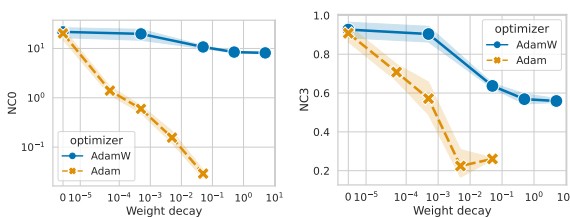

Figure 1: NC0 (left) and NC3 (right) metrics at the end of training. Lower values indicate stronger NC. AdamW shows consistently higher metrics than Adam. Averages computed over runs with varying learning rates and momentum; shaded regions show $\pm 1$ standard deviation. X-axis is log-scaled. Note that there are no values for Adam for WD larger than 0.05 as the model did not train due to over regularization.

We denote by $\mathbf{I}$ the identity matrix, by $\mathbf{1}$ the all-ones column vector, and by $\mathbf{J}$ the all-ones matrix, i.e., $\mathbf{J} = \mathbf{1}\mathbf{1}^\top$.

Table 1: Overview of our theoretical results on NC0.

| Result | Optimizers | Model | Convergence to 0? | learning rate |
|---|---|---|---|---|
| Theorem 3.1 | SGD with DWD | Any | yes, exponential | constant |
| Theorem 3.2 | SGD with CWD | Any | yes, exponential | constant |
| Theorem 3.3 | SignGD with DWD | UFM | yes | step-wise decay |
| Theorem 3.4 | SignGD with CWD | UFM | no | - |

## 2 NEURAL COLLAPSE

Neural collapse (NC), observed during the terminal phase of training (TPT) in deep neural networks (DNN), manifests itself through several geometric properties involving the last-layer features and weights in the $K$-class classification task:

$$\min_{\mathbf{W},\theta} \sum_{n=1}^{N} \ell(\mathbf{W} h_\theta(\mathbf{x}_n), y_n) + \frac{\lambda}{2}\|\mathbf{W}\|^2 + \frac{\lambda}{2}\|\text{Vec}(\theta)\|^2 \tag{1}$$

where $(\mathbf{x}_n, y_n)_{n=1}^N \subset \mathbb{R}^D \times [K]$ is the training set, $\mathbf{W} \in \mathbb{R}^{K \times P}$ is the last-layer weights, $h_\theta(\mathbf{x}_n) \in \mathbb{R}^P$ is the last-layer feature as the output of some backbone parameterized by $\theta$, $\ell : \mathbb{R}^K \times [K] \to [0, \infty)$ is the loss function, and $\lambda > 0$ is the L2-regularization constant.

These properties, formalized by their corresponding metrics in the original paper Papyan et al. (2020), are:

1. **NC1 - Variability Collapse:** Features collapse to their respective class means, indicating that within-class variability vanishes.

2. **NC2 - Convergence of Centered Class Means to Simplex ETF:** Centered Class means converge to a simplex equiangular tight frame (ETF).

3. **NC3 - Convergence to Self-Duality:** Rows of the last-layer weight $\mathbf{W} \in \mathbb{R}^{K \times P}$ align with the columns of the class means, creating a dual relationship between weights and features.

4. **NC4 - Simplification to Nearest-Class-Center:** The classifier's decision boundaries are simplified to those of a nearest-class-mean (NCC) classifier.

A solution satisfying all of these properties is referred to as a *NC solution*. In addition to these prior NC properties, we introduce another novel NC property **NC0**, whose convergence to zero is a necessary condition (though not sufficient) for NC.

**NC0 - Zero Row Sum of Last-Layer Weight:** The row sum of the last-layer weight $\mathbf{W}$ in the model converges to zero.

The first observation is that NC0 is a necessary condition for NC2 and NC3:

**Proposition 2.1.** *NC2 and NC3 implies NC0.*

*Proof.* For each class $k \in [K]$, we define the class mean $\mu_k = \frac{1}{|\{n:y_n=k\}|} \sum_{n:y_n=k} h_\theta(\mathbf{x}_n) \in \mathbb{R}^P$ and the centered class mean $\bar{\mu}_k = \mu_k - \frac{1}{N} \sum_{n=1}^N h_\theta(\mathbf{x}_n)$. We concatenate them into a matrix $\mathbf{M} = (\bar{\mu}_k)_{k=1}^K \in \mathbb{R}^{P \times K}$ with $\mathbf{M1} = 0$, since we centered the class means. By NC2, $\mathbf{M}$ converge to a simplex ETF in the ambient space $\mathbb{R}^P$, meaning $\mathbf{M}/\|\mathbf{M}\|_F \to \mathbf{QM}^*$ where $\mathbf{M}^* \in \mathbb{R}^{K \times K}$ is a unit matrix with columns forming a $K$-simplex EFT in $\mathbb{R}^K$ and $\mathbf{Q} \in \mathbb{R}^{P \times K}$ is the isometric injection map into the ambient space. Since $\mathbf{M1} = \mathbf{0}$ and $\mathbf{Q}$ is injective, the unit matrix $\mathbf{M}^*$ has to be in the form: $\mathbf{M}^* \stackrel{\text{def.}}{=} \mathbf{P} \frac{1}{\sqrt{K-1}} \left(\mathbf{I} - \frac{1}{K}\mathbf{J}\right)$ for some orthogonal matrix $\mathbf{P}$. But it can be absorbed into $\mathbf{Q}$ as the matrix $\mathbf{QP}$ is still an isometric injection. Hence, without loss of generality, we assume $\mathbf{M}^* \stackrel{\text{def.}}{=} \frac{1}{\sqrt{K-1}} \left(\mathbf{I} - \frac{1}{K}\mathbf{J}\right)$ and hence

$$\mathbf{M}^\top \mathbf{M}/\|\mathbf{M}^\top \mathbf{M}\|_F^2 \to (\mathbf{QM}^*)^\top \mathbf{QM}^* = (\mathbf{M}^*)^2 = \mathbf{M}^*.$$

On the other hand, NC3 states that $\mathbf{M}/\|\mathbf{M}\| - \mathbf{W}^\top/\|\mathbf{W}\| \to 0$ as $t \to \infty$. Hence we have $\frac{\mathbf{WW}^\top}{\|\mathbf{W}\|_F^2} - \mathbf{M}^* \to 0$ as $t \to \infty$. Now note that $\mathbf{1}^\top \mathbf{M}^* \mathbf{1} = 0$, hence $\|\mathbf{W}^\top \mathbf{1}\|^2 = \mathbf{1}^\top \mathbf{WW}^\top \mathbf{1} \to 0$. Note that the last line holds if and only if NC0 holds. □

NC0 offers two key advantages. First, it serves as a diagnostic tool: if NC0 does not converge, then at least one of NC2 or NC3 must fail, providing a clear signal that neural collapse cannot occur. Second, NC0 is more mathematically tractable than the original NC metrics, whose dynamics are difficult to analyze and remain underexplored. As we demonstrate in Section 3, NC0's evolution during training can be reliably tracked and used to explain empirical trends observed across different optimizers. In addition, our extensive experiments also show that NC0 is correlating well with prior NC metrics, particularly for small learning rates (see Figure 2). For a more detailed explanation and formal definitions of NC properties and their metrics, we refer the reader to Section B.

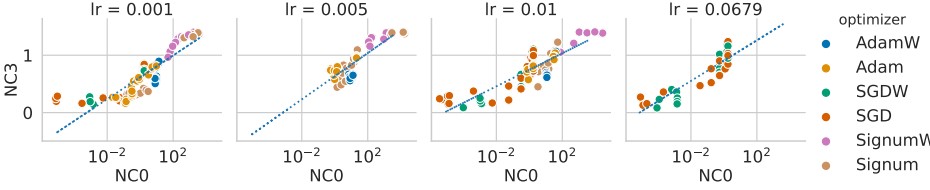

Figure 2: NC0 weakly correlates with NC3 across different optimizers and learning rates. Details on the regression fit can be found in Appendix D.3

# 3 MAIN RESULT

## 3.1 EXPERIMENTAL SETUP

We conducted extensive experiments training a ResNet9 and VGG9 using various optimizers, including Adam, AdamW, SGD, SGD with decoupled weight decay (SGDW), Signum (Bernstein et al., 2018), and Signum with decoupled weight decay (SignumW) trained on MNIST, FashionMNIST and Cifar10. Every optimizer is trained with three different learning rates (LR), six different values of momentum, and six different values of weight decay to also control the effect of hyperparameters on the emergence of NC. This resulted in a total of $2 \times 3 \times 6 \times 108 = 3,888$ training runs. Note that we only keep runs with reasonably high training accuracy. Too large weight decay over regularize the model and the model does not train anymore. Thus, the number of valid training runs is actually smaller than 3,888. All networks were trained for 200 epochs using a batch size of 128, with the learning rate being decayed by a factor of 10 after one-third and two-thirds of the training duration, as described in the original work by Papyan et al. (2020). In addition, we conducted ablation studies to control for the number of training epochs and to verify that the results also hold for unconstrained feature models (UFM)[1], leading to a total of over 3,900+ training runs. Further details and all experimental results can be found in Appendix D. Ablation studies on the effect of training epochs can be found in Appendix D.4.1

---

[1]see Appendix C.5 for an introduction to UFM.

Table 2: Final NC metrics for the same setting as in Figure 6, following the setup of Papyan et al. (2020). Lower values (↓) indicate stronger neural collapse. Values in parentheses represent percentages relative to the metric at initialization.

| Optimizer | NC0↓ | NC1↓ | NC2↓ | NC3↓ |
|---|---|---|---|---|
| SGD | **2.14e-04** ($< -99.5\%$) | 0.05 ($-99.3\%$) | **0.29** ($-63.0\%$) | **0.35** ($-75.1\%$) |
| SGDW | 0.55 ($-68.9\%$) | 0.26 ($-96.3\%$) | 0.46 ($-42.4\%$) | 0.80 ($-43.5\%$) |
| Adam | 0.34 ($-80.6\%$) | **0.04** ($-99.5\%$) | 0.29 ($-63.9\%$) | 0.29 ($-79.5\%$) |
| AdamW | 5.33 ($\gg 100\%$) | 0.20 ($-97.2\%$) | 0.54 ($-32.4\%$) | 0.78 ($-45.2\%$) |
| Signum | 0.78 ($-55.3\%$) | 0.13 ($-98.1\%$) | 0.50 ($-36.8\%$) | 0.58 ($-59.0\%$) |
| SignumW | 3185.69 ($\gg 100\%$) | 0.30 ($-95.7\%$) | 1.15 ($+44.2\%$) | 1.40 ($-1.2\%$) |

## 3.2 WEIGHT DECAY IS ESSENTIAL AND MOMENTUM ACCELERATES NC

Our experiments show that weight decay is necessary to reduce the NC metric across all optimizers and hyperparameter settings, as shown in Figure 3 for Signum and SGD, and earlier in Figure 1 for Adam and AdamW as well as in our ablation studies in Appendix D.4.1 and Appendix D.4.6. While the experiments cannot fully exclude the possibility that NC can be achieved eventually in the asymptotic limit without weight decay, we argue that WD is essential to observe the emergence of NC in *practical finite-length training settings on realistic models*[2].

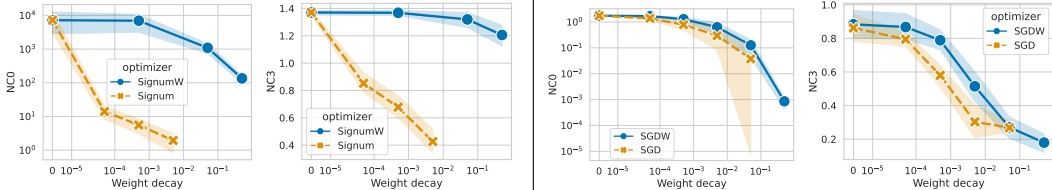

Figure 3: NC0 and NC3 metrics at the end of training for a ResNet9 trained on FashionMNIST for Signum and SignumW (left side) and SGD and SGDW (right side). Shaded area refers to one standard deviation across all trainings run with corresponding optimizer. Note that there are fewer values for Signum and SGD as the model did not train due to over regularization for too large WD.

From the figures, we can conclude that larger weight decay leads to a stronger decrease of NC metrics. In particular, we show that adaptive optimizers with decoupled weight decay have much larger NC metrics, which are strictly away from zero, showing no sign of NC. In addition, we show empirically that momentum amplifies the effect of weight decay on the decrease of NC metrics in SGD, as shown in the heatmap in Figure 5. This implies that one achieves a decrease in the NC metrics both by increasing weight decay for fixed momentum or by increasing momentum for fixed non-zero weight decay. The effect of momentum on the NC metrics becomes larger for larger values of weight decay. We remark that this goes beyond the acceleration of convergence of the train loss, as we study in an ablation study in Appendix D.4.5. In particular, we show in Figure 4 that two training runs with different momentum and otherwise same hyperparameters can reach the same train loss, while reaching different NC metrics. This indicates that they have converged to solutions with very different geometric structure.

The experimental results are complemented by Theorem 3.1 and Theorem 3.2 showing that NC0 converges to 0 with an exponential rate trained with SGD, which is proportional to momentum and weight decay, highlighting that NC cannot be achieved without weight decay and that momentum accelerates the convergence of NC metrics.

**Theorem 3.1** (SGD with decoupled weight decay promotes NC0). *Assume a model of the form $f(\mathbf{W}, \theta, x) = \mathbf{W}h_\theta(x)$ is trained using cross-entropy loss with stochastic gradient descent (SGD) and momentum $\beta \in [0, 1)$, weight decay $\lambda \in [0, 1)$, and learning rate $\eta > 0$ on* all *parameters $\theta, \mathbf{W}$.*

---

[2]We note that Ji et al. (2021) show both theoretically and empirically the emergence of NC on the unconstrained layer-peeled model (ULPM) objective under gradient flow without weight decay.

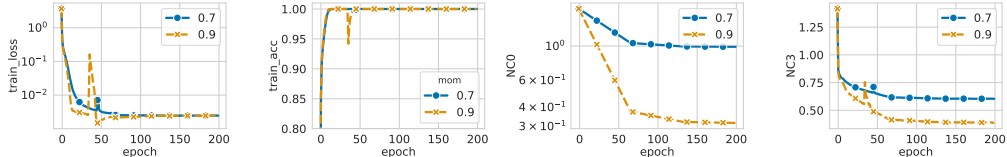

Figure 4: Train loss, train accuracy and NC metrics for fixed WD=0.005 and mom=0.7 and 0.9. Although both runs converge to almost exactly the same train loss, the final NC metrics differ considerably. Plots including NC1 and NC2 can be found in Figure 24.

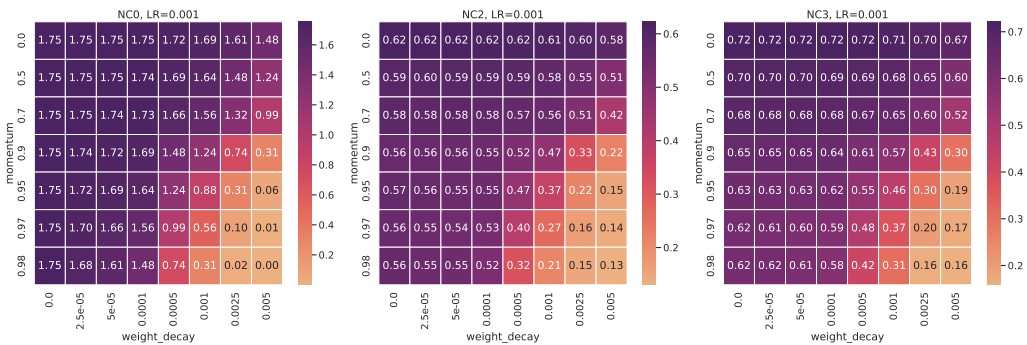

Figure 5: Heatmap of NC0, NC2 and NC3 for varying values of momentum and weight decay on ResNet9 trained on FashionMNIST with SGD.

*For instance, the last-layer weights* $\mathbf{W}$ *are updated according to:*

$$\mathbf{V}_{t+1} = \beta\mathbf{V}_t + \nabla_{\mathbf{W}_t}L_{\text{CE}},$$
$$\mathbf{W}_{t+1} = (1 - \eta\lambda)\mathbf{W}_t - \eta\mathbf{V}_{t+1}.$$

*If* $0 < \eta\lambda < 2$*, then the NC0 metric* $\alpha_t \coloneqq \frac{1}{K}\|\mathbf{W}_t^{\top}\mathbf{1}\|_2^2$ *decays exponentially to zero in t.*

*Proof.* The key observation is that the row sum of the loss gradient $\nabla L_{\text{CE}}(\mathbf{W}_t)^{\top}\mathbf{1}_K$ is zero, which largely simplifies the NC0 metric to only be dependent on the weight decay $\lambda$ and momentum $\beta$. For the details of the proof, please refer to Subsection E in the Appendix. $\square$

**Theorem 3.2** (SGD with coupled weight decay promotes NC0). *Assume a model of the form* $f(\mathbf{W}, \theta, x) = \mathbf{W}h_\theta(x)$ *is trained using cross-entropy loss with stochastic gradient descent (SGD) and momentum* $\beta \in [0, 1)$*, weight decay* $\lambda \in [0, 1)$*, and learning rate* $\eta > 0$ *on* all *parameters* $\theta, \mathbf{W}$*. For instance, the last-layer weights* $\mathbf{W}$ *are updated according to:*

$$\mathbf{V}_{t+1} = \beta\mathbf{V}_t + \nabla_{\mathbf{W}_t}L_{\text{CE}} + \lambda\mathbf{W}_t,$$
$$\mathbf{W}_{t+1} = \mathbf{W}_t - \eta\mathbf{V}_{t+1}.$$

*If* $0 < \eta\lambda < 2(1 + \beta)$*, then the NC0 metric* $\alpha_t \coloneqq \frac{1}{K}\|\mathbf{W}_t^{\top}\mathbf{1}\|_2^2$ *decays exponentially to zero in t.*

*Proof.* Similar to the proof of Theorem 3.1 For the details of the proof, please refer to Subsection E in the Appendix. $\square$

### 3.3 WEIGHT DECAY COUPLING MATTERS

While weight decay has been theoretically shown to be essential for NC in prior works (Pan & Cao, 2024; Jacot et al., 2024), these works ignore how weight decay is applied by treating $L_2$-regularization of the gradient and applying weight decay directly on parameters as equivalent. However, we note that this equivalency only holds for vanilla SGD and not for adaptive optimizers, such as Adam or AdamW, nor when momentum is applied. In particular, our experiments reveal that NC does not emerge under SignumW and AdamW under realistic settings. This highlights the crucial role of coupled weight

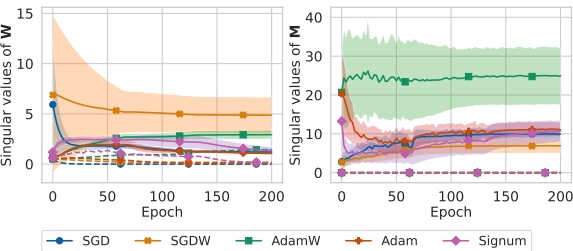

Figure 6: NC metrics throughout training on a ResNet9 trained on FashionMNIST.

decay – that is $L_2$-regularization applied directly within the gradient update – as a requirement for NC. This subtle yet important distinction has been largely overlooked in prior literature.

Importantly, tracking the evolution of the NC metrics (Figure 6) and the singular values of centered class means $\mathbf{M}$ and the last-layer weight $\mathbf{W}$ (Figure 7) throughout training (here shown for a ResNet9 trained on FashionMNIST), one can see that using adaptive optimizers with decoupled weight decay leads to fundamentally different dynamics of the NC metrics and singular values despite all models reaching TPT, where training error is (almost) zero.

Figure 7: Singular values of last-layer weights $\mathbf{W}$ (left) and centered class means $\mathbf{M}$ (right) throughout training. The dotted line corresponds to the smallest singular value and the full line corresponds to the average singular value, excluding the smallest singular value. Singular values for SignumW are out-of-range and are shown in Figure 29 in the appendix.

Specifically, Figure 7 shows that the smallest singular value of $\mathbf{W}$ increases during training with AdamW and SignumW, indicating failure to satisfy NC3. Additionally, NC0 and the nonzero singular values of $\mathbf{M}$ grow throughout training and exhibit high variance, suggesting that NC2 is also less well-fulfilled in these settings.

In Figure 6, we further observe that SGD and Adam achieve the lowest NC metric values, while AdamW, SignumW, and SGDW saturate early at much higher levels. Although the NC metrics for Signum are slightly larger than for SGD and Adam, they continue to decrease over time, suggesting potential convergence to NC under longer training.

Finally, our experiments in Figure 1 and Figure 3 demonstrate that the NC0 and NC3 metrics of AdamW and SignumW remain significantly larger than those of Adam and Signum, even when using weight decay several orders of magnitudes higher. This indicates that models trained with AdamW or SignumW are consistently farther from achieving NC. Note that the NC metrics for SGD and SGDW remain relatively close, consistent with our theoretical results in Theorem 3.1 and Theorem 3.2, while the gap between coupled and decoupled weight decay has a more pronounced effect in adaptive optimizers than in SGD. This suggests the effect is not simply due to greater weight decay accumulation through momentum but stems from a deeper interaction with the optimization dynamics.

### 3.4 INTERPOLATING ADAMW AND ADAM

To further investigate why AdamW fails to exhibit neural collapse (NC) while Adam does, we conducted an ablation study by "interpolating" between the two optimizers. Specifically, we implemented a variant that combines both coupled weight decay (as in Adam) and decoupled weight decay (as

in AdamW). For each run, we varied the strength of the coupled weight decay while adjusting the decoupled component such that the total weight decay remained fixed at 0.0005. The momentum was set to 0.9 across all configurations.

As shown in Figure 8, increasing the coupled component leads to a smooth improvement in NC metrics—particularly NC0, NC2, and NC3—while the validation accuracy remains largely unaffected. This experiment suggests that coupled weight decay is a critical factor in enabling neural collapse, yet it is not strictly necessary for achieving strong generalization performance, as all configurations yield similar validation accuracy. This strengthens a point raised earlier about the limitations of NC to understand generalization Hui et al. (2022).

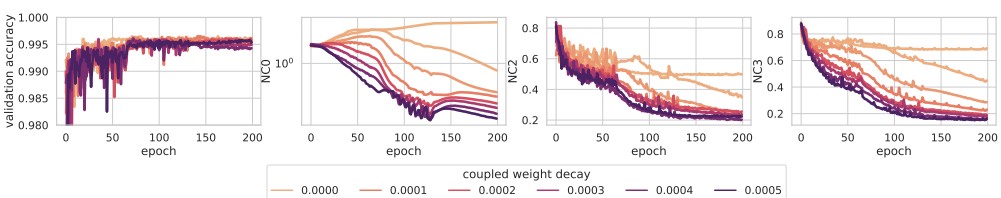

Figure 8: Interpolating Adam and AdamW by varying the coupled and decoupled weight decay. Total weight decay was fixed to $0.0005$. Note that coupled weight decay = 0 is equivalent to AdamW and coupled weight decay = 0.0005 is equivalent to Adam. Experiments trained on ResNet9 with MNIST.

This observation is supported by our theoretical results in Theorem 3.3 and Theorem 3.4, which show that SignGD with decoupled weight decay fails to satisfy NC0 and therefore cannot converge to a neural collapse solution, whereas SignGD with coupled weight decay exhibits different behaviour. We note that SignGD corresponds to a special case of Adam and AdamW when the parameters $\beta_1$, $\beta_2$, and $\varepsilon$ are set to zero.

**Theorem 3.3** (Sign GD with decoupled weight decay avoids NC0). *Consider sign GD with (decoupled) weight decay $\lambda > 0$ and step size $\eta > 0$ on the UFM loss $L_{CE}(\mathbf{W}\mathbf{H}, \mathbf{I}) = \sum_{n=1}^{N} L_{CE}(\mathbf{W}\boldsymbol{h}_n, \mathbf{e}_n)$, where the feature $\mathbf{H} = \mathbf{M}^*$ is fixed to an NC solution and only the weight $\mathbf{W}$ is trained:*

$$\mathbf{W}_{t+1} = \mathbf{W}_t - \eta(sign(\nabla_{\mathbf{W}_t} L_{CE}) + \lambda \mathbf{W}_t)$$

*Then the NC0 metric $\alpha = \|\mathbf{W}_t^\top \mathbf{1}_K\|_2^2$ increases monotonically from zero to the limit:*

$$\lim_{t \to \infty} \alpha_t = \frac{(K-2)^2}{\lambda^2}.$$

*In particular, $\alpha_t$ does not vanish as $t \to \infty$.*

*Proof idea:* The key observation is that the signed loss gradient $sign(\nabla L_{\mathbf{CE}}(\mathbf{W}_t))$ in this setting is constant in $t$, simplifying the following computation. See Appendix E for the full proof. □

**Theorem 3.4** (Sign GD with coupled weight decay can lead to NC0). *Consider sign GD with (coupled) weight decay $\lambda > 0$ and step size $\eta > 0$ on the UFM loss $L_{CE}(\mathbf{W}\mathbf{H}, \mathbf{I}) = \sum_{n=1}^{N} L_{CE}(\mathbf{W}\boldsymbol{h}_n, \mathbf{e}_n)$, where the feature $\mathbf{H} = \mathbf{M}^*$ is fixed to an NC solution and only the weight $\mathbf{W}$ is trained:*

$$\mathbf{W}_{t+1} = \mathbf{W}_t - \eta(sign(\nabla_{\mathbf{W}_t} L_{CE} + \lambda \mathbf{W}_t))$$

*We initialize $\mathbf{W}_0 = 0 \in \mathbb{R}^{K \times K}$ and define the covariance matrix $\mathbf{C}_t = \mathbf{W}_t \mathbf{W}_t^\top$ and the scalar $\alpha_t = \langle \mathbf{C}_t, \hat{\mathbf{J}} \rangle_F$ where $\hat{\mathbf{J}} = \frac{1}{K} \mathbf{1} \mathbf{1}^\top$. Then there exists a learning rate decay scheme $\eta = \eta(t) \xrightarrow[t \to \infty]{} 0$ such that $\alpha_t \xrightarrow[t \to \infty]{} 0$.*

*Proof.* See Appendix E. □

The key difference between the results of Theorem 3.3 and Theorem 3.4 lies in how coupled weight decay affects the signed gradient during training. As the weight norm $\|\mathbf{W}\|$ increases, the coupled decay term can eventually flip the sign of the gradient, altering the trajectory of the NC0 metric $\alpha_t$.

Figure 9: Training dynamic of NC0 with optimizers SGD, Adam, AdamW, Adam0 ($\beta_1 = \beta_2 = 0$), AdamW0 ($\beta_1 = \beta_2 = 0$). For AdamW and SignSGD the inlay shows the NC0 metric more detailed for the last 2000 steps. Note that 5 steps correspond to one training epoch.

Initially, $\alpha_t$ grows at a similar rate in both cases, but their behaviors diverge once the decay term becomes dominant.

To illustrate this effect, we conducted a small-scale experiment using a simple MLP on a separable dataset with various optimizers. As shown in Figure 9, SignSGD displays non-monotonic dynamics in $\alpha_t$, while SignSGDW exhibits steady convergence to a positive value. Similar patterns appear in Adam and AdamW, though more smoothed due to their adaptive updates.

## 4    DISCUSSION AND LIMITATIONS

In this section, we discuss new insights, additional considerations and limitations from the main results in Section 3. Additionally, we explore potential follow-up research directions that could provide theoretical explanations or extend our experiments to broader settings.

### 4.1    INTERPRETING NC METRICS IN PRACTICE

While NC is defined by the convergence of all NC metrics to zero in the limit, practical experiments never achieve exact zeros. Since NC is inherently a continuous rather than discrete phenomenon, it becomes necessary to define what constitutes the presence of NC in practice. This important issue has not been thoroughly addressed in the existing literature.

A further complication is that different NC metrics operate on different scales and these scales vary across settings of architectures and datasets. For example, in our experiments, the smallest observed values for NC2 and NC3 are on the order of 0.1, whereas NC1 can reach values an order of magnitude smaller.

In this work, we therefore refer to the emergence of NC in terms of relative strength. Specifically, we use the NC metric values at initialization as a baseline for models that do not exhibit NC, and use the smallest values achieved across all experiments as a reference point for models that do. This framing allows us to discuss the strength of NC emergence across different optimizers and settings.

### 4.2    THE REDUNDANT NC4 PROPERTY

Readers may notice that we omit NC4 from the results in Section 3. This is because we observed that NC4 is consistently satisfied whenever the training accuracy approaches 100%, regardless of whether the other NC metrics (NC1–NC3) exhibit collapse. As shown in Figure 57, NC4 is largely uncorrelated with the other metrics. To maintain a clearer and more focused presentation, we therefore exclude NC4 from our main analysis.

### 4.3    PARTIAL NEURAL COLLAPSE

Another subtlety we observe is what we term *partial neural collapse*. As shown in Table 3, AdamW can achieve minimal values for NC1 and NC2 among all optimizers, even while NC0 diverges and NC3 is not satisfied. This indicates that NC properties may not always emerge jointly, contrary to the original claim in Papyan et al. (2020). Understanding the theoretical conditions under which only a subset of NC properties holds remains an intriguing open question.

Table 3: Final NC metrics for the run with the smallest absolute NC3 metric and $> 99\%$ training accuracy for each optimizer. Lower values ($\downarrow$) indicate stronger neural collapse. Values in parentheses represent percentages relative to the metric at initialization. Hyperparameters used for each optimizer can be found in Table 5.

| Optimizer | $NC0_\downarrow$ | $NC1_\downarrow$ | $NC2_\downarrow$ | $NC3_\downarrow$ |
|---|---|---|---|---|
| SGD | **1.53e-05** ($< -99.5\%$) | 0.02 ($< -99.5\%$) | 0.19 ($-75.8\%$) | 0.13 ($-90.9\%$) |
| SGDW | 1.54e-04 ($< -99.5\%$) | **0.01** ($< -99.5\%$) | 0.15 ($-81.7\%$) | **0.10** ($-92.7\%$) |
| Adam | 0.12 ($< -93.2\%$) | 0.04 ($-99.5\%$) | 0.23 ($-71.6\%$) | 0.17 ($-88.2\%$) |
| AdamW | 8.09 ($\gg 100\%$) | **0.01** ($< -99.5\%$) | **0.14** ($-82.1\%$) | 0.49 ($-65.1\%$) |

### 4.4 LIMITATIONS OF THEORETICAL SUPPORT

Our experiments on Adam and AdamW are conducted on realistic models and datasets, whereas our theoretical results (Theorem 3.3, Theorem 3.4) focus on a simplified setting: SignGD applied to the unconstrained feature model. While this restricted setup already demonstrates that AdamW fails to achieve NC, it does not fully capture the complexity of deep neural networks or adaptive optimizers in practice. Nevertheless, we believe our proof techniques could be extended to explain why Adam may lead to NC in more general settings. Moreover, our theoretical analysis is limited to the training dynamics of NC0, chosen for its analytical tractability and strong empirical correlation with other NC metrics. A full theoretical understanding of NC1–NC3 under realistic optimization dynamics remains an open challenge, and we leave this direction for future work.

### 4.5 FUTURE RESEARCH

Other than the topic we have discussed in the previous subsections, our findings also open other intriguing avenues for future research.

- Empirical studies should be expanded to include larger models, such as Vision Transformers (ViTs) and DenseNets, as well as more diverse datasets, to assess the broader generality of our findings. Our preliminary results on ViT are available in Appendix D.4.10, and largely confirm our findings also extend to Transformers.

- Due to computational constraints, our study only analyzed NC properties in the last layer. However, previous works (Masarczyk et al., 2023; Rangamani et al., 2023) suggest that these properties may also manifest in intermediate layers. Investigating NC behavior across different depths could provide further insights into hierarchical feature representations.

- In addition to the optimizers (SGD, Adam, AdamW, Signum) studied in this work, novel first-order methods such as Lion (Chen et al., 2023) and Mars (Yuan et al., 2024), and second-order methods, such as Shampoo (Gupta et al., 2018), SOAP (Vyas et al., 2024) and Muon (Jordan et al.) demonstrated promising improvements in convergence and generalization. However, their effects on NC remain largely unexplored.

## 5 CONCLUSION

In this paper we have conducted an extensive number of experiments to elucidate the role of the optimization algorithm in the emergence of the neural collapse (NC) phenomenon. In particular, our experiments consistently show that coupled weight decay is necessary for achieving small NC metrics. While the role of weight decay in the context of NC has been studied in the literature before, this is the first paper distinguishing between coupled and decoupled weight decay. Moreover, our theoretical results show that the resulting training dynamics differ considerably and one needs to take this into account. These findings underscore the limitations of existing theoretical frameworks, which have studied NC mainly under gradient flow or gradient descent, and highlight the need for further investigation into the interplay between optimizers and NC.

ACKNOWLEDGMENTS

WO acknowledges that this research was partially funded by National Science Centre, Poland grant no 2022/45/N/ST6/04098.

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

# Appendix

## A LLM USAGE STATEMENT

We disclaim that we have used Large Language Models to refine a few sentences and additionally as a proxy of a search engine to retrieve additional related work.

The appendix is organized as follows. In Section B, we formally define the neural collapse (NC) phenomenon and introduce the metrics used in the experiments presented in the main text. In Appendix C, we review prior works related to our paper. Section D provides detailed descriptions and additional observations from our experiments. In Section E, we present the full proof of the theorems stated in the main text.

## B NC METRICS

Neural collapse (NC), discovered by Papyan et al. (2020), is a striking phenomenon observed during the terminal phase of training (TPT) deep neural networks (DNN) for multi-class classification tasks, particularly when trained with cross-entropy (CE) loss. Formally, let the (trained) last-layer features of the DNN be denoted by $\mathbf{h}_n$, and concatenate them into a matrix $\mathbf{H} \in \mathbb{R}^{p \times N}$, where $p$ is the width of the last layer and $N$ is the number of training samples indexed by $n$. The output logits of the network are then computed as $\mathbf{W}_L \mathbf{H} \in \mathbb{R}^{K \times N}$, where $\mathbf{W}_L \in \mathbb{R}^{K \times p}$ is the last-layer weight, $\mathbf{b} \in \mathbb{R}^K$ is the bias vector, and $K$ is the number of classes. [3]

The DNN is trained using the CE loss computed on the logits:

$$\text{CE}(\mathbf{W}_L, \mathbf{H}) = -\sum_{n=1}^{N} \log \left( \frac{\exp(\mathbf{W}_L \mathbf{h}_n)_{y_n}}{\sum_{k=1}^{K} \exp(\mathbf{W}_L \mathbf{h}_n)_k} \right),$$

where $y_n \in [K]$ denotes the class label index of the feature vector $\mathbf{h}_n$. Let $\mathcal{C}_k \stackrel{\text{def.}}{=} n \in [N] : y_n = k$ be the index set of data points belonging to class $k \in [K]$. In this paper, we assume that the classes are balanced, i.e., $|\mathcal{C}_k|$ is equal for all $k \in [K]$. For the effects of class imbalance on NC, we refer the reader to Han et al. (2022); Thrampoulidis et al. (2022); Behnia et al. (2023).

Let $\boldsymbol{\mu}_k \stackrel{\text{def.}}{=} \frac{1}{|\mathcal{C}_k|} \sum_{n \in \mathcal{C}_k} \mathbf{h}_n$ be the class mean for each class $k$. The global mean of all classes is given by $\boldsymbol{\mu}_G = \frac{1}{K} \sum_{k=1}^{K} \boldsymbol{\mu}_k$ and centered class means are defined as $\bar{\boldsymbol{\mu}}_k = \boldsymbol{\mu}_k - \boldsymbol{\mu}_G$. Let the between-class covariance $\boldsymbol{\Sigma}_B \in \mathbb{R}^{p \times p}$ and the within-class covariance $\boldsymbol{\Sigma}_W \in \mathbb{R}^{p \times p}$ be:

$$\boldsymbol{\Sigma}_B = \frac{1}{K} \sum_{k=1}^{K} \bar{\boldsymbol{\mu}}_k \bar{\boldsymbol{\mu}}_k^\top,$$

$$\boldsymbol{\Sigma}_W = \frac{1}{K} \frac{1}{N} \sum_{k=1}^{K} \sum_{n=1}^{N} (\mathbf{h}_n^k - \boldsymbol{\mu}_k)(\mathbf{h}_n^k - \boldsymbol{\mu}_k)^\top,$$

where $\mathbf{h}_n^k$ correspond to the feature vectors of class $k$.

We also concatenate the centered class means into a matrix $\mathbf{M} \stackrel{\text{def.}}{=} (\bar{\boldsymbol{\mu}}_1, ..., \bar{\boldsymbol{\mu}}_K) \in \mathbb{R}^{p \times K}$.

With these definitions in place, we now conceptually outline the NC properties and introduce corresponding metrics to quantitatively measure these properties in our experiments.

---

[3]For simplicity, we interchangeably refer to an input $\mathbf{x} \in \mathbb{R}^d$ and its corresponding last-layer feature $\mathbf{h} \in \mathbb{R}^p$ after the parameters of the network have converged during TPT and the mapping $\mathbf{x} \mapsto \mathbf{h}$ is fixed.

**NC1 - Variability Collapse** The first property of neural collapse (NC1) describes the collapse of features to their respective class means. Formally, this means that the distance between a feature vector $\mathbf{h}_n$ and its corresponding class mean $\boldsymbol{\mu}_k$ approaches zero:

$$\|\mathbf{h}_n - \boldsymbol{\mu}_k\|_2 \to 0, \forall k \in [K], \ n \in \mathcal{C}_k.$$

A corresponding metric is defined as Zhu et al. (2021); Kothapalli (2023); Ammar et al. (2024):

$$\mathcal{NC}1 \overset{\text{def.}}{=} \frac{1}{K} \text{Tr}[\boldsymbol{\Sigma}_W \boldsymbol{\Sigma}_B^\dagger] \tag{2}$$

where $\dagger$ denotes the Moore-Penrose pseudo-inverse.

**NC2 - Convergence of Class Means to Simplex ETF** The second property of neural collapse (NC2) describes the convergence of class means to a simplex equiangular tight frame (ETF), where the angles between the means are maximally symmetric. Formally, this property can be expressed as:

$$\begin{cases} \|\bar{\boldsymbol{\mu}}_j\|_2 - \|\bar{\boldsymbol{\mu}}_k\|_2 & \to 0 \\ \left\langle \frac{\bar{\boldsymbol{\mu}}_j}{\|\bar{\boldsymbol{\mu}}_j\|_2}, \frac{\bar{\boldsymbol{\mu}}_k}{\|\bar{\boldsymbol{\mu}}_k\|_2} \right\rangle & \to \frac{K}{K-1}\delta_{jk} - \frac{1}{K-1}, \end{cases} \forall j, k \in [K].$$

To measure this property, we define two metrics capturing the equinormality and equiangularity of the centered class means Papyan et al. (2020); Ammar et al. (2024):

$$\mathcal{NC}2_n = \frac{\text{std}_k\{\|\bar{\boldsymbol{\mu}}_k\|_2\}}{\text{avg}_k\{\|\bar{\boldsymbol{\mu}}_k\|_2\}}; \tag{3}$$

$$\mathcal{NC}2_a = \text{avg}_{k \neq k'} \left| \left\langle \frac{\bar{\boldsymbol{\mu}}_k}{\|\bar{\boldsymbol{\mu}}_k\|_2}, \frac{\bar{\boldsymbol{\mu}}_{k'}}{\|\bar{\boldsymbol{\mu}}_{k'}\|_2} \right\rangle + \frac{1}{K-1} \right|. \tag{4}$$

Here, $\text{std}_\bullet(\cdot)$ and $\text{avg}_\bullet(\cdot)$ denote the standard deviation and mean, respectively, over the specified index.

An alternative metric for NC2, introduced by Kothapalli (2023), directly measures the deviation of the centered class means from a simplex ETF:

$$\mathcal{NC}2 \overset{\text{def.}}{=} \frac{1}{K^2} \left\| \frac{\mathbf{M}^\top \mathbf{M}}{\|\mathbf{M}^\top \mathbf{M}\|_F} - \mathbf{M}^* \right\|_F \tag{5}$$

where

$$\mathbf{M}^* \overset{\text{def.}}{=} \frac{1}{\sqrt{K-1}} \left( \mathbf{I}_K - \frac{1}{K} \mathbf{J}_K \right),$$

$\mathbf{I}_K \in \mathbb{R}^{K \times K}$ is the identity matrix and $\mathbf{J} \in \mathbb{R}^{K \times K}$ is the matrix of ones. Note that $\mathcal{NC}2_n, \mathcal{NC}2_a \to 0 \iff \mathcal{NC}2 \to 0$.

**NC2W - Convergence of Weight Rows to Simplex ETF** In addition to NC2, we define a related property, NC2W, which describes the convergence of the rows of the last-layer weights $\mathbf{W}_L \in \mathbb{R}^{K \times p}$ to a simplex ETF. If the third NC property, NC3 (described later), holds, then NC2 and NC2W are equivalent. However, to study partial NC, it is essential to decouple these properties and measure NC2 and NC2W separately.

To measure NC2W, Zhu et al. (2021) introduced the following metric:

$$\mathcal{NC}2\mathcal{W} \overset{\text{def.}}{=} \frac{1}{K^2} \left\| \frac{\mathbf{W}_L \mathbf{W}_L^\top}{\|\mathbf{W}_L \mathbf{W}_L^\top\|_F} - \mathbf{M}^* \right\|_F. \tag{6}$$

While this metric measures the overall alignment of $\mathbf{W}_L$ with a simplex ETF, it does not account for the equinormality and equiangularity of the rows of $\mathbf{W}_L$. To address this, we introduce the following metrics:

$$\mathcal{NC}2\mathcal{W}_n = \frac{\text{std}_k\{\|\mathbf{w}_k\|_2\}}{\text{avg}_k\{\|\mathbf{w}_k\|_2\}} \tag{7}$$

$$\mathcal{NC}2\mathcal{W}_a = \text{avg}_{k \neq k'} \left| \left\langle \frac{\mathbf{w}_k}{\|\mathbf{w}_k\|_2}, \frac{\mathbf{w}_{k'}}{\|\mathbf{w}_{k'}\|_2} \right\rangle + \frac{1}{K-1} \right| \tag{8}$$

where $\mathbf{w}_k^\top \in \mathbb{R}^p$ is the $k$-th row of $\mathbf{W}_L$.

**NC2M - Convergence of Product to Simplex ETF**  Finally, Zhu et al. (2021); Kothapalli (2023) proposed a metric that interpolates between NC2 and NC2W: [4]

$$\mathcal{NC}2\mathcal{M} \overset{\text{def.}}{=} \frac{1}{K^2} \left\| \frac{\mathbf{W}_L \mathbf{M}}{\|\mathbf{W}_L \mathbf{M}\|_F} - \mathbf{M}^* \right\|_F. \tag{9}$$

Note that $\mathcal{NC}2, \mathcal{NC}2\mathcal{W} \to 0 \implies \mathcal{NC}2\mathcal{M} \to 0$ but the converse does not hold.

**NC3 - Convergence to Self-Duality**  The third property of neural collapse (NC3) describes that the rows of the last-layer weight align with the column of the class means, that is,

$$\left\| \frac{\mathbf{W}_L}{\|\mathbf{W}_L\|_F} - \frac{\mathbf{M}^\top}{\|\mathbf{M}^\top\|_F} \right\|_F \to 0;$$

the corresponding metric is an obvious one Papyan et al. (2020); Garrod & Keating (2024):

$$\mathcal{NC}3 \overset{\text{def.}}{=} \frac{1}{Kp} \left\| \frac{\mathbf{W}_L}{\|\mathbf{W}_L\|_F} - \frac{\mathbf{M}^\top}{\|\mathbf{M}^\top\|_F} \right\|_F \tag{10}$$

**NC4 - Simplification of Nearest-Class-Center (NCC)**  The fourth property of neural collapse (NC4) describes that the classifier decision boundaries become equivalent to those derived by a nearest-class-mean classifier, that is,

$$\arg\max_k \langle \mathbf{w}_k, \mathbf{h} \rangle \to \arg\min_k \|\mathbf{h} - \boldsymbol{\mu}_k\|_2$$

for any test feature $\mathbf{h} \in \mathbb{R}^p$; hence we can fix a test set of features $\{\mathbf{h}_n^{\text{test}}\}_{n=1}^{N^{\text{test}}}$ define the metric:

$$\mathcal{NC}4 \overset{\text{def.}}{=} \frac{1}{N^{\text{test}}} \sum_{n=1}^{N^{\text{test}}} \mathbf{1}\{\arg\max_k \langle \mathbf{w}_k, \mathbf{h}_n^{\text{test}} \rangle = \arg\min_k \|\mathbf{h}_n^{\text{test}} - \boldsymbol{\mu}_k\|_2\} \tag{11}$$

where $\mathbf{1}$ is the indicator function.

The above NC properties hold if their corresponding metrics approach zero (except for NC4, which approach one) as the training step $t \to \infty$. A solution $\mathbf{W}_L, \mathbf{H}$ satisfying these properties is referred to as an NC solution.

To observe the interpolation between partial and full NC, we introduce a weaker property:

**NC0 - Zero Row Sum of Last-Layer Weight**  This new property describes that the rows of the last-layer weight $\mathbf{W}_L$ sums up to zero with the corresponding metric

$$\mathcal{NC}0 \overset{\text{def.}}{=} \frac{1}{p} \left\| \mathbf{W}_L^\top \mathbf{1} \right\|_2, \tag{12}$$

Note that $\mathcal{NC}2\mathcal{W} \to 0 \implies \mathcal{NC}0 \to 0$ but the converse does not hold.

The analogous property for the last-layer features, **Zero Column Sum of Last-Layer Features**, holds automatically because the columns of $\mathbf{M}$ are centered class means:

$$\sum_{k=1}^{K} \bar{\boldsymbol{\mu}}_k = \sum_{k=1}^{K} (\boldsymbol{\mu}_k - \boldsymbol{\mu}_G) = 0.$$

Thus, NC0 for the last-layer weights already represents a form of duality similar to NC3.

---

[4]In the original works, this metric was used to evaluate self-duality. However, in this paper, we decouple the NC properties to study the effects of implicit biases on each individually.

# C  ADDITIONAL RELATED WORK

## C.1  WEIGHT DECAY AND NEURAL COLLAPSE

Weight Decay has been shown to be essential for NC in prior works, like (Zhu et al., 2021; Pan & Cao, 2024; Jacot et al., 2024). However, their statements on weight decay are for (quasi-)optimal solutions in oversimplified models, which ignore the complex interaction between non-convex loss landscape and optimizers. Please see Appendix C.5 for an example.

## C.2  EMPIRICAL STUDIES ON THE EMERGENCE OF NEURAL COLLAPSE

Neural collapse has also been studied beyond the original problem setting, which assumes few balanced classes as well as noise-free labels. Notably, Wu & Papyan (2024) studied the occurrence of NC for large language models, which do not satisfy any of the original assumption. Jiang et al. (2023) studied neural collapse for a large number of classes, while Mouheb et al. (2024) studied the influence of imbalanced in medical image classification on NC.

## C.3  APPLICATIONS OF NEURAL COLLAPSE

The observation of neural collapse (NC) has inspired a growing body of follow-up work that applies NC metrics across various settings. In the context of out-of-distribution (OOD) detection, Ammar et al. (2024) propose a novel post-hoc detection method based on the geometric properties of NC, while Harun et al. (2025) show that explicitly controlling for NC1 can enhance OOD detection performance. Notably, the latter also claim that AdamW leads to NC, based on empirical results where NC3 values hover around 0.5 across different models—mirroring the misleading metrics reported in Table 3. As we demonstrate in the main text, however, this does not indicate true NC. This discrepancy underscores the need for a more precise and systematic framework for evaluating NC – one of the central contributions of this work.

In a separate line of inquiry, Liu et al. (2023) study the impact of class imbalance on NC and propose explicit feature regularization terms to induce NC under imbalanced distributions, resulting in improved model performance.

## C.4  COUPLED WEIGHT DECAY IN THE CONTEXT OF NEURAL COLLAPSE

To the best of our knowledge, no prior work has investigated the role of optimizer choice in the context of NC. When minimizing the objective in Equation (1) or Equation (13), the weight decay induced by the L2-regularization parameter $\lambda$ is coupled with the training loss. However, with the introduction of AdamW Loshchilov & Hutter (2019), decoupled weight decay has become the default in many modern optimizers. This paper aims to bridge this gap by systematically examining the impact of coupled versus decoupled weight decay on the emergence of NC.

## C.5  UNCONSTRAINED FEATURE MODEL

The unconstrained feature model (UFM) Mixon et al. (2022); Zhu et al. (2021) is a simplified theoretical framework commonly used to study neural collapse (NC). In UFM, the last layer feature is replaced by a trainable matrix $\mathbf{H} = (\mathbf{h}_n)_{n=1}^N$, referred to as the *unconstrained feature*, which mimics the role of feature extraction layers in deep neural networks (DNN). For analytical simplicity, the layer following the unconstrained feature is often assumed to be linear $\mathbf{W}$, making UFM a special case of deep linear networks (DLN):

$$\min_{\mathbf{W},\mathbf{H}} \sum_{n=1}^N \ell(\mathbf{W}\mathbf{h}_n, \mathbf{y}_n) + \frac{\lambda}{2}\|\mathbf{W}\|^2 + \frac{\lambda}{2}\|\mathbf{H}\|^2, \tag{13}$$

simplifying the minimization problem in Equation (1). In this paper, the loss $\ell$ is always assumed to be the cross-entropy (CE) loss, because it is the standard loss used in multi-classification tasks.

Zhu et al. (2021) has reported positive results on NC using UFM. Informally it holds that:

**Theorem C.1** (Theorem 3.1 and 3.2 in Zhu et al. (2021)). *Any global optimal solution of UFM is an NC solution, while all other critical points are strict saddles. As a result, for random initialization, it is almost surely that gradient descent finds an NC solution.*

Zhu et al. (2021) also experimented NC on realistic models with optimizers like SGD and Adam, concluding the universality of NC across different optimizers.

# D  EXPERIMENT

The experiments of this work, particularly regarding computing the NC metrics, were based on code in Wu & Papyan (2024), which can be found at Github repository `https://github.com/rhubarbwu/neural-collapse`, which was published under the MIT license. The implementation of VGG9 was based on Code taken from `https://github.com/jerett/PyTorch-CIFAR10`. The author granted explicit permission to use the code.

An overview of the experiments that were conducted in this work can be found in Table 4, which resulted in a total number of 36 different experimental settings of (architecture $\times$ optimizer $\times$ dataset) combinations. Each optimizer optimizer was trained using three different learning rates, six different values of momentum and six different values of weight decay, resulting in 108 training runs per optimizer and 3.888 training runs in total. Some of the runs diverged or only achieved suboptimal training performance, which were then discarded. In total we had 2.500 "valid" training runs, which reached at least 99% training accuracy, which were considered for for the subsequent data analysis.

Table 4: Overview of experiments conducted in this work.

| Architectures | Optimizers | Datasets |
|---|---|---|
| ResNet9, VGG9 | SGD, SGDW, Adam, AdamW, Signum, SignumW | MNIST, FashionMNIST, CIFAR10 |

## D.1  DETAILS ON CHOICE OF HYPERPARAMETERS

Every model was trained over 200 epochs with a batch size of 128. The learning rate $\lambda$ was chosen to be in $\lambda \in \{0.001, 0.01, 0.0679\}$ for SGD and SGDW (the last learning rate was also reported in the original work by Papyan et al. (2020)) and $\lambda \in \{0.001, 0.005, 0.01\}$ for Adam, AdamW, Signum, and SignumW because most trainings diverged with larger learning rates during initial experimental training runs. The learning rate was decayed by a factor of 10 after one third and two third of training as has been done in original work by Papyan et al. (2020). Momentum $\mu$ (or $\beta_1$ for Adam, AdamW, Signum, and SignumW) was chosen to be in the range $\mu \in \{0, 0.5, 0.7, 0.9, 0.95, 0.98\}$ for all optimizers and weight decay WD was chosen to be in the range $WD \in \{0, 5e^{-5}, 5e^{-4}, 5e^{-3}, 0.05, 0.5\}$ for SGD, SGDW, Adam, and Signum and $WD \in \{0, 5e^{-4}, 0.05, 0.5, 5, 10\}$ for SignumW and AdamW. The main motivation for using AdamW and Signum W with much larger weight decay values was based on the hypothesis that the effect of weight decay is reduced due to decoupling. The $\beta_2$ parameter in Adam and AdamW was left to its default value of 0.999.

## D.2  DETAILS ON COMPUTATIONAL RESOURCES

All experiments, including preliminary experiments as well as the final 3.888 experiments were run on 5 NVIDIA RTX4090 GPUs with 24 GB RAM. Since the models and the batch size was comparably small, actually only 3 GB GPU memory per training was required. Each training took between 8 and 16 minutes, leading to a total of 500-1000 GPU hours of training.

Table 5: Hyperparameters for each optimizer to achieve the smallest NC3 metric shown in Table 3.

| Optimizer | Learning rate | Momentum/$\beta_1$ | Weight decay |
|---|---|---|---|
| SGD | 0.01 | 0.9 | 0.05 |
| SGDW | 0.0679 | 0.5 | 0.05 |
| Adam | 0.005 | 0.98 | 0.05 |
| AdamW | 0.005 | 0.95 | 5 |
| Signum | 0.001 | 0.9 | 0.05 |
| SignumW | 0.001 | 0.98 | 10 |

Table 6: Summary of regression fit between NC3 and NC0

| Experiment | n | $\hat{\beta}$ | SE($\hat{\beta}$) | t-value | p-value | 95 % CI | $R^2$/ Adj $R^2$ | F-statistic |
|---|---|---|---|---|---|---|---|---|
| LR=0.001 | 170 | 0.1903 | 0.008 | 24.262 | 0.000 | [0.175, 0.206] | 0.778 / 0.777 | 588.6 |
| LR=0.005 | 74 | 0.2017 | 0.012 | 16.252 | 0.000 | [0.177, 0.226] | 0.786 / 0.783 | 264.1 |
| LR=0.01 | 114 | 0.1439 | 0.007 | 19.892 | 0.000 | [0.13, 0.158] | 0.779 / 0.777 | 395.7 |
| LR=0.0679 | 41 | 0.1771 | 0.012 | 14.367 | 0.000 | [0.152, 0.202] | 0.841 / 0.837 | 206.4 |
| all | 399 | 0.1582 | 0.005 | 32.760 | 0.000 | [0.149, 0.168] | 0.730 / 0.729 | 1073 |

## D.3 DETAILS ON REGRESSION FIT BETWEEN NC3 AND NC0

In this subsection we provide additional details regarding the regression fit between NC3 and NC0. For the sake of completeness, we show the regression fit in Figure 10 again below. In addition, we have also computed a regression fit across all training runs, which converged, and all learning rates, shown in Figure 11. A summary of the regression fit can be found in Table 6, showing that more than 70% of the variation in NC3 can be explained by NC0.

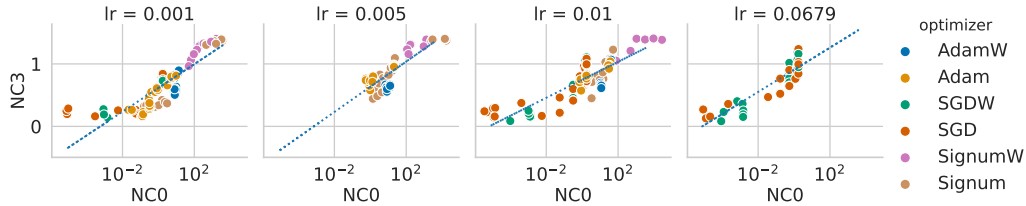

Figure 10: Figure 2 shown again for ease of reading. NC0 weakly correlates with NC3 across different optimizers and learning rates (here shown for ResNet9 trained on FashionMNIST).

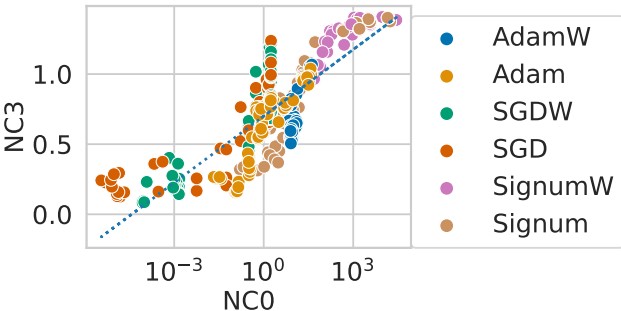

Figure 11: NC0 correlates with NC3 even when considered across all learning rates together (here shown for ResNet9 trained on FashionMNIST).

## D.4 ADDITIONAL EXPERIMENTAL RESULTS

### D.4.1 ABLATION STUDY ON TRAINING EPOCHS

As Neural collapse occurs at the terminal phase of training, it is natural to control for the effect that the number of training epochs has on the final NC metrics. After all, it is possible that the emergence of NC occurs at different speeds for different optimizers.

For this reason, we conducted two ablation studies, in which we prolong the training in two settings: We train a ResNet9 in FashionMNIST, which corresponds to the setting which is shown in Figure 1, for 2000 epochs with LR=0.0005 and momentum=0.9 for both optimizers. We note that in this setting, AdamW reaches 100% training accuracy already after around 700 epochs for all training runs with WD $\leq 0.05$. The results can be found in Figure 14 While this leads to some improvement of the final

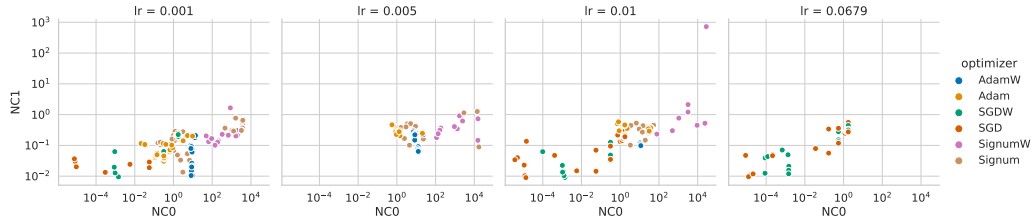

Figure 12: NC0 vs. NC1 across different optimizers and learning rates (here shown for ResNet9 trained on FashionMNIST).

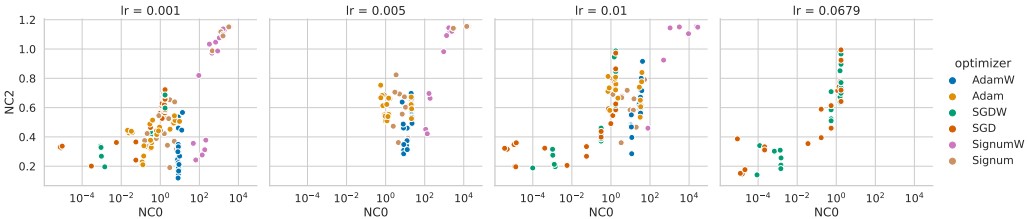

Figure 13: NC0 vs. NC2 across different optimizers and learning rates (here shown for ResNet9 trained on FashionMNIST).

NC1 and NC2 metric for AdamW for some values of weight decay, this has barely an effect on NC0 and NC3.

Furthermore we extend training to up to 2000 epochs for selected runs from Figure 5. Concretely, these runs trained with a LR of 0.001 and the following combination of WD and momentum (mom, WD) $\in \{(0,0), (0.97, 5e^{-5}), (0, 5e^{-4}), (0.9, 5e^{-4}), (0.9, 0), (0.95, 0.0025)\}$, which corresponds to different parts in the heatmap. The results can be found in Figure 15. While one can observe a general decrease of the NC metrics in all cases, the overall trend for increasing weight decay remains unchanged. Both figures indicate that training the models considered in this work for 200 epochs is sufficient to draw the conclusions that we make about the necessity of coupled WD for the emergence of full NC.

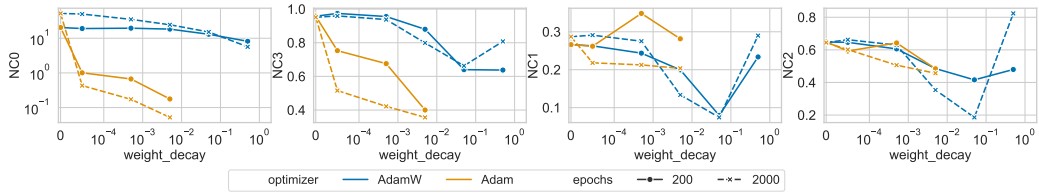

Figure 14: ResNet9 trained on FashionMNIST with Adam and AdamW for more epochs.

### D.4.2 UNCONSTRAINED FEATURE MODEL

We also validated our results on the unconstrained feature model (UFM) (see Appendix C.5 for reference) with width $d = 512$, $K = 10$ classes and $N = 10.000$ samples. The UFM was trained with Adam, AdamW and SGDMW with momentum=0.9 and varying lr$\in \{0.1, 0.3, 0.5, 1.0\}$ and weight decay ranging from $0.0$ to $0.05$. We then filtered the results, by only including models which achieved 100% training accuracy. The results in can be found in Figure 16. The plots show that the NC metrics, in particular NC0 and NC3 remain at least one magnitude of order larger than the same metrics for Adam and SGDMW, highlighting that AdamW converges to a different solution than Adam, which is not NC.

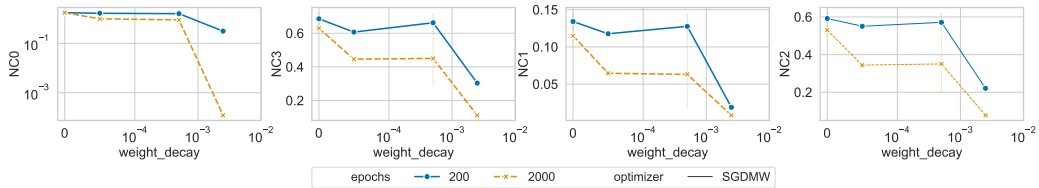

Figure 15: Selected runs from Figure 5 trained for more number of epochs.

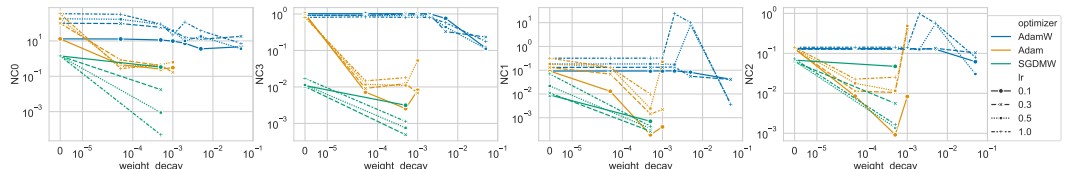

Figure 16: NC0 (left), NC3 (center left), NC1 (center right), and NC2 (right) for increasing weight decay.

### D.4.3  TRAINING DYNAMICS OF MINIMAL NC3 RUNS

In this section we provide the dynamics of the NC metrics as well as the singular values from the training runs which reached the smallest final NC3 metric as reported in Table 5. The purpose is to disentangle the effect of using first-order optimizers (such as SGD and SGDW) vs. second-order like optimizers (such as Adam and AdamW) from the effect of applying coupled vs. decoupled weight decay. The main question we try to answer here is: Is the difference between Adam and SGD with respect to the emergence of NC larger than the difference between AdamW and Adam? Figure 17 shows that all runs reach a perfect train accuracy well before the end of training, such that they have reached the terminal-phase of training (TPT) at epoch 200. Looking at the NC1-NC3 metrics in Figure 18 and Figure 20, one can see that the NC metrics for SGD and SGDW are close to each other. It is harder to judge whether AdamW or Adam are closer to NC, as NC3 is considerably larger for AdamW, while NC1 is slightly larger for Adam, compared to the other optimizers. Nonetheless, the NC0 metric in Figure 18 and the evolution of the singular values of $\mathbf{W}$ in Figure 19 (left) indicate that AdamW has considerably different training dynamics than Adam, as both NC0 as well as the smallest singular value increase instead of converging to zero for AdamW, but not for Adam. While the NC0 metric of Adam is still orders of magnitude larger than for SGD and SGDW and the smallest singular value of $\mathbf{W}$ converges to a small, but non-zero value, Adam shares similar trends as SGD and SGDW and as such converges to a solution which is arguably closer to NC3 than AdamW. Whether the solution found by Adam can already be classified as NC or not is an inherent problem of interpreting the NC metrics in practical settings, as we have also discussed in Section 4.1.

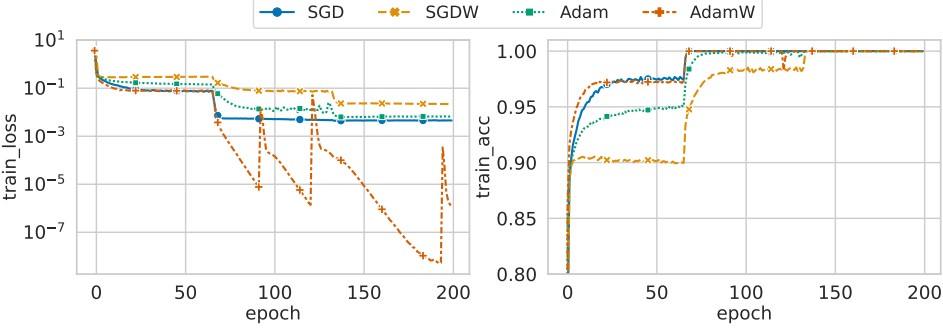

Figure 17: Train loss (left) and train accuracy (right) for training runs with smallest final NC3 metric for different optimizers.

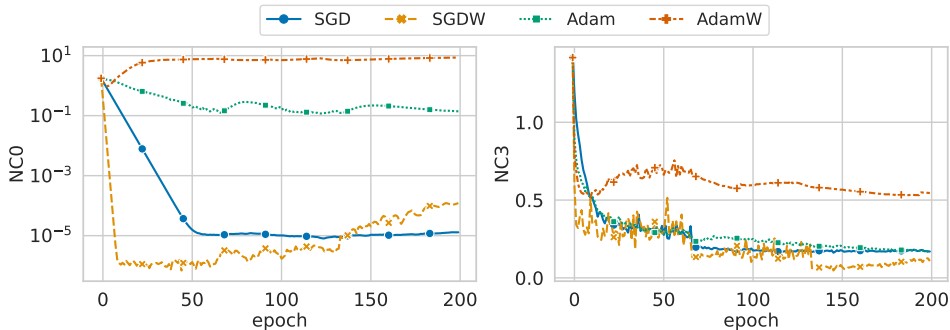

Figure 18: NC0 (left) and NC3 (right) for training runs with smallest final NC3 metric for different optimizers.

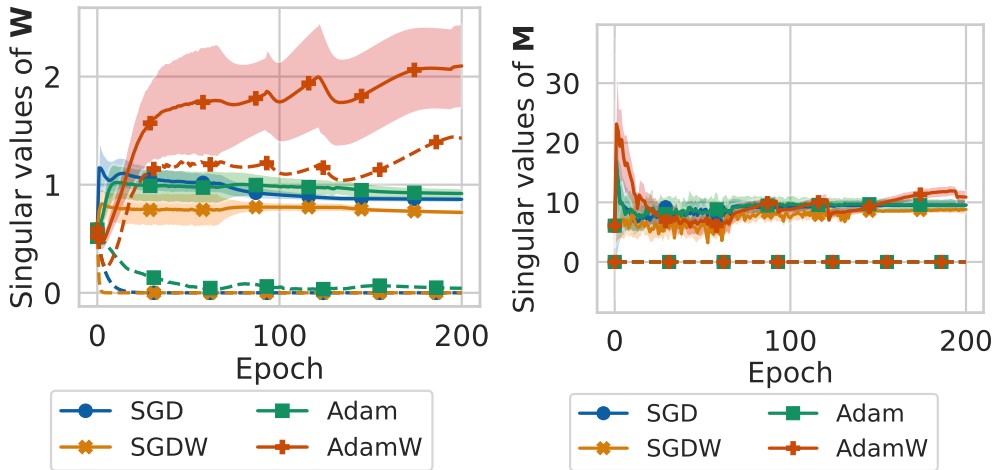

Figure 19: Singular values of last-layer weights **W** (left) and centered class means **M** (right) throughout training for runs corresponding to Table 5. The dotted line corresponds to the smallest singular value and the full line corresponds to the average singular value, excluding the smallest singular value.

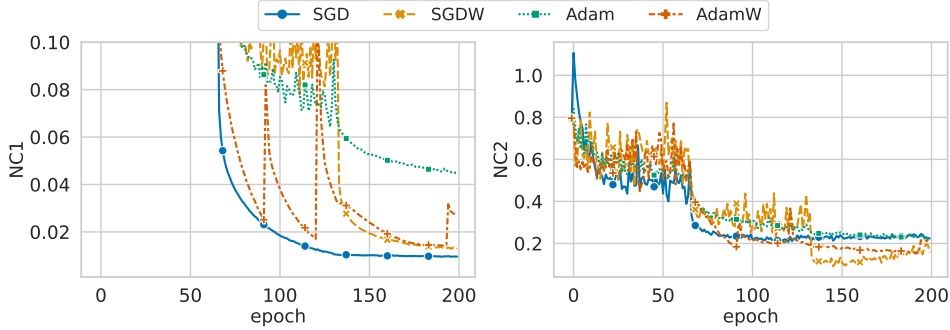

Figure 20: NC1 (left) and NC2 (right) for training runs with smallest final NC3 metric for different optimizers.

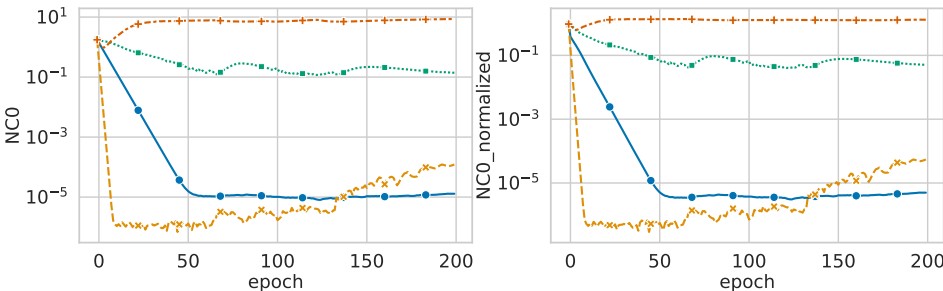

Figure 21: NC0 (left) and normalized NC0 (right) for training runs with smallest final NC3 metric for different optimizers.

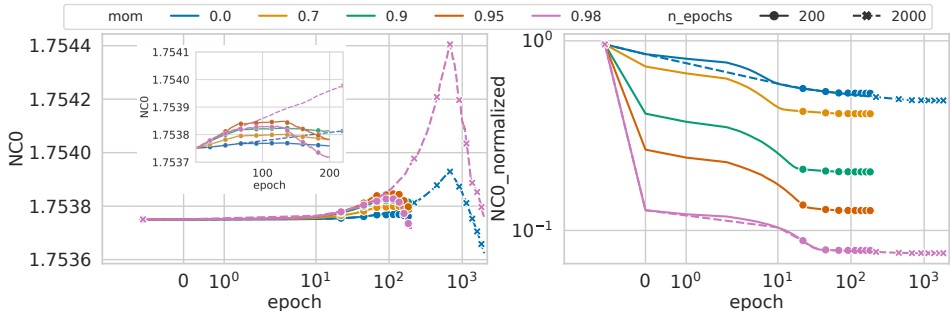

Figure 22: NC0 (left) and normalized NC0 (right) for training runs with zero weight decay from the ablation study in Appendix D.4.6. Note that the x-axis is in logarithmic scale and that the point at epoch -1 corresponds to the model at initialization.

### D.4.4 ABLATION STUDY ON NORMALIZING THE NC0 METRIC

We evaluate whether measuring a normalized NC0 metric affects the conclusions that we draw in our work. Concretely, we compute the normalization as

$$\text{NC0}_{\text{normalized}} := \frac{1}{p}\|\mathbf{W}_L^\top \mathbf{1}\|_2 / \|\mathbf{W}_L\|_F. \tag{14}$$

We compute both NC0 as well as normalized NC0 for the setting of minimal NC3 that we studied in Appendix D.4.3, which we show in Figure 21. While the absolute values differ slightly between NC0 and NC0$_{\text{normalized}}$, both the trends as well as the final values are almost the same.

For zero weight decay, one would expect to see more difference between the dynamics of NC0 and normalized NC0, which we show in Figure 22. While one can observe the monotontic effect of momentum on normalized NC0, but not on NC0, we point out that in this case normalized NC0 does not correlate with NC1-NC3 anymore. On the contrary, NC1-NC3, while still comparably large, are smaller with less momentum.

As the dynamics of NC0 and normalized NC0 are almost the same for larger values of WD or normalized NC0 is not consistent with NC1-NC3 for zero WD, we are tentative to conclude that the normalization will not affect the conclusions that we draw in this work.

### D.4.5 ABLATION STUDY ON EFFECT OF MOMENTUM ON NC EMERGENCE

We conduct another ablation study to further evaluate the effect of momentum on the NC emergence. The main question that we try to answer with this ablation is whether the effect of momentum on smaller NC metrics can be simply traced back to the fact that momentum accelerates convergence or if it affects the emergence of NC beyond this.

Concretely, we track the evolution of the NC metrics together with the train loss and accuracy for the same setting as in Figure 5 over time for a fixed value fo weight decay=0.005 and varying values of momentum. This is because the final train loss value varies for different values of WD due to

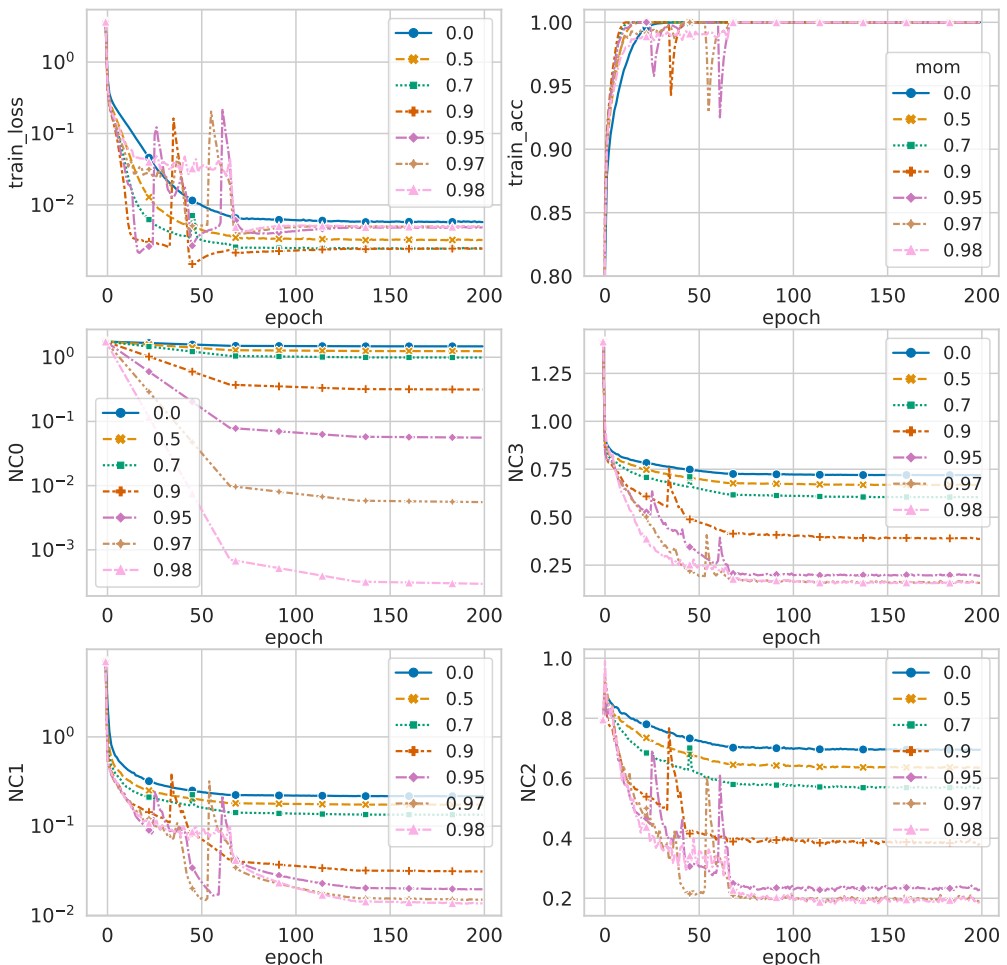

Figure 23: Train loss, train accuracy and NC metrics for fixed WD=0.005 and different values of momentum on a ResNet9 trained with SGD with otherwise same hyperparameters as in Figure 5.

its regularizing effect. The results can be seen in Figure 23. There are two things to be observed: While the accelerating effect of momentum is mainly visible in the early phase of training (up to 50-100 epochs), modulo some loss spikes for high momentum, the final train loss is not smallest for the largest value of momentum. While this is not surprising per se, as too large momentum can lead to a overshooting of the training trajectory, the NC metrics show a clear monotonic behavior with respect to the momentum. Furthermore, while the training runs with momentum=0.7 and 0.9 reach almost the exact same final train loss, the disparity in NC metrics indicates that they converged to solutions with very different geometric structure. This can be seen more clearly in Figure 24. Both observations suggest that momentum affects the emergence of NC beyond simply accelerating the speed of convergence. To the best of our knowledge, connecting the magnitude of momentum to NC is novel and not been discussed in prior work.

### D.4.6 ABLATION STUDY ON NC EMERGENCE UNDER ZERO WEIGHT DECAY

To investigate whether WD is necessary or not for the emergence of NC, we track the NC metrics while training a ResNet9 on FashionMNIST (Note that this is the same problem setting as in Appendix D.4.5.) using SGD with zero WD and varying values of momentum with an initial LR=0.01 for 200 epochs. Additionally, we train the model also with zero momentum and high momentum=0.98 for 2000 epochs, with LR decay after 1/3 and 2/3 of training. Importantly, all training runs reach perfect train accuracy after 40 epochs. The training dynamics can be found in Figure 25. We draw two conclusions from this ablation study:

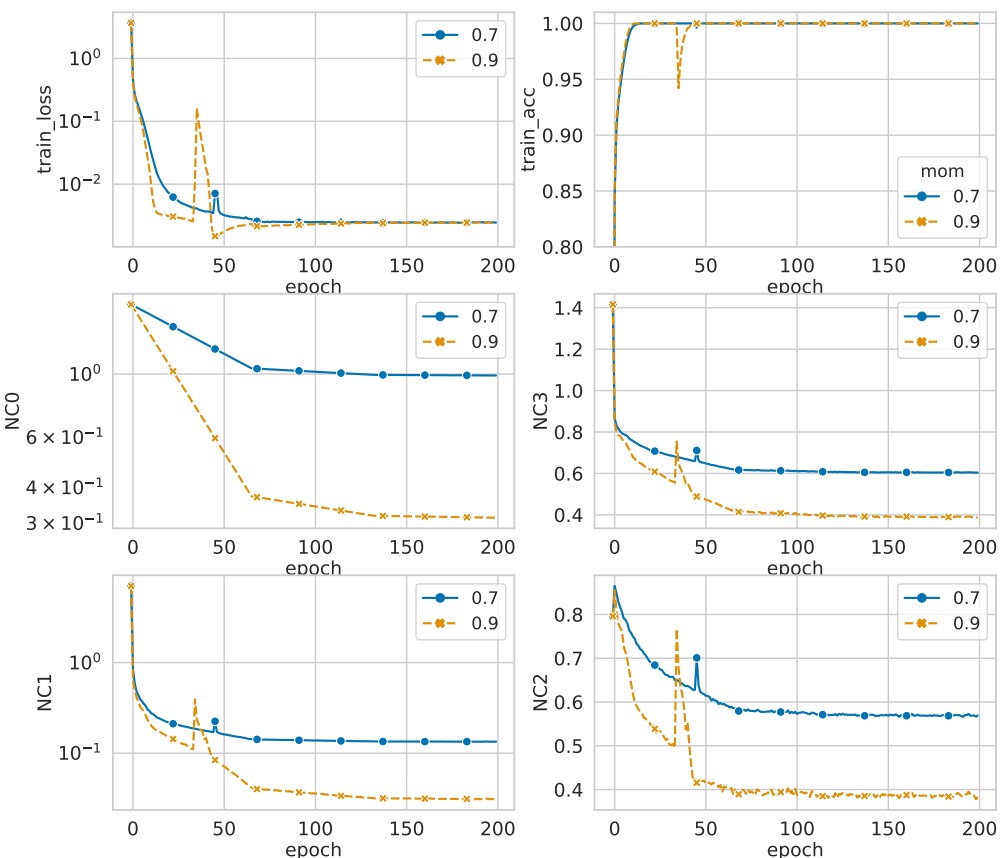

Figure 24: Train loss, train accuracy and NC metrics for fixed WD=0.005 and mom=0.7 and 0.9. Although both runs converge to almost exactly the same train loss, the final NC metrics differ considerably.

Table 7: Smallest NC metrics achieved with and without weight decay for training a ResNet9 on FashionMNIST.

| SGD with ... | NC1 | NC2 | NC3 |
|---|---|---|---|
| no WD (2000 epochs) | $\approx 0.2$ | $\approx 0.55$ | $\approx 0.7$ |
| WD (200 epochs) | $\approx 0.02$ | $\approx 0.2$ | $\approx 0.13$ |

1. The final NC metrics NC0-NC3 after 2000 epochs are slightly smaller than after 200 epochs, consistent with our ablation study in D.4.1. that longer training reduces the NC metrics. This decrease is however fairly small.

2. The final NC metrics (both for 200 epochs and 2000 epochs of training) remain considerably higher than what is achieved by the "best" run of SGD in terms of NC metrics with 200 epochs of training for all NC metrics, even with 10 times longer training. See Figure 18 and Figure 20 for a comparison.

The final NC1-NC3 metrics achieved with WD after 200 epochs and without WD after 2000 epochs can be found in Table 7. While the experiments cannot fully exclude the possibility that NC can be achieved eventually in the asymptotic limit, we argue that WD is essential to observe the emergence of NC *in practical finite-length training settings*.

### D.4.7 MORE DETAILED PLOTS ON COUPLED VS. DECOUPLED WEIGHT DECAY

As we average across different values of momentum and learning rates in Figure 1 and Figure 3, we provide more detailed plots here in Figure 26, Figure 27, and Figure 28. It can be seen that for the adaptive optimizers and SGDW the variance for varying values of momentum is comparably small for each fixed learning rate, with the variance generally increasing with larger weight decay. For SGD the variance for NC0 is higher for large values of weight decay, consistent with what is shown in Figure 3 (right) and what is shown in Figure 5.

### D.4.8 MISSING PLOT: SINGULAR VALUE OF $\mathbf{W}$ AND $\mathbf{M}$ WITH SIGNUMW

The missing plot of the evolution of the singular values of the last-layer weights $\mathbf{W}$ and feature matrix $\mathbf{M}$ can be found in Figure 29.

### D.4.9 COUPLED VS. DECOUPLED DECAY ON OTHER DATASETS

The comparison between coupled and decoupled decay on SGD, Adam, and Signum on other combinations of (architecture $\times$ dataset) can be found in the following pages below, which confirm our observations made earlier on the ResNet9 trained on FashionMNIST. While NC0 (visually) correlates well with NC3, it correlates considerably less with NC1 and NC2, although a general trend is still visible across all experiments.

**ResNet50 on ImageNet1K** We also conducted experiments on a ResNet50 trained on ImageNet1K Deng et al. (2009). The model was trained with Adam and AdamW for 90 epochs. We left out other optimizers due to limited resources. For both optimizers the learning rate was chosen as 0.0003 with a step-wise decay after 1/3 and 2/3 of training, momentum was chosen from $\{0.0, 0.5, 0.9\}$ and weight decay was chosen from $\{0.0, 1e^{-5}, 1e^{-4}, 1e^{-3}\}$. The resulting NC metrics can be found in Figure 30 and Figure 31, and confirm the conclusion that AdamW does not have full NC emergence.

**VGG9 on FashionMNIST** The comparison between coupled and decoupled weight decay on SGD, Adam, and Signum on a VGG9 trained on FashionMNIST can be found in Figure 32 and Figure 33. The relation between NC0 and NC3 can be found in Figure 36, between NC0 and NC1 in Figure 34, and between NC0 and NC2 in Figure 35.

**ResNet9 on Cifar10** The comparison between coupled and decoupled weight decay on SGD, Adam, and Signum on a ResNet9 trained on Cifar10 can be found in Figure 37 and Figure 38. The relation between NC0 and NC3 can be found in Figure 41, between NC0 and NC1 in Figure 39, and between NC0 and NC2 in Figure 40.

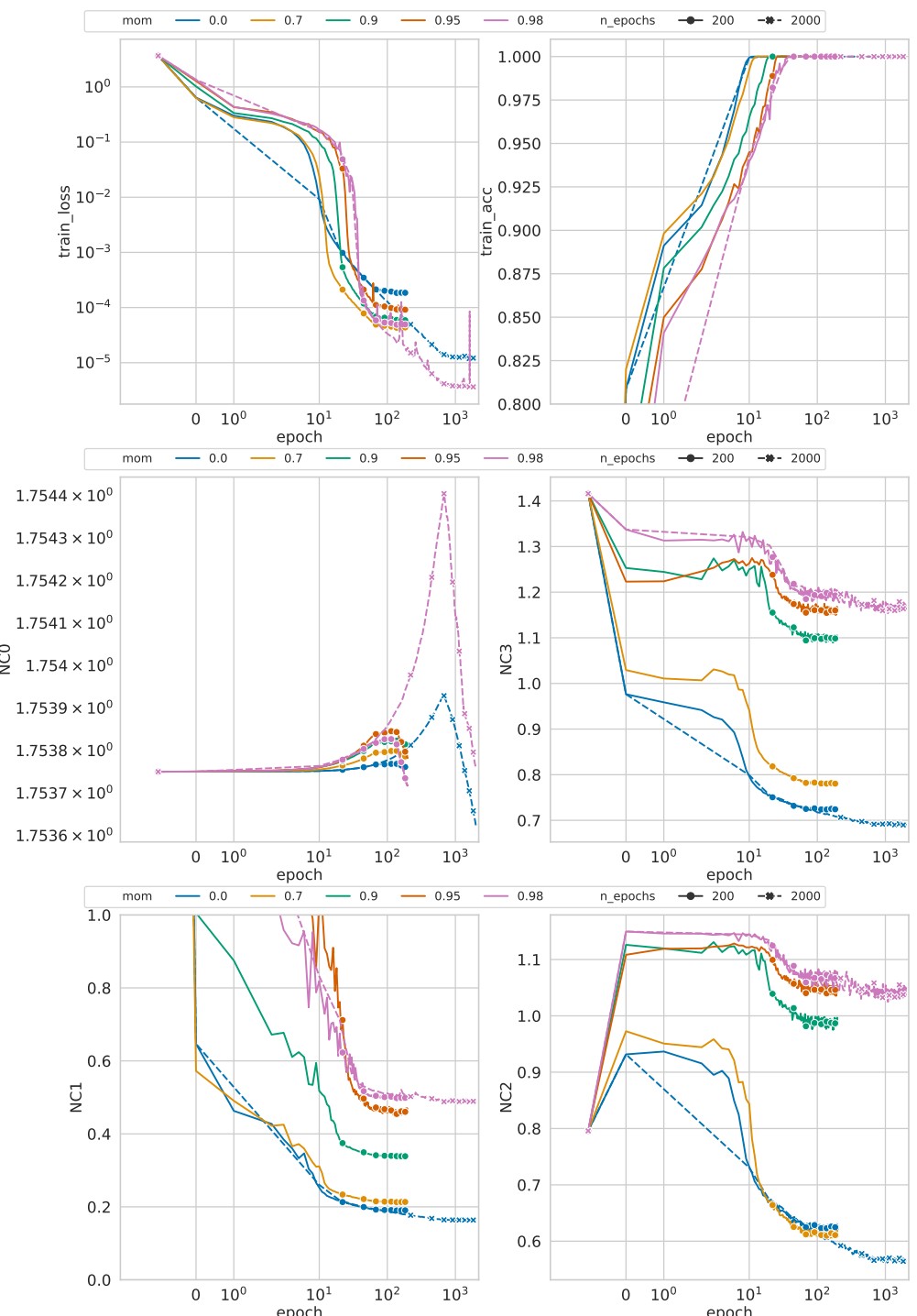

Figure 25: Training loss and train accuracy (top row), NC0 and NC3 (middle row), and NC1 and NC2 (bottom row) for a ResNet9 trained on FashionMNIST with SGD without weight decay for varying values of momentum and number of epochs. Note that the x-axis is in log-scale to improve readability.

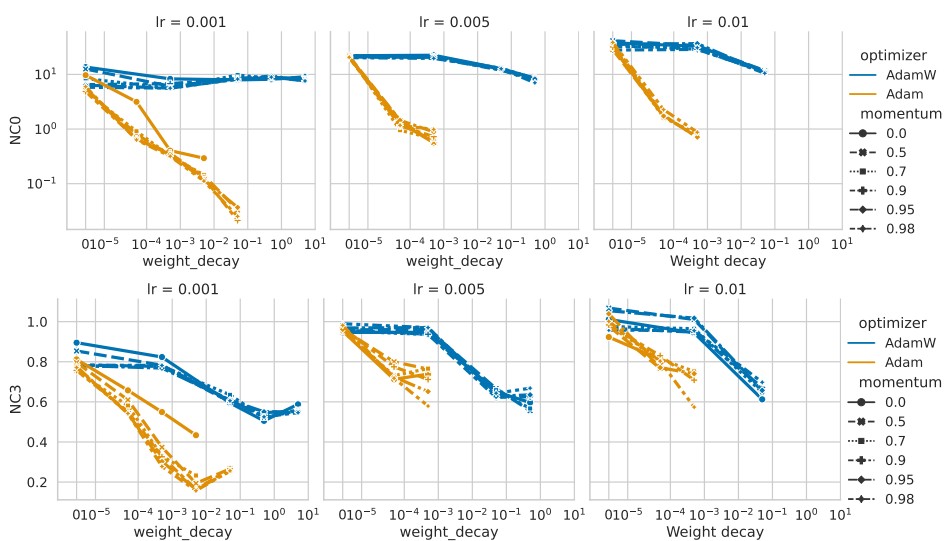

Figure 26: NC0 metric (top) and NC3 metric (bottom for different values of weight decay, momentum and LR for Adam vs. AdamW.

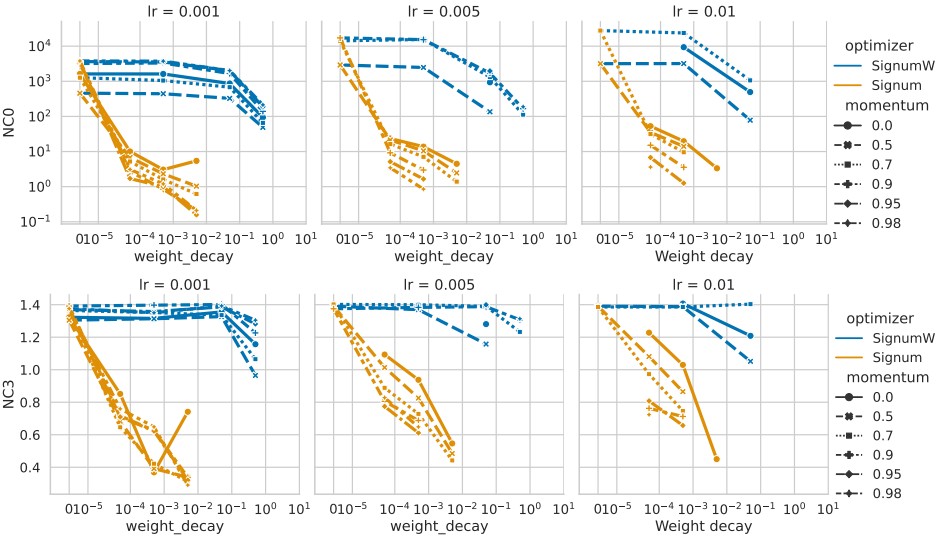

Figure 27: NC0 metric (top) and NC3 metric (bottom for different values of weight decay, momentum and LR for Signum vs. SignumW.

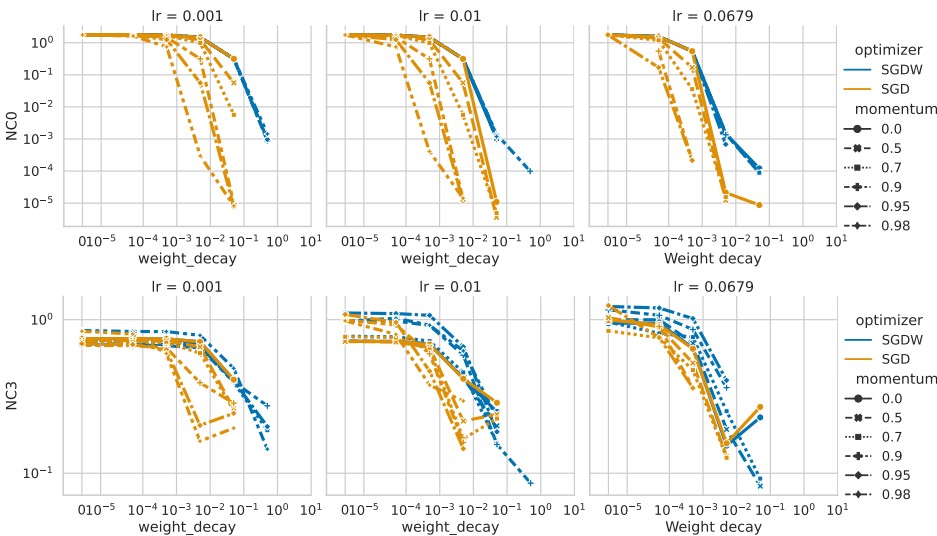

Figure 28: NC0 metric (top) and NC3 metric (bottom for different values of weight decay, momentum and LR for SGD vs. SGDW.

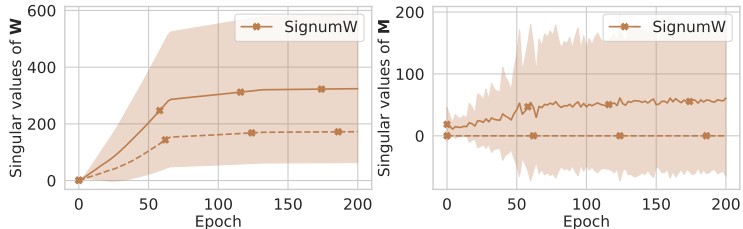

Figure 29: Singular values of last-layer weights **W** (left) and feature matrix **M** (right) throughout training for SignumW on ResNet9 trained on FashionMNIST. Dotted line corresponds do smallest singular value and full line corresponds to the average singular value excluding the smallest singular value.

**VGG9 on Cifar10**    The comparison between coupled and decoupled weight decay on SGD, Adam, and Signum can be found in Figure 42 and Figure 43. The relation between NC0 and NC3 can be found in Figure 46, between NC0 and NC1 in Figure 44, and between NC0 and NC2 in Figure 45.

**ResNet9 on MNIST**    The comparison between coupled and decoupled weight decay on SGD, Adam, and Signum on a ResNet9 trained on MNIST can be found in Figure 47 and Figure 48. The relation between NC0 and NC3 can be found in Figure 51, between NC0 and NC1 in Figure 49, and between NC0 and NC2 in Figure 50.

**VGG9 on MNIST**    The comparison between coupled and decoupled weight decay on SGD, Adam, and Signum on a VGG9 trained on MNIST can be found in Figure 52 and Figure 53. The relation between NC0 and NC3 can be found in Figure 56, between NC0 and NC1 in Figure 54, and between NC0 and NC2 in Figure 55.

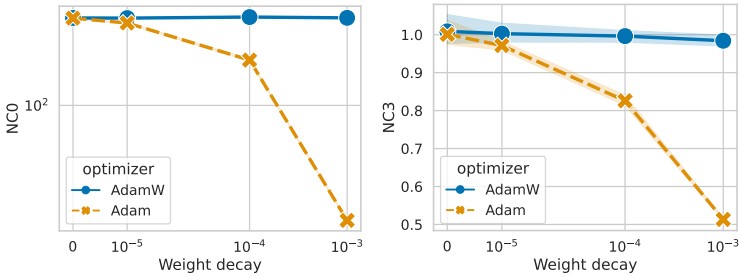

Figure 30: NC0 (left) and NC3 (right) metrics plotted against weight decay on a ResNet50 trained on ImageNet1K for Adam and AdamW. Shaded area refers to one standard deviation across all trainings run with corresponding optimizer.

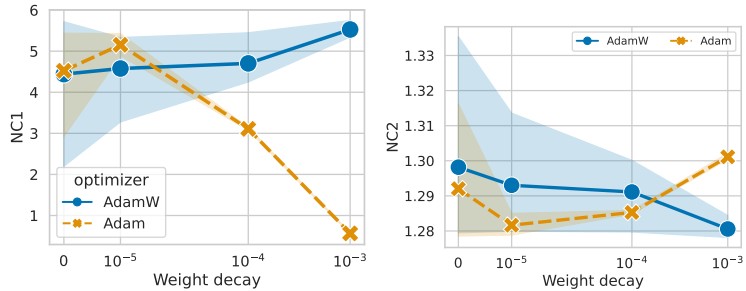

Figure 31: NC1 (left) and NC2 (right) metrics plotted against weight decay on a ResNet50 trained on ImageNet1K for Adam and AdamW. Shaded area refers to one standard deviation across all trainings run with corresponding optimizer.

### D.4.10 PRELIMINARY EXPERIMENTAL RESULTS ON VISION TRANSFORMER

We have also conducted preliminary experiments pretraining small Vision Transformers (ViT) on Cifar10 from scratch. Given that training ViTs is computationally much more expensive given the larger size of the model, we had to limit ourselves to a more restricted number of experiments. Specifically, we chose to train the ViT with Adam, AdamW, and SGD for 200 epochs with a batch size of 512 with momentum $\mu$ in the range $\mu \in \{0, 0.8, 0.9, 0.95\}$ and weight decay WD $\in \{0, 1e^{-5}, 1e^{-4}, 5e^{-4}, 1e^{-3}, 0.05, 0.5\}$ for Adam and SGD and WD $\in \{0, 1e^{-4}, 0.05, 0.5, 1, 2, 4\}$ for AdamW. We discarded all runs, which did not achieve a training accuracy of at least 50%. This mainly corresponded to training runs of SGD and Adam either with momentum=0 or WD$\geq 0.05$.

The ViT implementation is based on code from `https://github.com/tintn/ vision-transformer-from-scratch/tree/main`, which is published under the MIT license. Specifically, the transformer model was chosen with a hidden dimension of 512, 6 hidden layers, and 8 attention heads, with no dropout applied.

Compared to the training procedure used in other settings, we employ a cosine-decay learning rate schedule with warm-up, where 5% of the total training steps are allocated to warm-up, and the base learning rate is set to $1 \times 10^{-3}$. Weight decay is applied to all layers except for LayerNorm and biases, which is standard practice.

The highest final test accuracy across all trainings was achieved by AdamW ($\beta_1 = 0.95, \text{WD} = 0.5$) with 83.67%, with a final test loss of 0.895. Notably, higher accuracy levels can be attained by increasing the network size and applying data augmentation or by using a pre-trained model as in Ammar et al. (2024). However, to ensure consistency with the experiments in the main study, we do not perform data augmentation due to limited computational resources. This likely explains the relatively lower test accuracy. Investigating the impact of data augmentation on the convergence to NC remains an interesting avenue for future work.

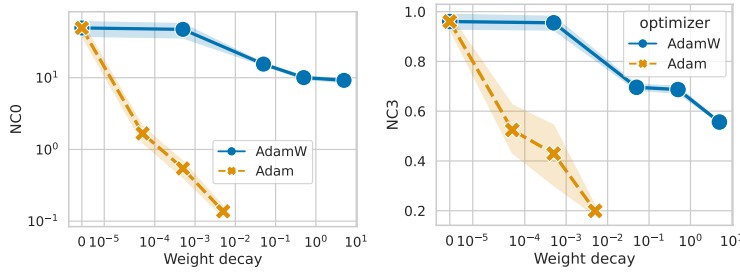

Figure 32: NC0 (left) and NC3 (right) metrics plotted against weight decay on a VGG9 trained on FashionMNIST for Adam and AdamW. Shaded area refers to one standard deviation across all trainings run with corresponding optimizer.

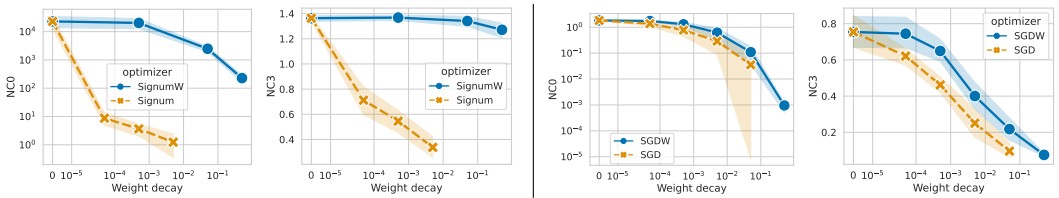

Figure 33: NC0 and NC3 metrics plotted against weight decay on a VGG9 trained on FashionMNIST for Signum and SignumW (left side) and SGD and SGDW (right side). Shaded area refers to one standard deviation across all trainings run with corresponding optimizer.

While we observe the general trend of decreasing NC metrics with increasing values of weight decay for SGD (Figure 58a), we note that in the case of ViTs the NC0 metric for both Adam and AdamW first increases before decreasing (Figure 58b, left), while the NC3 metric for both Adam and AdamW has a U-shape (Figure 58b, right). We also note that the ViT is much more sensitive to the choice of weight decay and the training and validation accuracy degrades quickly due to overregularization, as can be seen in Figure 58c. A further investigation of these observations is left for future work.

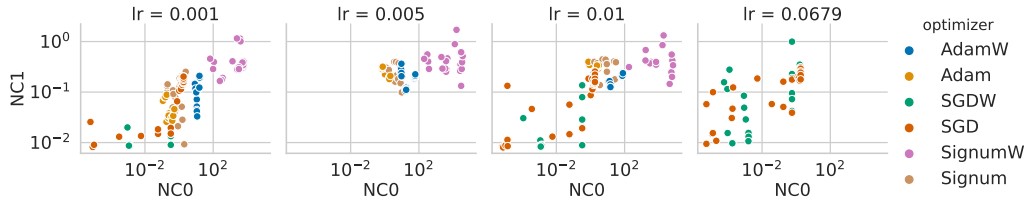

Figure 34: NC0 vs. NC1 on VGG9 trained on FashionMNIST. Note that the x-axis is plotted in log-scale.

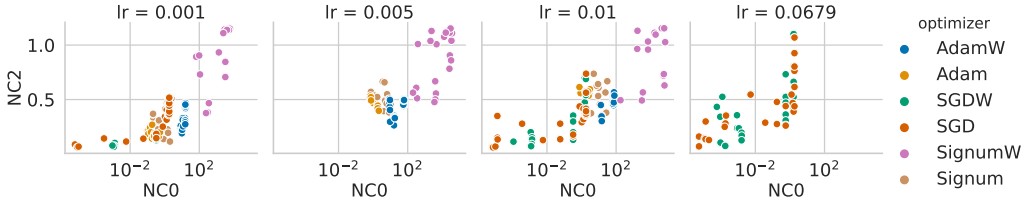

Figure 35: NC0 vs. NC2 on VGG9 trained on FashionMNIST. Note that the x-axis is plotted in log-scale.

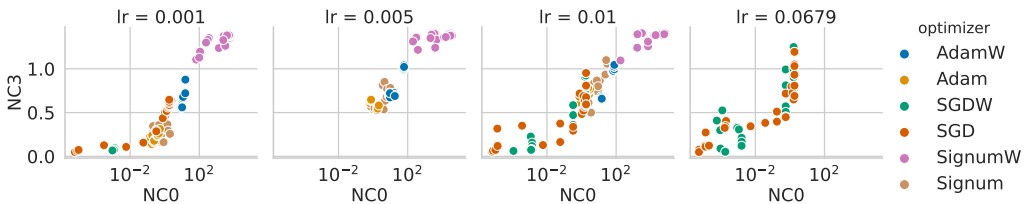

Figure 36: NC0 vs. NC3 on VGG9 trained on FashionMNIST. Note that the x-axis is plotted in log-scale.

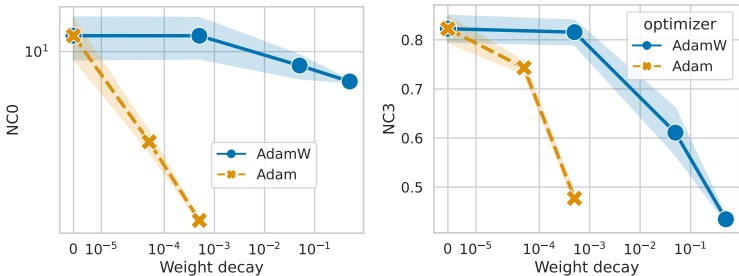

Figure 37: NC0 (left) and NC3 (right) metrics plotted against weight decay on a ResNet9 trained on Cifar10 for Adam and AdamW. Shaded area refers to one standard deviation across all trainings run with corresponding optimizer.

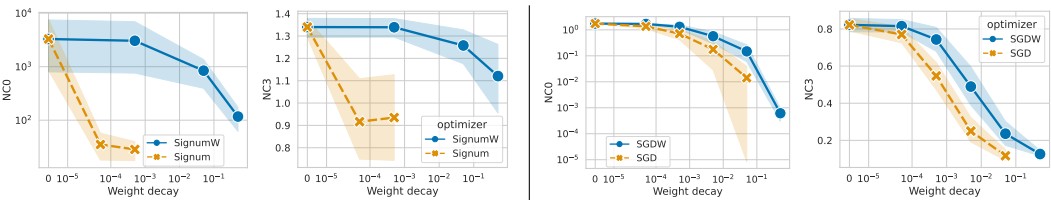

Figure 38: NC0 and NC3 metrics plotted against weight decay on a ResNet9 trained on Cifar10 for Signum and SignumW (left side) and SGD and SGDW (right side). Shaded area refers to one standard deviation across all trainings run with corresponding optimizer.

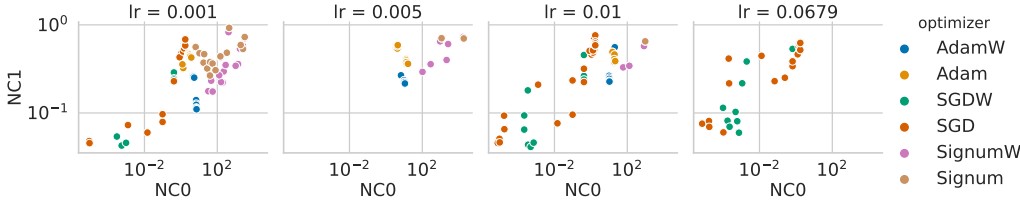

Figure 39: NC0 vs. NC1 on ResNet9 trained on Cifar10. Note that the x-axis is plotted in log-scale.

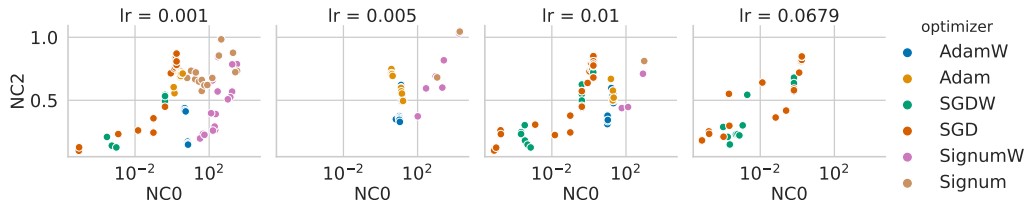

Figure 40: NC0 vs. NC2 on ResNet9 trained on Cifar10. Note that the x-axis is plotted in log-scale.

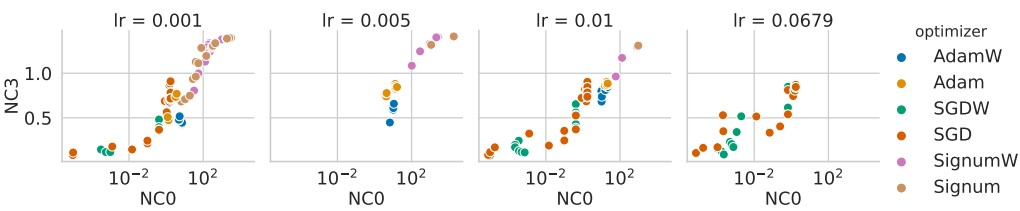

Figure 41: NC0 vs. NC3 on ResNet9 trained on Cifar10. Note that the x-axis is plotted in log-scale.

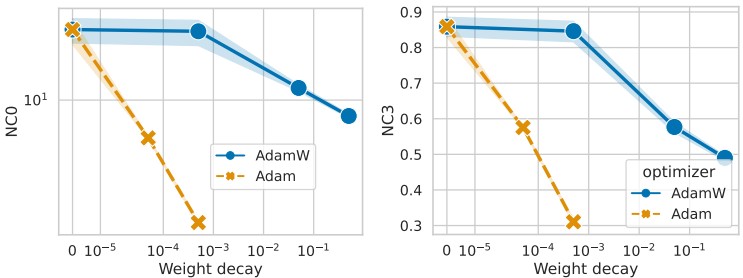

Figure 42: NC0 (left) and NC3 (right) metrics plotted against weight decay on a VGG9 trained on Cifar10 for Adam and AdamW. Shaded area refers to one standard deviation across all trainings run with corresponding optimizer.

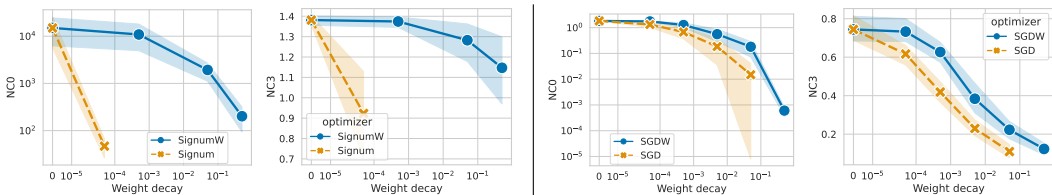

Figure 43: NC0 and NC3 metrics plotted against weight decay on a VGG9 trained on Cifar10 for Signum and SignumW (left side) and SGD and SGDW (right side). Shaded area refers to one standard deviation across all trainings run with corresponding optimizer.

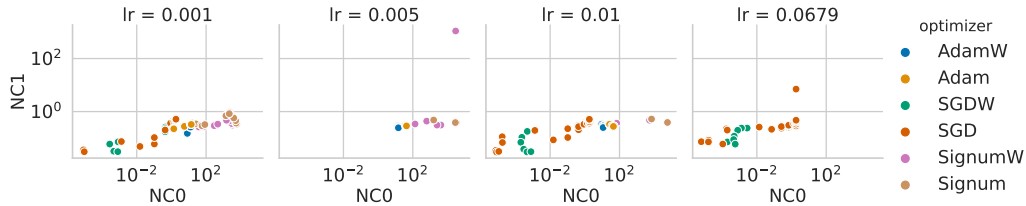

Figure 44: NC0 vs. NC1 on VGG9 trained on Cifar10. Note that the x-axis is plotted in log-scale.

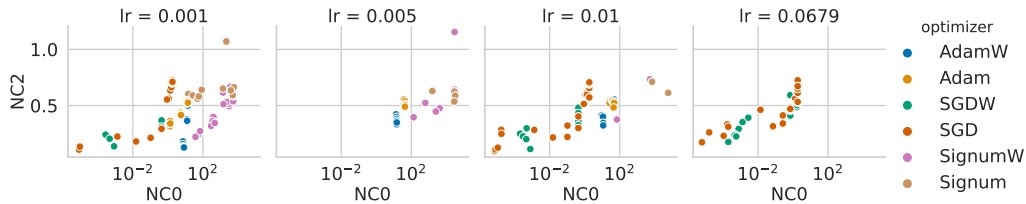

Figure 45: NC0 vs. NC2 on VGG9 trained on Cifar10. Note that the x-axis is plotted in log-scale.

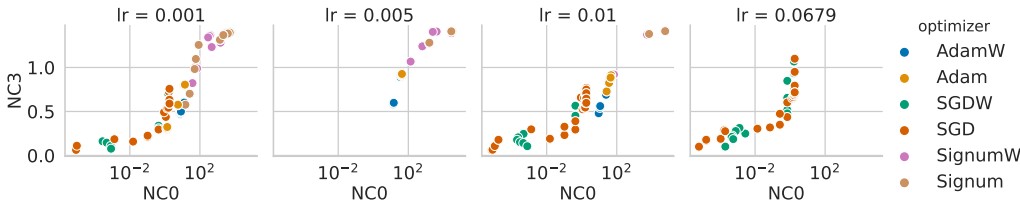

Figure 46: NC0 vs. NC3 on VGG9 trained on Cifar10. Note that the x-axis is plotted in log-scale.

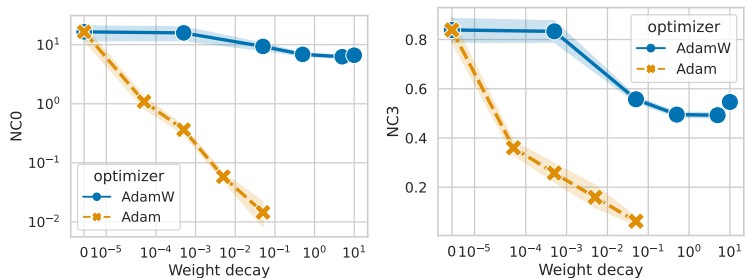

Figure 47: NC0 (left) and NC3 (right) metrics plotted against weight decay on a ResNet9 trained on MNIST for Adam and AdamW. Shaded area refers to one standard deviation across all trainings run with corresponding optimizer.

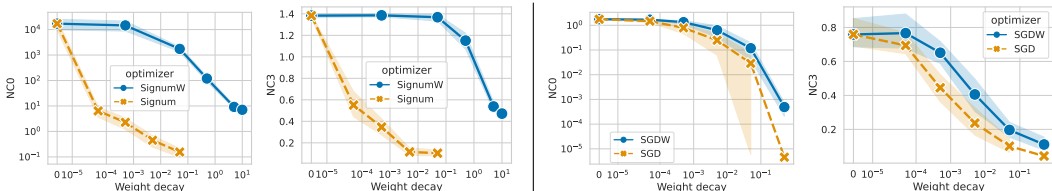

Figure 48: NC0 and NC3 metrics plotted against weight decay on a ResNet9 trained on MNIST for Signum and SignumW (left side) and SGD and SGDW (right side). Shaded area refers to one standard deviation across all trainings run with corresponding optimizer.

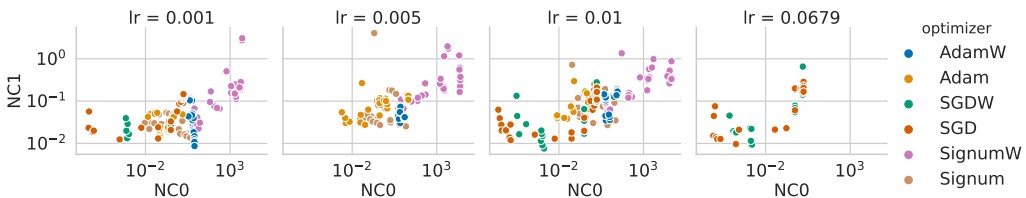

Figure 49: NC0 vs. NC1 on ResNet9 trained on MNIST. Note that the x-axis is plotted in log-scale.

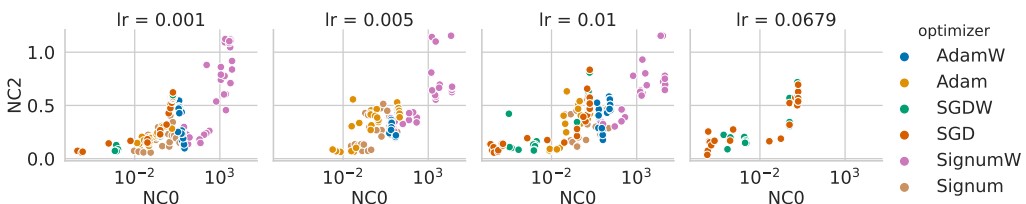

Figure 50: NC0 vs. NC2 on ResNet9 trained on MNIST. Note that the x-axis is plotted in log-scale.

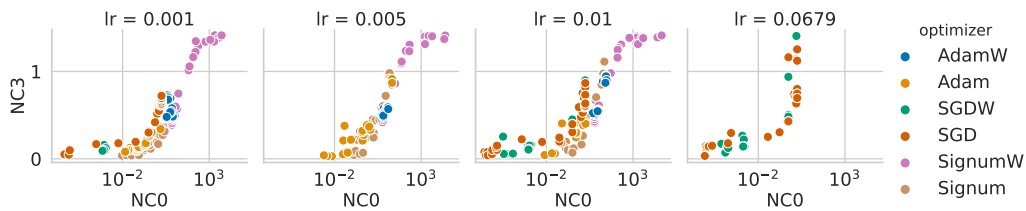

Figure 51: NC0 vs. NC3 on ResNet9 trained on MNIST. Note that the x-axis is plotted in log-scale.

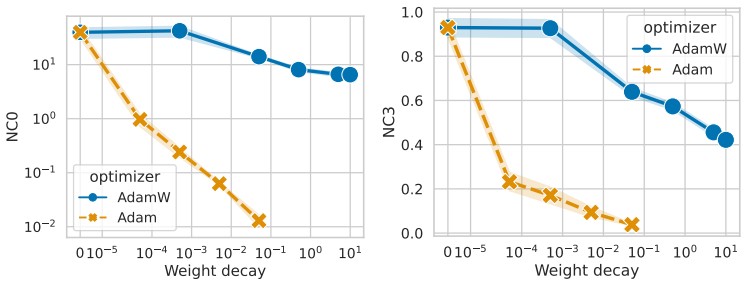

Figure 52: NC0 (left) and NC3 (right) metrics plotted against weight decay on a VGG9 trained on MNIST for Adam and AdamW. Shaded area refers to one standard deviation across all trainings run with corresponding optimizer.

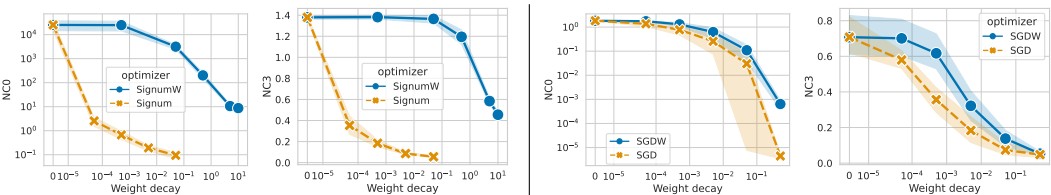

Figure 53: NC0 and NC3 metrics plotted against weight decay on a VGG9 trained on MNIST for Signum and SignumW (left side) and SGD and SGDW (right side). Shaded area refers to one standard deviation across all trainings run with corresponding optimizer.

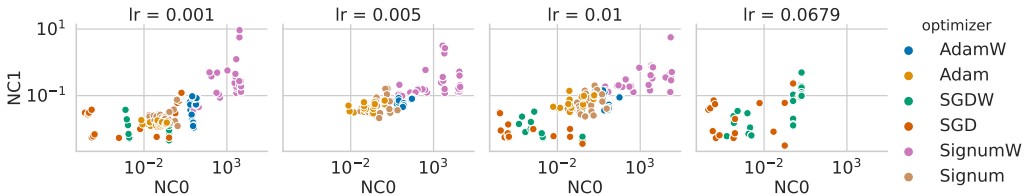

Figure 54: NC0 vs. NC1 on VGG9 trained on MNIST. Note that the x-axis is plotted in log-scale.

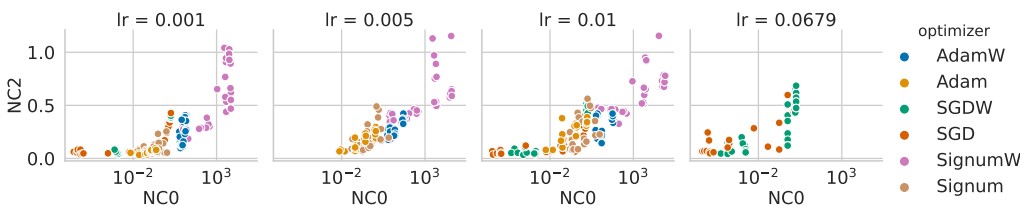

Figure 55: NC0 vs. NC2 on VGG9 trained on MNIST. Note that the x-axis is plotted in log-scale.

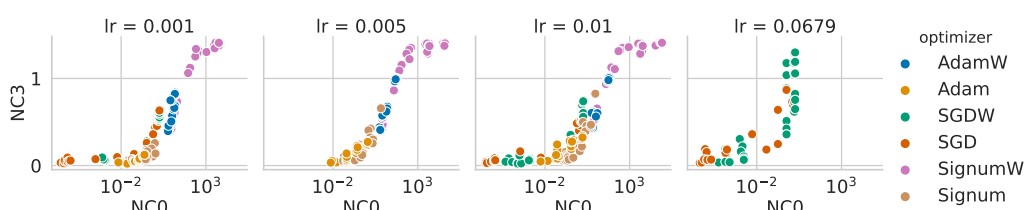

Figure 56: NC0 vs. NC3 on VGG9 trained on MNIST. Note that the x-axis is plotted in log-scale.

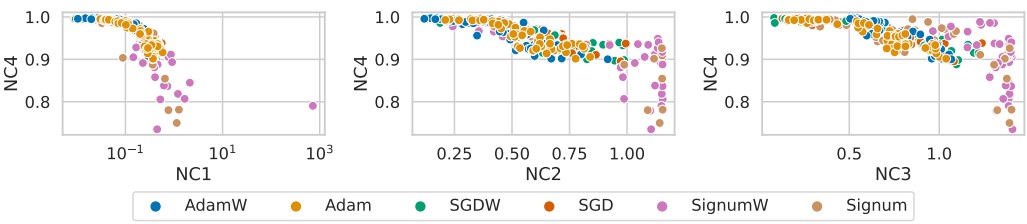

Figure 57: NC4 is largely uncorrelated with NC1-3 across different optimizers and learning rates.

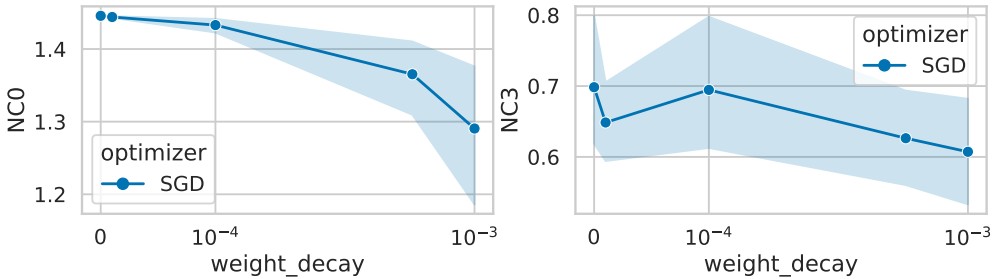

(a) NC0 (left) and NC3 (right) metric for varying values of weight decay on a ViT trained with SGD on Cifar10.

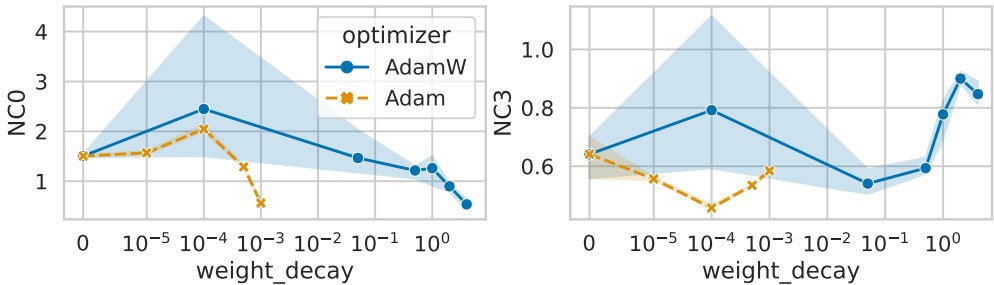

(b) NC0 (left) and NC3 (right) metric for varying values of weight decay on a ViT trained with Adam and AdamW on Cifar10.

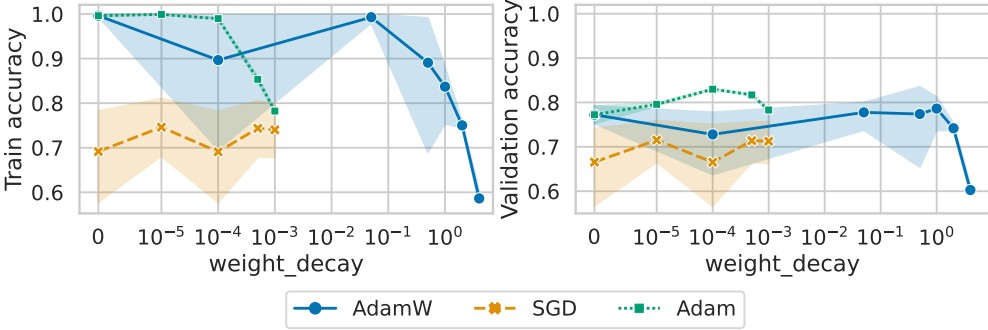

(c) Training accuracy (left) and validation accuracy (right) for varying values of weight decay on a ViT trained on Cifar10.

# E  PROOFS

In this section, we will present the proof which is omitted in the main text.

**Theorem E.1** (Effect of decoupled SGD update on NC0). *Assume a model of the form $f(\mathbf{W}, \theta, x) = \mathbf{W}h_\theta(x)$ is trained using cross-entropy loss with SGD with decoupled weight decay for all parameters $\mathbf{W}, \theta$. For instance, the last layer weight $\mathbf{W}$ has the following update rule:*

$$\mathbf{V}_{t+1} = \beta\mathbf{V}_t + \nabla_{\mathbf{W}_t}L_{\text{CE}},$$
$$\mathbf{W}_{t+1} = (1 - \eta\lambda)\,\mathbf{W}_t - \eta\mathbf{V}_{t+1},$$

*where $\beta \in [0, 1)$, $\eta > 0$, and $\lambda \in \mathbb{R}$. Define the NC0 metric*

$$\alpha_t := \frac{1}{K}\big\|\mathbf{W}_t^\top\mathbf{1}\big\|_2^2.$$

*Then, for all $t \geq 0$,*

$$\alpha_t = (1 - \eta\lambda)^{2t}\,\alpha_0.$$

*In particular, if $0 < \eta\lambda < 2$, then $\alpha_t$ decays exponentially to zero:*

$$\alpha_t = (1 - \eta\lambda)^{2t}\,\alpha_0 \xrightarrow[t\to\infty]{} 0.$$

*Proof.* We track the evolution of the *row sums* of $\mathbf{W}_t$ and $\mathbf{V}_t$. Define

$$\mathbf{m}_t := \mathbf{W}_t^\top\mathbf{1} \in \mathbb{R}^K, \qquad \mathbf{q}_t := \mathbf{V}_t^\top\mathbf{1} \in \mathbb{R}^K.$$

By definition of $\alpha_t$ we have

$$\alpha_t = \frac{1}{K}\|\mathbf{m}_t\|_2^2.$$

Note that by Lemma E.5, the cross-entropy gradient with respect to the last layer satisfies

$$\big(\nabla_{\mathbf{W}_t}L_{\text{CE}}\big)^\top\mathbf{1} = \mathbf{0}$$

for all $\mathbf{W}_t$. Consider the momentum update

$$\mathbf{V}_{t+1} = \beta\mathbf{V}_t + \nabla_{\mathbf{W}_t}L_{\text{CE}}.$$

Multiplying on the right by $\mathbf{1}$ and using the above result, we obtain

$$\mathbf{q}_{t+1} = \mathbf{V}_{t+1}^\top\mathbf{1} = \big(\beta\mathbf{V}_t + \nabla_{\mathbf{W}_t}L_{\text{CE}}\big)^\top\mathbf{1} = \beta\mathbf{V}_t^\top\mathbf{1} + \big(\nabla_{\mathbf{W}_t}L_{\text{CE}}\big)^\top\mathbf{1} = \beta\mathbf{q}_t + \mathbf{0}.$$

Thus $\mathbf{q}_{t+1} = \beta\mathbf{q}_t$, and by induction

$$\mathbf{q}_t = \beta^t\mathbf{q}_0.$$

Since $\mathbf{V}_0 = \mathbf{0}$, we have $\mathbf{q}_0 = \mathbf{0}$, hence

$$\mathbf{q}_t = \mathbf{0} \qquad \text{for all } t \geq 0.$$

Consider now the decoupled weight update

$$\mathbf{W}_{t+1} = (1 - \eta\lambda)\,\mathbf{W}_t - \eta\mathbf{V}_{t+1}.$$

Multiplying on the right by $\mathbf{1}$ gives

$$\mathbf{m}_{t+1} = \mathbf{W}_{t+1}^\top\mathbf{1} = \big((1-\eta\lambda)\mathbf{W}_t - \eta\mathbf{V}_{t+1}\big)^\top\mathbf{1} = (1-\eta\lambda)\mathbf{W}_t^\top\mathbf{1} - \eta\mathbf{V}_{t+1}^\top\mathbf{1} = (1-\eta\lambda)\mathbf{m}_t - \eta\mathbf{q}_{t+1}.$$

Using $\mathbf{q}_{t+1} = \mathbf{0}$ for all $t$, we obtain the simple linear recursion

$$\mathbf{m}_{t+1} = (1 - \eta\lambda)\,\mathbf{m}_t.$$

Solving this recursion yields

$$\mathbf{m}_t = (1 - \eta\lambda)^t\mathbf{m}_0.$$

Substituting the expression for $\mathbf{m}_t$ into the definition of $\alpha_t$ gives

$$\alpha_t = \frac{1}{K}\|\mathbf{m}_t\|_2^2 = \frac{1}{K}\big\|(1 - \eta\lambda)^t\mathbf{m}_0\big\|_2^2 = (1 - \eta\lambda)^{2t}\frac{1}{K}\|\mathbf{m}_0\|_2^2 = (1 - \eta\lambda)^{2t}\alpha_0.$$

This establishes the exact formula claimed in the theorem.

$\square$

**Theorem E.2** (Effect of SGD update with coupled weight decay on NC0). *Assume a model of the form $f(\mathbf{W}, \theta, x) = \mathbf{W}h_\theta(x)$ is trained using cross-entropy loss with stochastic gradient descent (SGD) and momentum $\beta \in [0, 1)$, weight decay $\lambda \in [0, 1)$, and learning rate $\eta > 0$ sufficiently small. The last-layer weights $\mathbf{W}_t$ are updated according to:*

$$\begin{aligned}
\mathbf{V}_{t+1} &= \beta\mathbf{V}_t + \nabla_{\mathbf{W}_t}L_{\text{CE}} + \lambda\mathbf{W}_t, \\
\mathbf{W}_{t+1} &= \mathbf{W}_t - \eta\mathbf{V}_{t+1},
\end{aligned} \tag{15}$$

*where $\beta \in [0, 1)$, $\eta > 0$, and $\lambda \in \mathbb{R}$. Then there exists a constant $C \geq 1$ such that*

$$\alpha_t = \frac{1}{K}\left\|\mathbf{m}_t\right\|_2^2 \leq C\rho^{2t}\alpha_0 \qquad \text{for all } t \geq 0, \tag{16}$$

*where $\rho := \max\{|r_+|, |r_-|\}$ and $r_\pm$ are the roots of*

$$r^2 - (1 + \beta - \eta\lambda)r + \beta = 0. \tag{17}$$

*In particular: if $\eta\lambda < 2(1 + \beta)$, then $\rho < 1$ and the NC0 metric $\alpha_t$ decays exponentially in t.*

*Proof.* We follow the same strategy as in the decoupled case: track the evolution of the row sums of $\mathbf{V}_t$ and $\mathbf{W}_t$.

From (15),
$$\mathbf{V}_{t+1} = \beta\mathbf{V}_t + \nabla_{\mathbf{W}_t}L_{\text{CE}} + \lambda\mathbf{W}_t.$$

Right-multiplying by $\mathbf{1}$ and using Lemma (E.5), we get

$$\begin{aligned}
\mathbf{q}_{t+1} &= \mathbf{V}_{t+1}^\top\mathbf{1} \\
&= \left(\beta\mathbf{V}_t + \nabla_{\mathbf{W}_t}L_{\text{CE}} + \lambda\mathbf{W}_t\right)^\top\mathbf{1} \\
&= \beta\mathbf{V}_t^\top\mathbf{1} + \left(\nabla_{\mathbf{W}_t}L_{\text{CE}}\right)^\top\mathbf{1} + \lambda\mathbf{W}_t^\top\mathbf{1} \\
&= \beta\,\mathbf{q}_t + \lambda\,\mathbf{m}_t.
\end{aligned}$$

Thus

$$\mathbf{q}_{t+1} = \beta\,\mathbf{q}_t + \lambda\,\mathbf{m}_t. \tag{18}$$

From the weight update
$$\mathbf{W}_{t+1} = \mathbf{W}_t - \eta\mathbf{V}_{t+1},$$

we obtain

$$\begin{aligned}
\mathbf{m}_{t+1} = \mathbf{W}_{t+1}^\top\mathbf{1} &= \left(\mathbf{W}_t - \eta\mathbf{V}_{t+1}\right)^\top\mathbf{1} \\
&= \mathbf{W}_t^\top\mathbf{1} - \eta\mathbf{V}_{t+1}^\top\mathbf{1} = \mathbf{m}_t - \eta\,\mathbf{q}_{t+1}.
\end{aligned}$$

Using (18) this becomes

$$\mathbf{m}_{t+1} = \mathbf{m}_t - \eta\left(\beta\,\mathbf{q}_t + \lambda\,\mathbf{m}_t\right) = (1 - \eta\lambda)\,\mathbf{m}_t - \eta\beta\,\mathbf{q}_t. \tag{19}$$

We also have, from the weight update at time $t$,

$$\mathbf{m}_t = \mathbf{m}_{t-1} - \eta\,\mathbf{q}_t,$$

which is just (19) with index shifted by one. Hence

$$\mathbf{q}_t = \frac{1}{\eta}\left(\mathbf{m}_{t-1} - \mathbf{m}_t\right). \tag{20}$$

Substitute (20) into (19):

$$\begin{aligned}
\mathbf{m}_{t+1} &= (1 - \eta\lambda)\,\mathbf{m}_t - \eta\beta\cdot\frac{1}{\eta}\left(\mathbf{m}_{t-1} - \mathbf{m}_t\right) \\
&= (1 - \eta\lambda)\,\mathbf{m}_t - \beta\left(\mathbf{m}_{t-1} - \mathbf{m}_t\right) \\
&= (1 - \eta\lambda)\,\mathbf{m}_t - \beta\mathbf{m}_{t-1} + \beta\mathbf{m}_t \\
&= (1 + \beta - \eta\lambda)\,\mathbf{m}_t - \beta\,\mathbf{m}_{t-1}.
\end{aligned}$$

We are given $\mathbf{m}_0 = \mathbf{W}_0^\top \mathbf{1}$ and $\mathbf{q}_0 = \mathbf{V}_0^\top \mathbf{1} = \mathbf{0}$. Then

$$\mathbf{q}_1 = \beta \mathbf{q}_0 + \lambda \mathbf{m}_0 = \lambda \mathbf{m}_0,$$

and hence from the weight update

$$\mathbf{m}_1 = \mathbf{m}_0 - \eta \mathbf{q}_1 = \mathbf{m}_0 - \eta \lambda \mathbf{m}_0 = (1 - \eta \lambda)\, \mathbf{m}_0.$$

The recurrence is linear and homogeneous with constant coefficients. For each coordinate of $\mathbf{m}_t$, say $(\mathbf{m}_t)_k$, we have a scalar second-order recursion

$$(\mathbf{m}_{t+1})_k = (1 + \beta - \eta \lambda)\, (\mathbf{m}_t)_k - \beta\, (\mathbf{m}_{t-1})_k.$$

The characteristic polynomial is

$$r^2 - (1 + \beta - \eta \lambda)\, r + \beta = 0,$$

with roots $r_+$ and $r_-$ given by

$$r_\pm = \frac{1 + \beta - \eta \lambda \pm \sqrt{(1 + \beta - \eta \lambda)^2 - 4\beta}}{2}.$$

Thus each coordinate can be written as

$$(\mathbf{m}_t)_k = c_{+,k} r_+^t + c_{-,k} r_-^t,$$

for some coefficients $c_{+,k}, c_{-,k}$ determined by $(\mathbf{m}_0)_k$ and $(\mathbf{m}_1)_k$. Let

$$\rho := \max\{|r_+|, |r_-|\}$$

be the spectral radius of the recursion. Then there exists a constant $C \geq 1$ (depending only on $\beta, \lambda, \eta$) such that

$$\left\| \mathbf{m}_t \right\|_2 \ \leq \ C \rho^t \left\| \mathbf{m}_0 \right\|_2,$$

and therefore

$$\alpha_t = \frac{1}{K} \left\| \mathbf{m}_t \right\|_2^2 \ \leq \ C^2 \rho^{2t} \frac{1}{K} \left\| \mathbf{m}_0 \right\|_2^2 = C' \rho^{2t} \alpha_0$$

for some $C' \geq 1$, which is (16). Finally, for a general quadratic equation $r^2 + br + c = 0$, the roots are in the unit circle if $|c| < 1$, $1 + b + c > 0$ and $1 - b + c > 0$. Thus it is not difficult to check from the characteristic polynomial that $\eta \lambda < 2(1 + \beta)$ implies $\rho < 1$.

$$\square$$

Note that the above Theorem holds for any model $f(\mathbf{W}, \theta, x) = \mathbf{W} h_\theta(x)$ with last layer as linear classifier and with any backbone $h_\theta$ parameterized by $\theta$.

However, the dynamics of Adam is more complicated, hence we further restrict the setting to SignGD, a special case of Adam, training a UFM.

Here, we assume a balanced dataset with only one element in each class $k \in [K]$. It is obvious to extend our result to multiple elements per class. Hence the total input $N = K$ is equal to the number of classes and the UFM loss can be written as

$$L_{\text{CE}}(\mathbf{W}\mathbf{H}, \mathbf{I}) = \sum_{n=1}^{N} L_{\text{CE}}(\mathbf{W}\mathbf{h}_n, \mathbf{e}_n),$$

where we can decouple the regularization $\frac{\lambda}{2}\|\mathbf{W}\|^2 + \frac{\lambda}{2}\|\mathbf{H}\|^2$ into weight decay.

By Zhu et al. (2021), we know that the UFM

$$\max_{\mathbf{W}, \mathbf{H}} \sum_{n=1}^{N} L_{\text{CE}}(\mathbf{W}\mathbf{h}_n, \mathbf{y}_n) + \frac{\lambda}{2}\|\mathbf{W}\|^2 + \frac{\lambda}{2}\|\mathbf{H}\|^2,$$

has unique global minimum $\mathbf{W}, \mathbf{H}$ and no strict saddle points. In particular, $\mathbf{H} = \mathbf{U}\mathbf{M}^*$ for some orthogonal matrix $\mathbf{U} \in O(P)$. To further simplify the analysis, we assume that $P = N = K$ with $\mathbf{H} = \mathbf{M}^*$. Then we have the followings:

**Theorem E.3.** *Consider sign GD with (decoupled) weight decay $\lambda > 0$ and step size $\eta > 0$ on the UFM loss*

$$L_{CE}(\mathbf{W}\mathbf{H}, \mathbf{I}) = \sum_{n=1}^{N} L_{CE}(\mathbf{W}\boldsymbol{h}_n, \mathbf{e}_n),$$

*where the feature $\mathbf{H} = \mathbf{M}^*$ is fixed to an NC solution and only the weight $\mathbf{W}$ is trained:*

$$\mathbf{W}_{t+1} = \mathbf{W}_t - \eta(sign(\nabla_{\mathbf{W}_t} L_{CE}) + \lambda \mathbf{W}_t)$$

*with initialization $\mathbf{W}_0 = 0 \in \mathbb{R}^{K \times K}$. We define the covariance matrix $\mathbf{C}_t = \mathbf{W}_t \mathbf{W}_t^\top$ and the scalar $\alpha_t = \langle \mathbf{C}_t, \hat{\mathbf{J}} \rangle_F$ where $\hat{\mathbf{J}} = \frac{1}{K} \mathbf{1}\mathbf{1}^\top$. Then $\alpha_t$ will increase monotonically from zero to the limit:*

$$\lim_{t \to \infty} \alpha_t = \frac{(K-2)^2}{\lambda^2}.$$

*In particular, $\alpha_t$ does not vanish as $t \to \infty$.*

*Proof.* By Lemma E.5, we have $\nabla L_{\text{CE}}(\mathbf{W}) = \frac{1}{N}(\mathbf{S} - \mathbf{Y})\mathbf{H}^\top = \frac{1}{N}(\mathbf{S} - \mathbf{I}) \cdot \frac{1}{\sqrt{K-1}}(\mathbf{I} - \frac{1}{K}\mathbf{J}) = \frac{1}{N\sqrt{K-1}}(\text{softmax}(\mathbf{W}\mathbf{H}) - \mathbf{I})$ since $(\text{softmax}(\mathbf{W}\mathbf{H}) - \mathbf{I})\mathbf{J} = 0$. Since softmax has range between 0 and 1, we have

$$\text{sign}\left(\nabla L_{\text{CE}}(\mathbf{W}\mathbf{H})\right) = \mathbf{J} - 2\mathbf{I},$$

that is, the signed gradient is $-1$ on the diagonal and $+1$ elsewhere. Note that this holds for all $\mathbf{W} \in \mathbb{R}^{K \times K}$. The sign GD updates can hence be written as:

$$\mathbf{W}_{t+1} = \mathbf{W}_t - \eta \Big[ \underbrace{\mathbf{J} - 2\mathbf{I}}_{\text{sign}(\nabla_{\mathbf{W}_t} L_{\text{CE}})} + \lambda \mathbf{W}_t \Big]. \tag{21}$$

Since $\text{sign}\left(\nabla L_{\text{CE}}(\mathbf{W}_t)\right)$ is constant, the dynamics collapse onto a scalar $w_t$:

$$\mathbf{W}_t = w_t (\mathbf{J} - 2\mathbf{I}),$$

which has the following recursive form:

$$w_{t+1} = (1 - \eta\lambda)w_t - \eta, \quad w_0 = 0.$$

Solve it and obtain

$$w_t = -\frac{1}{\lambda}\Big[1 - (1 - \eta\lambda)^t\Big].$$

Recall the definition:

$$\mathbf{C}_t = \mathbf{W}_t \mathbf{W}_t^\top \quad \hat{\mathbf{J}} = \frac{1}{K}\mathbf{1}\mathbf{1}^\top \text{ and } \quad \alpha_t = \langle \mathbf{C}_t, \hat{\mathbf{J}} \rangle_F.$$

Since $\|(\mathbf{J} - 2\mathbf{I})^\top \mathbf{1}\|^2 = (K-2)^2 K$ and the factor of $1/K$ gives $(K-2)^2$, we have

$$\alpha_t = (K-2)^2 w_t^2$$

Therefore

$$\alpha_t = (K-2)^2 \left[ -\frac{1}{\lambda}\Big(1 - (1 - \eta\lambda)^t\Big) \right]^2 = \frac{(K-2)^2}{\lambda^2}\Big[1 - \big(1 - \eta\lambda\big)^t\Big]^2.$$

As $t \to \infty$, $\big(1 - \eta\lambda\big)^t \to 0$, so

$$\alpha_\infty = \frac{(K-2)^2}{\lambda^2}.$$

$\square$

**Theorem E.4.** *Consider sign GD with (coupled) weight decay $\lambda > 0$ and step size $\eta > 0$ on the UFM loss*

$$L_{CE}(\mathbf{W}\mathbf{H}, \mathbf{I}) = \sum_{n=1}^{N} L_{CE}(\mathbf{W}\boldsymbol{h}_n, \mathbf{e}_n),$$

*where the feature $\mathbf{H} = \mathbf{M}^*$ is fixed to an NC solution and only the weight $\mathbf{W}$ is trained :*

$$\mathbf{W}_{t+1} = \mathbf{W}_t - \eta(sign(\nabla_{\mathbf{W}_t} L_{CE} + \lambda \mathbf{W}_t))$$

*with initialization $\mathbf{W}_0 = 0 \in \mathbb{R}^{K \times K}$. We define the covariance matrix $\mathbf{C}_t = \mathbf{W}_t \mathbf{W}_t^\top$ and the scalar $\alpha_t = \langle \mathbf{C}_t, \hat{\mathbf{J}} \rangle_F$ where $\hat{\mathbf{J}} = \frac{1}{K} \mathbf{1}\mathbf{1}^\top$. Then there exists a learning rate decay scheme $\eta = \eta(t) \xrightarrow[t \to \infty]{} 0$ such that $\alpha_t \xrightarrow[t \to \infty]{} 0$.*

*Proof.* Throughout the training, we apply mathematical induction on the structure of $\mathbf{W}_t$: for all $t$, there exists $a_t, b_t \geq 0$ such that

$$\mathbf{W}_t = (a_t + b_t)\mathbf{I} - b_t\mathbf{J}.$$

It is not hard to see that $\alpha = \frac{1}{N}(a_t - (K-1)b_t)^2$. Note that for $t = 0$, the signed gradient is the same as in the case with decoupled weight decay in Theorem 3.3:

$$\text{sign}(\nabla_{\mathbf{W}_t} L_{\text{CE}} + \lambda\mathbf{W}_t) = \text{sign}(\nabla_{\mathbf{W}_0} L_{\text{CE}}) = \text{sign}(\text{softmax}(0) - \mathbf{I}) = \mathbf{J} - 2\mathbf{I}.$$

Hence, $\mathbf{W}_1 = \eta(2\mathbf{I} - \mathbf{J})$ where $a_1 = b_1 = \eta$. Since $\mathbf{H} = \mathbf{M}^* = \frac{1}{\sqrt{K-1}}(\mathbf{I} - \mathbf{J}/k)$,

$$\begin{aligned}
\mathbf{WH} &= ((a_t + b_t)\mathbf{I} - b_t\mathbf{J}) \cdot \frac{1}{\sqrt{K-1}}(\mathbf{I} - \mathbf{J}/k) \\
&= \frac{1}{\sqrt{K-1}}\left((a_t + b_t)\mathbf{I} - b_t\mathbf{J} - (a_t + b_t)\mathbf{J}/k + (b_t/k)\mathbf{J}^2\right) \\
&= \frac{a_t + b_t}{\sqrt{K-1}}\mathbf{H} = \gamma_t\mathbf{H}
\end{aligned}$$

where we define $\gamma_t = \frac{a_t + b_t}{\sqrt{K-1}}$. By Lemma E.5 and the above expression, the loss gradient becomes:

$$\begin{aligned}
\nabla_{\mathbf{W}_t} L_{\text{CE}} &= \frac{1}{N\sqrt{K-1}}(\text{softmax}(\mathbf{WH}) - \mathbf{I}) \\
&= \frac{1}{N\sqrt{K-1}}(\text{softmax}(\gamma_t\mathbf{H}) - \mathbf{I}) \\
&= \psi_t(-K\mathbf{I} + \mathbf{J})
\end{aligned}$$

where $\psi_t = \frac{1}{N\sqrt{K-1}} \cdot \frac{1}{e^{\gamma_t/\sqrt{K-1}} + (K-1)} = \frac{1}{N\sqrt{K-1}} \cdot \frac{1}{e^{(a_t+b_t)/(K-1)} + (K-1)}$. Hence the update weight will also of form

$$\mathbf{W}_{t+1} = (a_{t+1} + b_{t+1})\mathbf{I} - b_{t+1}\mathbf{J}.$$

Hence the update rule of the signed GD with coupled weight decay can be written as:

$$\begin{aligned}
a_{t+1} &= a_t + \eta \cdot \text{sign}\left((K-1)\psi_t - \lambda a_t\right) \\
b_{t+1} &= b_t + \eta \cdot \text{sign}(\psi_t - \lambda b_t)
\end{aligned}$$

Then for each fixed $\eta > 0$, starting from $t = 0$, let $\Delta_t = (a_{t+1} - a_t, b_{t+1} - b_t)$, the training can be divided into three phases:

1. $\Delta_t = (+\eta, +\eta)$ as long as $(K-1)\psi_t \geq \lambda a_t$ and $\psi_t \geq \lambda b_t$. Note that $\psi_t \propto \frac{1}{e^{(a_t+b_t)/(K-1)}+(K-1)}$ hence $\psi_{t+1} < \psi_t$ as $\Delta_t = (+\eta, +\eta)$. Since $\psi_t$ is strictly decreasing with $a_t + b_t$, assume $\eta$ is small enough, there exists a constant $T_1$ such that $\Delta_{T_1} = (+\eta, +\eta)$ but $\Delta_{T_1+1} = (+\eta, -\eta)$ where $(K-1)\psi_{T_1} \geq \lambda a_t \geq \lambda b_t > \psi_{T_1}$.

2. $\Delta_t = (+\eta, \pm\eta)$ indicating $a_t$ increases striclty in each step and $b_t$ starts to oscillate as long as $(K-1)\psi_t \geq \lambda a_t$: each time $\Delta_{t-1} = (+\eta, -\eta)$, we have $a_t + b_t = a_{t-1} + b_{t-1}$ and thus $\psi_t = \psi_{t-1}$. Hence $\psi_t$ decreases monotonically but not strictly. Similar to above, there exists a constant $T_2 > T_1$ such that $(K-1)\psi_{T_2} \geq \lambda a_{T_2}$ but $(K-1)\psi_{T_2+1} < \lambda a_{T_2+1}$.

3. For $t > T_2$, $\Delta_t = (\pm\eta, \pm\eta)$ where i) $\psi_t$ becomes constant for $\Delta_t$ oscillates between $(+\eta, -\eta)$ and $(-\eta, +\eta)$ or ii) $\psi_t$ oscillate for $\Delta_t$ oscillates between $(+\eta, +\eta)$ and $(-\eta, -\eta)$. In wither case, we have $\max_{t > T_2}\{|(K-1)\psi_t - \lambda a_t|, |\psi_t - \lambda b_t|\} < \lambda\eta$ as each update will flip the sign.

Hence for each $\eta$, we update $T_2 = T_2(\eta)$ steps until $\max_{t>T_2}\{|(K-1)\psi_t - \lambda a_t|, |\psi_t - \lambda b_t|\} < \lambda\eta$. Next, we apply learning rate decay to $\eta'$ so that $\lambda\eta' < \min\{|(K-1)\psi_{T_2+1} - \lambda a_{T_2+1}|, |\psi_{T_2+1} - \lambda b_{T_2+1}|\} < \lambda\eta$. Repeat the above argument and find a $T_2' > 0$ such that $\psi_t$ oscillates or remains constant after $t > T_2 + T_2'$, and hence $\max_{t>T_2+T_2'}\{|(K-1)\psi_t - \lambda a_t|, |\psi_t - \lambda b_t|\} < \lambda\eta'$. Induction

on this argument shows that there exists a learning rate decay scheme $\eta = \eta(t) \to 0$ such that $\max_t\{|(K-1)\psi_t - \lambda a_t|, |\psi_t - \lambda b_t|\} \xrightarrow[t\to\infty]{} 0$, in which case:

$$
\begin{aligned}
\alpha_t &= (a_t - (K-1)b_t)^2 \\
&= \lambda^{-2}\left(\lambda a_t - (K-1)\lambda b_t\right)^2 \\
&\leq \lambda^{-2}\left((K-1)\psi_t + |(K-1)\psi_t - \lambda a_t| - (K-1)\psi_t + (K-1)|\psi_t - \lambda b_t|\right)^2 \\
&= \lambda^{-2}\left(|(K-1)\psi_t - \lambda a_t| + (K-1)|\psi_t - \lambda b_t|\right)^2 \\
&\leq \lambda^{-2}K^2 \max_t\{|(K-1)\psi_t - \lambda a_t|, |\psi_t - \lambda b_t|\} \xrightarrow[t\to\infty]{} 0.
\end{aligned}
$$

Hence $\alpha_t = (a_t - (K-1)b_t)^2 \xrightarrow[t\to\infty]{} 0.$ $\qquad\square$

### E.1 TECHNICAL LEMMATA

**Lemma E.5.** *Let $(\mathbf{X}, \mathbf{Y}) \in \mathbb{R}^{d\times N} \times \mathbb{R}^{K\times N}$ be a dataset where the labels $\mathbf{Y}$ are written in columns of one-hot vectors. For each pair $(\mathbf{x}, \mathbf{y}) \in \mathbb{R}^D \times \mathbb{R}^K$, and a weight $\mathbf{W}_1 \in \mathbb{R}^{K\times d}$, define the cross-entropy as:*

$$
\ell(\mathbf{W}_1) \stackrel{\text{def.}}{=} -\sum_{k=1}^{K} \mathbf{y}_k \log\left(\text{softmax}(\mathbf{W}_1\mathbf{x})\right)_k = \log\left(1 + \sum_{k\neq y} \exp(\mathbf{w}_k - \mathbf{w}_y)^\top \mathbf{x}_i\right)
$$

*where $y = \arg\max_{k\in[K]}[\mathbf{y}]_k$ is the class index of $\mathbf{x}$. Let $\mathcal{L}_1(\mathbf{W}_1) = CE(\mathbf{W}_1\mathbf{X}, \mathbf{Y})$ be the average cross-entropy loss of the dataset $(\mathbf{X}, \mathbf{Y})$. Then the loss gradient $\nabla\mathcal{L}_1(\mathbf{W}_1)$ is*

$$
\nabla\mathcal{L}_1(\mathbf{W}_1) = \frac{1}{N}(\mathbf{S} - \mathbf{Y})\mathbf{X}^\top
$$

*where $\mathbf{S} = (\mathbf{s}_1, ...\mathbf{s}_N)$ and $\mathbf{s}_i = \text{softmax}(\mathbf{W}_1\mathbf{x}_i)$ for each $i$. In particular, $\mathbf{1}_K^\top\nabla\mathcal{L}_1(\mathbf{W}_1) = 0$.*

*Proof.* The expression of the loss gradient comes from simple calculus. The second statement comes from the fact that the L1 norms of a post-softmax vector and an one-hot vector are both equal to 1, that is,
$$
\mathbf{1}_K^\top\mathbf{s}_i = \mathbf{1}_K^\top\mathbf{y}_i = 1\,\forall i.
$$
$\qquad\square$

**Lemma E.6.** *Assume the weight $\mathbf{W}_t$ is updated as follows:*

$$
\begin{aligned}
\mathbf{V}_{t+1} &= \beta\mathbf{V}_t + \mathbf{G}_t + \lambda\mathbf{W}_t \\
\mathbf{W}_{t+1} &= \mathbf{W}_t - \eta\mathbf{V}_{t+1},
\end{aligned}
$$

*where $\mathbf{G}_t$ depends on $\mathbf{W}_t$. Define*

$$
\alpha \stackrel{\text{def.}}{=} \frac{1}{K}\|\mathbf{W}_t^\top\mathbf{1}\|_2^2 \geq 0.
$$

*Then we have the expression:*

$$
\frac{1}{\eta}(\alpha_{t+1} - \alpha_t) = -2\beta\omega_t - 2\gamma_t - 2\lambda\alpha_t + \eta\nu_{t+1}
$$

*where $\omega_t \stackrel{\text{def.}}{=} \langle\mathbf{V}_t\mathbf{W}_t^\top, \hat{\mathbf{J}}\rangle$, $\gamma_t \stackrel{\text{def.}}{=} \langle\mathbf{G}_t\mathbf{W}_t^\top, \hat{\mathbf{J}}\rangle$, $\nu_t \stackrel{\text{def.}}{=} \langle\mathbf{V}_t\mathbf{V}_t^\top, \hat{\mathbf{J}}\rangle$.*

*Proof.* Let $\mathbf{C}_t \stackrel{\text{def.}}{=} \mathbf{W}_t\mathbf{W}_t^\top$ be the covariance matrix. Notice that $\alpha_t = \langle\mathbf{C}_t, \hat{\mathbf{J}}\rangle$ where $\hat{\mathbf{J}} = \frac{1}{K}\mathbf{1}\mathbf{1}^\top$. By update rule of $\mathbf{W}_t$ and $\mathbf{V}_t$:

$$
\begin{aligned}
\frac{1}{\eta}(\mathbf{C}_{t+1} - \mathbf{C}_t) &= \frac{1}{\eta}\left((\mathbf{W}_t - \eta\mathbf{V}_{t+1})(\mathbf{W}_t - \eta\mathbf{V}_{t+1})^\top - \mathbf{C}_t\right) \\
&= -(\mathbf{V}_{t+1}\mathbf{W}_t^\top + \mathbf{W}_t\mathbf{V}_{t+1}^\top) + \eta\mathbf{V}_{t+1}\mathbf{V}_{t+1}^\top.
\end{aligned}
$$

Applying the dot product $\langle \cdot, \hat{\mathbf{J}} \rangle_F$ on both sides, and denote $\omega_t \overset{\text{def.}}{=} \langle \mathbf{V}_t \mathbf{W}_t^\top, \hat{\mathbf{J}} \rangle$, $\gamma_t \overset{\text{def.}}{=} \langle \mathbf{G}_t \mathbf{W}_t^\top, \hat{\mathbf{J}} \rangle$, $\nu_t \overset{\text{def.}}{=} \langle \mathbf{V}_t \mathbf{V}_t^\top, \hat{\mathbf{J}} \rangle$, we have

$$\frac{1}{\eta}(\alpha_{t+1} - \alpha_t) = -2\langle \mathbf{V}_{t+1} \mathbf{W}_t^\top, \hat{\mathbf{J}} \rangle + \eta \langle \mathbf{V}_{t+1} \mathbf{V}_{t+1}^\top, \hat{\mathbf{J}} \rangle$$

$$= -2\langle (\beta \mathbf{V}_t + \mathbf{G}_t + \lambda \mathbf{W}_t) \mathbf{W}_t^\top, \hat{\mathbf{J}} \rangle + \eta \nu_{t+1}$$

$$= -2\beta \omega_t - 2\gamma_t - 2\lambda \alpha_t + \eta \nu_{t+1} \tag{22}$$

where in the first line we use the fact that $\hat{\mathbf{J}}$ is symmetric.

$\square$

**Lemma E.7.** *Assume* $\lambda, \beta \in (0, 1)$ *such that* $\frac{2\lambda}{\log \beta^{-1}} < 1$. *The solution of the following ODE:*

$$\dot{\alpha}(t) = -\lambda \left( \int_0^t \beta^{t-\tau} \alpha(\tau) d\tau \right) \tag{23}$$

*with initial condition* $\alpha(0) = \alpha_0 > 0$ *admits the following bound:*

$$\alpha(t) \le C\alpha_0 \exp\left( -\frac{\lambda}{\log \beta^{-1}} t \right)$$

*for some absolute constant* $C > 1$.

*Proof.* Observe that we can write the integral in convolution:

$$\int_0^t \beta^{t-\tau} \alpha(\tau) d\tau = (\phi * \alpha)(t), \quad \text{where} \quad \phi(t) = \beta^t.$$

Hence (23) can be written as

$$\dot{\alpha}(t) = -\lambda (\phi * \alpha)(t).$$

Let $\mathcal{L}\{\psi(t)\}(s) = \int_0^\infty e^{-st} \psi(t) dt$ denote the Laplace transform. Denote

$$\mathcal{A}(s) = \mathcal{L}\{\alpha(t)\}(s), \quad F(s) = \mathcal{L}\{\phi(t)\}(s).$$

Taking the Laplace transform of both sides:

$$\mathcal{L}\{\dot{\alpha}(t)\}(s) = -\lambda \mathcal{L}\{(\phi * \alpha)(t)\}(s). \tag{24}$$

And by integration by part and the property of convolution,

$$\mathcal{L}\{\dot{\alpha}(t)\}(s) = s\mathcal{A}(s) - \alpha(0) \quad \text{and} \quad \mathcal{L}\{(\phi * \alpha)(t)\}(s) = F(s)\mathcal{A}(s).$$

Hence

$$s\mathcal{A}(s) - \alpha(0) = -\lambda F(s)\mathcal{A}(s).$$

Since $\beta^t = e^{(\log \beta)t}$, we get

$$F(s) = \mathcal{L}\{\beta^t\}(s) = \mathcal{L}\{e^{(\log \beta)t}\}(s) = \frac{1}{s - \log(\beta)} \quad \text{for } s > \log(\beta).$$

Substitute this back to Eq. (24) and we get:

$$s\mathcal{A}(s) - \alpha(0) = -\lambda \frac{1}{s - \log(\beta)} \mathcal{A}(s)$$

$$s\mathcal{A}(s) + \frac{\lambda}{s - \log(\beta)} \mathcal{A}(s) = \alpha(0)$$

$$\mathcal{A}(s) \underbrace{\left( s + \frac{\lambda}{s - \log(\beta)} \right)}_{\frac{s^2 - s \log(\beta) + \lambda}{s - \log(\beta)}} = \alpha(0)$$

$$\mathcal{A}(s) = \alpha(0) \cdot \frac{\left[ s - \log(\beta) \right]}{\underbrace{s^2 - s \log(\beta) + \lambda}_{(s-r_1)(s-r_2)}}$$

where $r_1, r_2 = \frac{\log(\beta) \pm \sqrt{\left[\log(\beta)\right]^2 - 4\lambda}}{2}$. We do partial fractions and matching coefficients gives:

$$\frac{s - \log(\beta)}{(s - r_1)(s - r_2)} = \frac{A}{s - r_1} + \frac{B}{s - r_2} \implies A + B = 1, \quad -\log(\beta) = -Ar_2 - Br_1.$$

Since $r_1 + r_2 = \log(\beta)$, one finds

$$A = \frac{r_2}{r_2 - r_1}, \quad B = -\frac{r_1}{r_2 - r_1}.$$

Thus

$$\mathcal{A}(s) = \alpha(0) \left[ \frac{r_2}{r_2 - r_1} \frac{1}{s - r_1} - \frac{r_1}{r_2 - r_1} \frac{1}{s - r_2} \right].$$

Recall the inverse of Laplacian transform: $\mathcal{L}^{-1}\{\frac{1}{s-r}\}(t) = e^{rt}$. Therefore,

$$\alpha(t) = \mathcal{L}^{-1}\{A(s)\}(t) = \alpha(0) \left[ \frac{r_2}{r_2 - r_1} e^{r_1 t} - \frac{r_1}{r_2 - r_1} e^{r_2 t} \right].$$

Equivalently,

$$\alpha(t) = \alpha(0) \left[ A e^{r_1 t} + B e^{r_2 t} \right], \quad A = \frac{r_2}{r_2 - r_1}, B = -\frac{r_1}{r_2 - r_1}, \tag{25}$$

where

$$r_1, r_2 = \frac{\log(\beta) \pm \sqrt{\left[\log(\beta)\right]^2 - 4\lambda}}{2}.$$

Since $\beta \in (0, 1)$, set $L = -\log(\beta) > 0$. By the first order approximation,

$$\sqrt{(\log \beta)^2 - 4\lambda} = \sqrt{L^2 - 4\lambda} = L - \frac{2\lambda}{L} + \mathcal{O}\left(\frac{\lambda^2}{L}\right)$$

Hence

$$r_1, r_2 = \frac{-L \pm \left(L - \frac{2\lambda}{L}\right)}{2} + \mathcal{O}\left(\frac{\lambda^2}{L}\right).$$

This gives:

$$r_1 = -\frac{\lambda}{L} + \mathcal{O}\left(\frac{\lambda^2}{L}\right), \quad r_2 = -L + \frac{\lambda}{L} + \mathcal{O}\left(\frac{\lambda^2}{L}\right).$$

Plugging $r_1, r_2$ into Eq. (25):

$$\alpha(t) \le C\alpha(0)e^{r_1 t} = C\alpha(0) \exp\left(-\frac{\lambda}{L}t\right)$$

for some absolute constant $C > 1$. Plug in $L = -\log(\beta) = \log \beta^{-1}$ to finish the proof.

$\square$

