# OpenReview forum: "Optimizer Choice Matters For The Emergence of Neural Collapse"
_ICLR.cc/2026/Conference — ICLR 2026 Poster_

### Official Review · Reviewer_2Q1N · 2025-10-17

**Soundness:** 3
**Presentation:** 3
**Contribution:** 2
**Rating:** 2
**Confidence:** 4

**Summary:**

The authors address an important literature gap in the understanding of the conditions under which neural collapse emerges. The paper points out that the emergence of neural collapse can largely depend on the choice of the optimizer. In particular, they show that in adaptive optimizers, particularly those of second-order nature, the emergence of NC (in particular the weight-feature alignment NC3) might depend on the way in which weight decay is implemented – whether it is coupled or decoupled. The authors show both empirically and theoretically, that AdamW or SignumW consistently don’t converge to zero/small values of NC3 metric, while their coupled counterparts (as well as SGD with momentum, either coupled or uncoupled) converge to much smaller values. The main analytic tool (also used extensively in experiments) is the so-called NC0 metric (the global bias of the rows of last layer’s weight matrix), which is a necessary condition for the emergence of full NC (especially NC2 and NC3 together). The authors show that this metric converges to a non-zero constant for signGD with decoupled weight decay, while for the coupled weight decay it might converge to zero provided vanishing learning rate. The authors complement these main findings by several other related observations and ablations.

**Strengths:**

-	S1: Both theoretical and empirical evidence presented in the paper is convincing (modulo a few caveats mentioned in the weaknesses). The paper makes it clear that the presented phenomenon is indeed happening. Some of the ablations in the appendix (for instance the one with 2000 training epochs) reassure me even more.
-	S2: The paper tackles an important topic of better understanding the conditions under which the neural collapse emerges.
-	S3: The paper is mostly well written and easy to follow.
-	S4: The paper also presents several non-central insights, such as the role of the momentum, analysis of the interpolation between the two Adam variants, discussion of the meaning of the NC metrics.
-	S5: The authors seem to be open about the limitations of the paper and discuss them appropriately.

**Weaknesses:**

-	W1: If I correctly understand the mathematical essence of the paper, it seems that authors are misinterpreting their results. To me, the main distinction should not be coupled vs. decoupled weight decay but rather first- vs. second-order optimizers. It seems that the crucial parameter that really determines the limiting behavior is not so much the implementation of the weight decay, but rather whether the gradient computation corresponds precisely to the original loss function and thus, whether the zero-sum gradient property from line 263 holds for that optimizer or not. In particular, while authors correctly interpret Theorem 3.3, I believe that Theorem 3.4 should be interpreted as a proof that SignGD with coupled weight decay also doesn’t converge to zero, even in this idealistic ETF features scenario. I think there is a good empirical evidence that Adam itself doesn’t converge to NC too – in Table 2 we see that the best NC0 metric reached by Adam is orders of magnitude higher than those reached by both SGD and SGDW. Similar conclusions can be drawn from Figure 7 where even for pure Adam, NC3 metric seems to converge to positive value well above 0, or Figure 6, where we see that the smallest eigenvalue of Adam converges to small, yet non-zero value. In Figure 5, while NC3 is still improving for Adam, the NC0 seems to have converged. The SGDW, on the other hand, despite the decoupling, does not suffer from the above-mentioned issues, which suggests that the crucial aspect is indeed the division by empirical exponentially averaged variance in second-order optimizers like Adam. Now, the reason why there is such a big empirical difference between Adam and AdamW is that the presence of WD in the internal updates of the momentum and variance parameters seems to alleviate (yet not fully erase) this systematic bias. That’s why the empirical results seem to suggest that the difference between decoupled and coupled weight decay is bigger than that between first and second-order optimizers. Thus, I agree that this difference should be discussed and the paper provides compelling evidence for it, but I don’t think it should be narrated without proper discussion (or even prioritization) of the differences between the optimizer types.

-	W2: Some of the empirical evidence gives mixed signals and gives the reader an impression that the authors did not properly calibrate the weight decay ranges for which the experiments were conducted. This can be seen by the fact that a lot of weight-decay vs. NC3 plots show concave behavior for the decoupled weight decay variants. This is true in Figure 3, but also in appendix figures 17, 20, 24, 25, 29, 30, 35, 40. This suggests that higher values of weight decay for the decoupled variants could reach as low values of NC3 as the coupled variants. Another strong evidence for this is in Figure 14 for NC0 metric, where we see a huge difference in the metric for the largest weight decay, again suggesting that the weight decay ranges should have been wider on the right end of the plots. While part of this phenomenon could be explained by the quadratic relationship between NC0 and regularization strength, the fact that NC3 metric is normalized properly makes it unclear to what extent are the curves concave because of this, and to what extent because of the actual convergence of NC3 to lower values.

-	W3: A fair comparison of weight decay and momentum values across experiments, or when comparing coupled and decoupled implementation, is a general issue in the paper. For instance, it is well known and it agrees with Theorem 3.1 of the paper, that the weight decay generally only influences the speed of convergence of NC metrics, but not necessarily whether they converge. Furthermore, it is also well-known that the training converges towards neural collapse also for zero values of weight decay, but the convergence is slower. Therefore, results such as the ones in Figure 4 should be calibrated to account for this, for instance by letting training run until a fixed small training loss is achieved. If the authors only meant that weight decay and momentum only affect the speed of convergence, then it should be more explicitly discussed in the paper, but these results are not novel.

-	W4: Related to the previous point, I believe that the NC0 metric should be normalized by the norm of the weight matrix. If the NC0 metric remains constant but both the weight matrix and the final layer features grow to infinity (as is known to be the case in the zero weight decay case), than geometrically NC can still be asymptotically approached. Therefore, normalizing NC0 metric would provide for a more fair comparison across various weight decay values.

-	W5: Finally, the strength of the message and/or contribution of the paper is relatively modest, especially because we see from the experiments that many conclusions the authors draw only hold for NC3 metric, but not for NC1/2 metrics. Thus, the coupling of weight decay / optimizer itself seem to be particularly important only for the NC3 metric. This limits the scope of the results, although I agree that the observation that full NC convergence might depend on the optimizer is still an important and insightful addition to our knowledge.

-	W6: Some writing aspects should be improved. In particular:
-    Equation 1 doesn’t use bias in the last layer, but in appendix authors include bias in the definition.
-	The original definition of the NC metrics formulated them for the centered class means. The authors should discuss why they decided to omit this detail in their definitions. This also influences the line 143-144 in the proof where authors talk about the convergence of centered class means.
-	Line 147: I don’t see why this should hold. We can always rotate M^* and counter-rotate P accordingly.
-	The constant T in Theorem 3.1 is redundant.
-	Line 273: you cite the wrong paper of Jacot. It should be Jacot 2024.
-	The definition of within-class variability in line 637 is wrong.
-	There is a notation inconsistency between the definition of the \alpha parameter in Theorem 3.3 and its appendix version.

**Questions:**

-	In Figure 1 we see a little increase in the NC3 metric of Adam optimizer for the largest value of weight decay. Do you have a guess why this would be?
-	Why is there such a small variance in the measured metrics in Figure 1 and similar figures, when you average them over a wide variety of learning rates and momentums?
-	Have you tried to train Adam with as high values of weight decay as AdamW? What would happen?
-	The NC0 values for SGD in Figure 8 seem weird. Do you have an explanation?
-	In Figure 15 we see that the metrics go up for Adam for large values of weight decay. Why is that?
-    How big of a difference there is in your opinion between real Adam and SignGD in terms of implicit biases? We know that Adam is known to converge in some regular cases, while SignGD does not seem to be likely to ever converge. For this reason, do you think that SignGD is a good approximation of Adam?


**SUMMARY:**
While I think the content of this paper is publishable material and the contributions are insightful, I cannot recommend acceptance at this point due to fair amount of concerns I have. I would recommend the authors to re-structure and re-interpret the work and resubmit. Moreover, since the scope of the paper is a bit modest (particularly because it mostly only concerns the NC3 metric), my recommendation would be to submit to TMLR.

---

> ### Author Response · Authors · 2025-11-18
> **Response to reviewers concerns and questions - Part 1**
>
> We would like to thank the reviewer for acknowledging our contributions as insightful and taking the time to review our work and providing very detailed feedback.
>
> In the sections below we will try our best to address the concerns raised in the review.
>
> As several of the points raised are substantial, we structure our response into separate parts.
>
> ## Weaknesses
> > **1.1 W1**: If I correctly understand the mathematical essence of the paper, it seems that authors are misinterpreting their results. To me, the main distinction should not be coupled vs. decoupled weight decay but rather first- vs. second-order optimizers.
>
> We respectfully disagree that we are misinterpreting the results. As indicated in the title, our central message is that the choice of optimizer strongly affects the emergence of neural collapse. This includes distinctions between first- and second-order optimizers (e.g., SGD vs. Adam), the use of coupled vs. decoupled weight decay, and additional factors such as momentum. As detailed below, we believe it is particularly important to highlight the role of coupled vs. decoupled weight decay in the context of adaptive methods. While our experimental results already pointed toward this conclusion, we have now reinforced it by adding a new theoretical result. We thank the reviewer for prompting this clarification. We also emphasize that the changes made to the paper - highlighted in different colors in the revised PDF - are minimal.
>
> > **1.2 W1**: It seems that the crucial parameter that really determines the limiting behavior is not so much the implementation of the weight decay, but rather whether the gradient computation corresponds precisely to the original loss function and thus, whether the zero-sum gradient property from line 263 holds for that optimizer or not. In particular, while authors correctly interpret Theorem 3.3, I believe that Theorem 3.4 should be interpreted as a proof that SignGD with coupled weight decay also doesn’t converge to zero, even in this idealistic ETF features scenario.
>
> We agree that the current form of Theorem 3.4 could not clearly support our claim that Adam (with decoupled weight decay) leads to NC0. To this end, we extend the result of Theorem 3.4 with a step-wise learning rate decay which proves that NC0 vanishes as $\eta=\eta(t)\xrightarrow[t\to\infty]{}0$.
>
> For sake of completeness, we have also rewritten and extended our results on SGD with coupled or decoupled weight decay. Please see the following summary table.
> The complete statements and proofs can be found in our revised paper.
>
>
> | Index       | Optimizers      | Model | Convergence to 0? | learning rate   |
> |-------------|-----------------|-------|-------------------|-----------------|
> | Theorem 3.1 | SGD with DWD    | Any   | yes, exponential  | constant        |
> | Theorem 3.2 | SGD with CWD    | Any   | yes, exponential  | constant        |
> | Theorem 3.3 | SignGD with DWD | UFM   | yes               | step-wise decay |
> | Theorem 3.4 | SignGD with CWD | UFM   | no                | -               |
>
> > **1.3 W1**: I think there is a good empirical evidence that Adam itself doesn’t converge to NC too – in Table 2 we see that the best NC0 metric reached by Adam is orders of magnitude higher than those reached by both SGD and SGDW. Similar conclusions can be drawn from Figure 7 where even for pure Adam, NC3 metric seems to converge to positive value well above 0, or Figure 6, where we see that the smallest eigenvalue of Adam converges to small, yet non-zero value. In Figure 5, while NC3 is still improving for Adam, the NC0 seems to have converged.
>
> Thanks to the new Theorem 3.3, we expect NC0 for Adam in Figure 7 would drop further if one prolongs the training with learning rate decay. Note that Theorem 3.3 on AdamW is still valid, showing AdamW and Adam have qualitatively different dynamics.
>
> > **1.4 W1**: The SGDW, on the other hand, despite the decoupling, does not suffer from the above-mentioned issues, which suggests that the crucial aspect is indeed the division by empirical exponentially averaged variance in second-order optimizers like Adam.
>
> Yes, the reviewer's observation on SGD is accurate and is clearly reflected in Figure 3 (right) and Figure 17 and 18 in the revised version. This is now additionally addressed in the new Theorems 3.1 and 3.2.
> We note that we already made a similar observation in the original version, which we can now further strengthen using our new theoretical result on SGDW (see line 330-336 in the revised version.)

---

> > ### Author Response · Authors · 2025-11-18
> > **Response to reviewers concerns and questions - Part 2**
> >
> > > **1.5 W1**: Now, the reason why there is such a big empirical difference between Adam and AdamW is that the presence of WD in the internal updates of the momentum and variance parameters seems to alleviate (yet not fully erase) this systematic bias. That’s why the empirical results seem to suggest that the difference between decoupled and coupled weight decay is bigger than that between first and second-order optimizers.
> > Thus, I agree that this difference should be discussed and the paper provides compelling evidence for it, but I don’t think it should be narrated without proper discussion (or even prioritization) of the differences between the optimizer types.
> >
> > It is a constructive idea to elaborate more on the difference between the optimizers. Indeed, the above theorems show that SGD and Adam achieve NC0 for different reasons: the former one holds for any constant learning rate, and the latter one requires learning rate decay to zero at $t=\infty$. We promise to extend the discussion on differences between the optimizer types in our revised paper.
> >
> > To further evaluate the differences between SGD, SGDW, Adam, and AdamW, we reproduce Figure 5 and 6 for the training runs which achieve the smallest NC3 for each optimizer.
> > The results can be found in Appendix D.4.3. and Fig. 16-19. We first note that SGDW and SGD achieve very similar final NC0-NC3 metrics, confirming the new results in Theorem 3.1, as well as the observation of the reviewer.
> > It is harder to judge whether AdamW or Adam are closer to a NC solution based on NC1-NC3. While NC3 is considerably larger for AdamW, NC1 is slightly larger for Adam, compared to the other optimizers.
> > Nonetheless, the NC0 metric in Fig. 17 and the evolution of the singular values of $\mathbf{W}$ in Fig. 18 (left) indicate that AdamW has considerably different training dynamics than Adam, as both NC0 as well as the smallest singular value increase instead of converging to zero for AdamW. While the NC0 metric of Adam is orders of magnitude larger than for SGD and SGDW and the smallest singular value of $\mathbf{W}$ converges to a small, but non-zero value, Adam shares similar trends to SGD and SGDW and as such converges to a solution which is arguably closer to NC3 than AdamW.
> > Whether the solution found by Adam can already be classified as NC or not is an inherent problem of interpreting the NC metrics in practical settings, as we have also discussed in section 4.1.
> >
> > > **2. W2**: Some of the empirical evidence gives mixed signals and gives the reader an impression that the authors did not properly calibrate the weight decay ranges for which the experiments were conducted. This can be seen by the fact that a lot of weight-decay vs. NC3 plots show concave behavior for the decoupled weight decay variants. [...]
> > Another strong evidence for this is in Figure 14 for NC0 metric, where we see a huge difference in the metric for the largest weight decay, again suggesting that the weight decay ranges should have been wider on the right end of the plots. While part of this phenomenon could be explained by the quadratic relationship between NC0 and regularization strength, the fact that NC3 metric is normalized properly makes it unclear to what extent are the curves concave because of this, and to what extent because of the actual convergence of NC3 to lower values.
> >
> > Thank you for raising this point. We apologize if the figures give the impression of not proper calibration of the weight decay ranges for which the experiments were conducted.
> > We explain the choice of the hyperparameter ranges, including weight decay in Appendix D.1. In particular, for SGD, SGDW, Adam and Signum weight decay was chosen in the range $\in \[0, 0.00005, 0.0005,0.005, 0.05, 0.5\]$ and for SignumW and AdamW in the range $\in [0, 0.0005, 0.05, 0.5, 5, 10]$. The main reason for using AdamW and Signum W with much larger weight decay values was based on the hypothesis that the effect of weight decay is reduced due to decoupling.
> > As the reader might observe already, the figures do not show the full range of weight decay values for each optimizer. This is because the model does not converge to high training accuracy if the weight decay is chosen too large, as in these cases the regularization term dominates the training dynamics. Thus, we have removed these training runs from the figures. **To avoid any confusion, we have added a remark in the main text as well as in Figure 1 and 3.**

---

> > > ### Author Response · Authors · 2025-11-18
> > > **Response to reviewers concerns and questions - Part 3**
> > >
> > > > **3. W3**: A fair comparison of weight decay and momentum values across experiments, or when comparing coupled and decoupled implementation, is a general issue in the paper. For instance, it is well known and it agrees with Theorem 3.1 of the paper, that the weight decay generally only influences the speed of convergence of NC metrics, but not necessarily whether they converge. Furthermore, it is also well-known that the training converges towards neural collapse also for zero values of weight decay, but the convergence is slower. Therefore, results such as the ones in Figure 4 should be calibrated to account for this, for instance by letting training run until a fixed small training loss is achieved. If the authors only meant that weight decay and momentum only affect the speed of convergence, then it should be more explicitly discussed in the paper, but these results are not novel.
> > >
> > > Thank you for this comment.
> > > We would like to clarify that the convergence of NC metrics depends on the presence of weight decay, but that momentum just accelerates convergence in general and also the convergence of the NC metrics as Theorem 3.1 shows.
> > > The point of Figure 4 was to highlight the results from Theorem 3.1 under **practical, non-asymptotic training**, namely the necessity of weight decay for NC (which agrees with prior work) and the accelerating effect of momentum. This is why we trained all models for the same number of 200 epochs. **Note that the effect of momentum goes beyond just accelerating training convergence, as can be seen in Figure 21 and 22 in the appendix D.4.4. that we have additionally provided.**
> > > Note for instance that the model reaches almost the same final training loss with momentum=0.7 and momentum=0.9. However, comparing the NC0-NC3 metrics, all metrics show a clear discrepancy, with higher momentum having smaller NC metrics.
> > > The accelerating effect of momentum on training convergence is mainly observable in the early phase of training, while momentum affects the convergence of the NC metrics beyond this. To the best of our knowledge, this is a novel observation which has not been reported in prior work.
> > >
> > > We would like to further remark that the final train loss is directly affected by the regularization effect of weight decay. Thus, training runs with large weight decay actually have a larger final train loss than runs with smaller values of weight decay, which would amplify the picture in Figure 4.
> > >
> > > > **4. W4**: Related to the previous point, I believe that the NC0 metric should be normalized by the norm of the weight matrix. If the NC0 metric remains constant but both the weight matrix and the final layer features grow to infinity (as is known to be the case in the zero weight decay case), than geometrically NC can still be asymptotically approached. Therefore, normalizing NC0 metric would provide for a more fair comparison across various weight decay values.
> > >
> > > We followed your suggestion and also computed the normalized version of NC0, which we compute as  $\mathcal{NC}0_{\text{norm}} := \frac{1}{p} \| \mathbf{W}_{L}^{\top} \mathbf{1}\|/\|\mathbf{W}\|_F$ for tracking the NC metrics for runs achieving minimal NC3. The results can be found in Figure 20. While the absolute values differ from each other, the overall trend remains unchanged for all optimizers.
> > >
> > > > **5. W5**: Finally, the strength of the message and/or contribution of the paper is relatively modest, especially because we see from the experiments that many conclusions the authors draw only hold for NC3 metric, but not for NC1/2 metrics. Thus, the coupling of weight decay / optimizer itself seem to be particularly important only for the NC3 metric. This limits the scope of the results, although **I agree that the observation that full NC convergence might depend on the optimizer is still an important and insightful addition to our knowledge.**
> > >
> > > Thank you for acknowledging that our observation that full NC convergence might depend on the optimizer is important and an insightful addition to our knowledge.
> > > We agree that our observations mainly hold for the NC3 metric, but not for NC1/2 metrics. We would like to argue however that this by itself is already a novel observation.
> > > Previous work has implicitly assumed or did not explicitly question the simultaneous convergence of the NC1-NC4 metrics. Our work is the first to study the role of the optimizer in the emergence of Neural collapse, particularly identifying the difference between coupled and decoupled weight decay in adaptive optimizers as an important ingredient.

---

> > > > ### Author Response · Authors · 2025-11-18
> > > > **Response to reviewers concerns and questions - Part 4**
> > > >
> > > > ### Improving writing aspects:
> > > >
> > > > > 7. Equation 1 doesn’t use bias in the last layer, but in appendix authors include bias in the definition.
> > > >
> > > > Thank you for pointing out. We have corrected this typo in the revised version.
> > > >
> > > > > 8. The original definition of the NC metrics formulated them for the centered class means. The authors should discuss why they decided to omit this detail in their definitions. This also influences the line 143-144 in the proof where authors talk about the convergence of centered class means.
> > > >
> > > > Thank you. It was a typo in the definition and we have corrected it. Note that the proof of Prop. 2.1 has been using the centered class mean and it remains correct.
> > > >
> > > > > 9. Line 147: I don’t see why this should hold. We can always rotate $M^*$ and counter-rotate $P$ accordingly.
> > > >
> > > > Sorry for the confusion. The orthogonal matrix $P$ you mentioned can be absorbed to $Q$ and does not affect the rest of the proof. We have added this observation into our proof.
> > > >
> > > >
> > > > > 10. The constant T in Theorem 3.1 is redundant.
> > > >
> > > > We have modified Theorem 3.1 and simplified the proof.
> > > >
> > > > > 11. Line 273: you cite the wrong paper of Jacot. It should be Jacot 2024.
> > > >
> > > > Thank you for pointing this out, we have corrected this now.
> > > >
> > > > > 12. The definition of within-class variability in line 637 is wrong.
> > > >
> > > > Thank you, we have replaced this by the correct definition now.
> > > >
> > > > > 13. There is a notation inconsistency between the definition of the \alpha parameter in Theorem 3.3 and its appendix version.
> > > >
> > > > Thank you for pointing this out, we have corrected this.
> > > >
> > > > ## Questions:
> > > >
> > > > > Why is there such a small variance in the measured metrics in Figure 1 and similar figures, when you average them over a wide variety of learning rates and momentums?
> > > >
> > > > The reason for the small variance is mainly because across all runs which achieve reasonable high training accuracy, weight decay seems to be the dominating factor for the final value of NC0 and NC3.
> > > >
> > > > > Have you tried to train Adam with as high values of weight decay as AdamW? What would happen?
> > > >
> > > > Thank you for this question! This is indeed something that we have considered already. As you can see in the appendix D.1 we used a much wider range for weight decay than is shown in most figures. (The largest tested WD for Adam is 0.5 and for AdamW is 10). The reason for this is that we discarded all runs which do not converge, which in our cases correspond to reaching a training accuracy of at least 99%. Note that with sufficiently large values, weight decay will dominate the training dynamics, leading to the model not learning anything meaningful.
> > > >
> > > > > The NC0 values for SGD in Figure 8 seem weird. Do you have an explanation?
> > > >
> > > > Could you perhaps elaborate what you think is weird in Figure 8 regarding the NC0 values for SGD?
> > > >
> > > > > In Figure 1 we see a little increase in the NC3 metric of Adam optimizer for the largest value of weight decay. Do you have a guess why this would be?
> > > > > In Figure 15 we see that the metrics go up for Adam for large values of weight decay. Why is that?
> > > >
> > > > Thank you for your question. While we do not have a definitive explanation, a plausible reason is that large weight-decay values may over-regularize the model. In Adam, weight decay also influences the computation of the first and second moments, which can exacerbate this effect. Consequently, for the largest weight-decay value considered, the model may already be impeded from training effectively.
> > > >
> > > > > How big of a difference there is in your opinion between real Adam and SignGD in terms of implicit biases?
> > > > We know that Adam is known to converge in some regular cases, while SignGD does not seem to be likely to ever converge. For this reason, do you think that SignGD is a good approximation of Adam?
> > > >
> > > > Thank you for this question. As SignGD can be considered as a special case of Adam, clearly there are cases in which SignGD and Adam behave significantly different from each other. In particular, since one applies the sign operator in SignGD, convergence is only possible with appropriate learning rate decay.
> > > > However, considering the implicit bias of Adam and SignGD with respect to NC, we believe that our empirical results, e.g. Figure 8, show that SignGD serves as a good proxy for Adam, at least qualitatively. In particular, the theoretical analysis of SignGD with coupled and decoupled weight decay provides an explanation between the empirical observed difference between Adam and AdamW with respect to the emergence of NC.

---

> > > > > ### Author Response · Authors · 2025-11-18
> > > > > **Summary of our response to reviewers concerns and questions**
> > > > >
> > > > > **SUMMARY**:
> > > > > We thank the reviewer for their thorough and constructive feedback, which helped us significantly strengthen the submission. We have incorporated several valuable clarifications and additions based on these comments. **Importantly**, however, **the core message of the paper remains unchanged**: the choice of optimizer plays a crucial role for the emergence of NC (as highlighted by our choice of the title submission), including the form of preconditioning (e.g., SGD vs. Adam), the treatment of weight decay (coupled vs. decoupled), and the strength of momentum.
> > > > > We have implemented all suggested revisions, highlighting in blue the new additions prompted by the reviewers’ questions and in dark green the modifications addressing requests for clarification.

---

> ### Comment · Reviewer_2Q1N · 2025-11-19
> **Thank you for your answers**
>
> Thank you for your detailed answers and for significant extensions that strengthen the submission, such as Theorem 3.4 and the new appendices D.4.3-D.4.5. I will now follow-up in the points which require further discussion.
>
> Let me first highlight points that I am satisfied with:
>
> - I mostly agree with your answers to W1 and appreciate that you agree that both distinctions are important. As you promised, some rewriting of the discussions will have to be made for final revision, but all-in-all I think now we at least agree on this point.
>
> - The ablation D.4.5 is really good and I am now convinced that momentum has an effect on NC that cannot be explained by training loss.
>
> - The ablation D.4.3 is also good and helps highlighting that both the optimizer type as well as the weight decay coupling are important.
>
> - The explanation of the weight decay ranges also seems to resolve the issue.
>
> Now I follow-up on some points where I have further comments:
>
> - **Theorem 3.4:** While I am generally satisfied with the theorem now, I have two follow-ups. First, the proof is now a bit hasty. I think a bit more formal finish (argument that the $a_t, b_t$ indeed reach the equilibrium you are claiming) is required. Second and *more importantly,* it is clear that the Adam gradient is still biased (non-zero sum over the rows of $W$) and therefore what I am wondering is, whether the balancing effect of weight decay in this case is a lucky coincidence or whether there is some deeper reason why it ends up oscillating around zero (and not some other constant) even with finite learning rate.
>
> - **Regarding W3:** while I am convinced by ablation in the Appendix D.4.5 about momentum, I still disagree that zero weight decay would not lead to NC. It is just known to be quite slow. See for instance [1]. Do you have some compelling evidence that would show the opposite?
>
> - **Regarding W4:** Thank you for doing the ablation. However, I think picking the runs that achieved the best NC3, which happen to be the ones where weight decay value is very large, is not a choice that would be expected to show the difference. I am sure the difference will be visible and relevant for the zero weight decay or very small values of weight decay. Alternatively, the normalization is important if comparing one algorithm across multiple weight decay values, such as in Figure 1. An ablation where we redo the Figure 1 and similar ones with the normalized metric, or an alternative to Appendix D.4.4. but with runs with zero or very small values of weight decay would be appreciated.
>
> - **Regarding Q1:** As a follow-up, I have to ask whether this is consistent with your observations in Figure 4 where you show that momentum seems to also have a strong influence on the NC0 and NC3.
>
> - **Regarding Q4:** The NC0 values for SGD in Figure 8 seem weird because the $y-$axis range is very very narrow.
>
> [1] Ji, Wenlong, et al. "An unconstrained layer-peeled perspective on neural collapse." arXiv preprint arXiv:2110.02796 (2021).

---

> ### Author Response · Authors · 2025-11-23
> **Response to remaining concerns and comments - Part 1**
>
> We thank the reviewer for their prompt follow-up, and for acknowledging that we have strengthened our submission through our response.
>
> We are glad to hear that we have addressed most major concerns of the reviewer and will try to further answer the remaining concerns below.
>
> > **Theorem 3.4**: While I am generally satisfied with the theorem now, I have two follow-ups. First, the proof is now a bit hasty. I think a bit more formal finish (argument that the indeed reach the equilibrium you are claiming) is required. Second and more importantly, it is clear that the Adam gradient is still biased (non-zero sum over the rows of ) and therefore what I am wondering is, whether the balancing effect of weight decay in this case is a lucky coincidence or whether there is some deeper reason why it ends up oscillating around zero (and not some other constant) even with finite learning rate.
>
> Thank you for this follow-up. We have further polished Theorem 3.4: In the main text, we show a cleaner and shorter statement saying that there exists a learning rate decay scheme such that NC0 will tend to zero as $t$ tends to infinity; in the appendix we provide a more detailed and formal proof on the dynamics and convergence.
>
> If we understand the second part of your question correctly, you are asking about the effect of weight decay on the NC0 dynamics. This can be seen in the update rule in the proof: if $\lambda=0$, the terms inside the sign function will never flip its sign and hence the NC0 dynamics will increase strictly to infinity (see phase 1 dynamics in the proof) and NC will never occur. In other words, weight decay is essential for NC0 to oscillate around some positive constant (proportional to $\eta^2$) after long enough steps with a fixed step size $\eta$. Whenever the dynamics becomes oscillation, if we decrease the step size and train further, the same argument will apply until the new dynamics becomes oscillation around some smaller constant. Hence there exists a learning rate decay scheme such that NC0 vanishes as t tends to infinity.
>
> In short, $\lambda>0$ causes NC0 to oscillate (instead of diverging to infinity); a learning rate decay causes NC0's oscillation around zero at $t=\infty$. Both are essential for NC to occur.
>
> Please let us know if we have misunderstood your question, we would be happy to elaborate more.
>
> > **Regarding W3**: while I am convinced by ablation in the Appendix D.4.5 about momentum, I still disagree that zero weight decay would not lead to NC. It is just known to be quite slow. See for instance [1]. Do you have some compelling evidence that would show the opposite?
>
> Thank you for following-up on this point and providing the reference. We would like to emphasize the theorems from ULPM limits to gradient flow on  two layer networks. For their empirical validation beyond these settings, we can see that although Fig. 2 in [1] shows that the NC metrics do decrease, they also "seem to converge to positive value well above 0", according to the reviewer's standard. Especially for NC3, one can see this in Fig. 4d, 5d, 7d, and 8d in [1]. Hence by the same logic, the reviewer would not find the statement that "NC occurs without weight decay" from [1] convincing.
>
> To further investigate this question whether WD is necessary or not for NC, we track the NC metrics while training a ResNet9 on FashionMNIST with SGD using zero WD and varying values of momentum for 200 epochs with an initial LR=0.01. Additionally, we train the model with zero momentum and high momentum=0.98 for 2000 epochs, with LR decay after 1/3 and 2/3 of training. Importantly, all training runs reach perfect train accuracy after ~40 epochs.
> The training dynamics can be found in **Fig. 24** (We have also tracked normalized NC0 (see **Fig. 21**), which might also partially answer your question on W4, which we further elaborate in the next point.)
> We can draw two conclusions from our experiments:
> 1. The final NC metrics NC0-NC3 after 2000 epochs are slightly smaller than after 200 epochs, consistent with our ablation study in D.4.1. that longer training reduces the NC metrics.
> 2. The final NC metrics (both for 200 epochs and 2000 epochs of training) remain considerably higher than what is achieved by the "best" run of SGD in terms of NC metrics with 200 epochs of training for all NC metrics, even with 10 times longer training. See **Fig. 17** and **Fig. 19** for a comparison.
> The smallest achieved NC1-3 metrics are respectively:
>
> | SGD with ... | NC1 | NC2 | NC3 |
> |-----|---|----|----|
> |weight decay (200 epochs)| 0.02 | 0.2 | 0.13|
> |no weight decay (2000 epochs)| 0.2 | 0.55 | 0.7|
>
> While the experiments cannot fully exclude the possibility that NC can be achieved eventually in the asymptotic limit, we argue that WD is essential to observe the emergence of NC at least in **practical finite-length training settings**.

---

> > ### Comment · Reviewer_2Q1N · 2025-11-24
> >
> > Thank you for continuing the constructive discussion and doing all the helpful ablation studies. My questions are now resolved and I also mostly agree with your results about the W4 where I agree that the normalized NC0 would probably not change any qualitative conclusions, but as we see from the results, the slopes of some curves (like in Figure 1) could still be quantitatively influenced.
> >
> > - **Regarding Theorem 3.4:** Unfortunately it seems there was a small misunderstanding. What astounds me is not necessarily that the NC0 metric would oscillate, but rather that the *constant* around which it oscillates converges to zero as a function of the learning rate *itself*. This seems to me to be a lucky coincidence, but perhaps your expert insight into the proofs can reveal a deep reason why the ultimate limit is zero and why the gradient asymptotically becomes unbiased, at least in the NC0 sense. In other words - even if the WD has the power to flip the signs of the gradients before the normalization happens, what I don't understand is, whether it is a coincidence that the positive and negative signs end up being balanced so that they eventually sum up to zero as we scale down the learning rate.
> >
> > - **Regarding W3:** I certainly agree that within the reasonable training budget it is more-or-less necessary to use weight decay and I also agree that you managed to demonstrate *this* in your ablation. I merely wanted to express my disagreement with the claim that weight decay is necessary for neural collapse *whatsoever*. This is clearly not the case from Figure 2 of [1] where yes, the NC metrics are still quite high, but they show clear linear convergence on the log-log scale and therefore a clear trend towards lower and eventually perfect values. And their theory also proves this. It is not done on 2-layer neural networks but on UFM and the fact that gradient flow is used does not disqualify the result because this result clearly generalizes to GD and SGD, even with momentum.

---

> > > ### Author Response · Authors · 2025-11-25
> > >
> > > Thank you again for your prompt response and helping us to improve our work through this constructive discussion.
> > > We are glad to hear that your questions are now resolved.
> > >
> > > Below we provide our response to your follow-up comments:
> > >
> > >
> > > > Regarding Theorem 3.4: Unfortunately it seems there was a small misunderstanding. What astounds me is not necessarily that the NC0 metric would oscillate, but rather that the constant around which it oscillates converges to zero as a function of the learning rate itself. This seems to me to be a lucky coincidence, but perhaps your expert insight into the proofs can reveal a deep reason why the ultimate limit is zero and why the gradient asymptotically becomes unbiased, at least in the NC0 sense. In other words - even if the WD has the power to flip the signs of the gradients before the normalization happens, what I don't understand is, whether it is a coincidence that the positive and negative signs end up being balanced so that they eventually sum up to zero as we scale down the learning rate.
> > >
> > > Thank you for your elaboration. First, we agree with your point that the oscillation mid-point of NC0 can be seen as a function of $\eta$, which converges to zero along with $\eta$. However, the oscillation dynamic is unavoidable for the sign gradient update with a fixed step size $\eta$: for each step, each coordinate of any weight is updated by exactly $\eta$. If the training does not diverge to infinity, then the dynamics would oscillate eventually. However, from the update rule of sign gradient descent (with coupled weight decay), we know that the equilibrium point is where $a = (K-1)b$, which is NC (and thus NC0). In short, oscillation is unavoidable for sign graident update; learning rate decay dimishes the oscillation and the weight will converge to the equilibrium point; the equilibrium point is where NC occurs. Hence it is no lucky coincidence that NC0 converges to zero in this case; rather in the proof we find a way for sign gradient descent so that the weights can converge to the equilibrium point of the update.
> > > We hope that this clarifies your comment.
> > >
> > > > Regarding W3: I certainly agree that within the reasonable training budget it is more-or-less necessary to use weight decay and I also agree that you managed to demonstrate this in your ablation. I merely wanted to express my disagreement with the claim that weight decay is necessary for neural collapse whatsoever. This is clearly not the case from Figure 2 of [1] where yes, the NC metrics are still quite high, but they show clear linear convergence on the log-log scale and therefore a clear trend towards lower and eventually perfect values. And their theory also proves this. It is not done on 2-layer neural networks but on UFM and the fact that gradient flow is used does not disqualify the result because this result clearly generalizes to GD and SGD, even with momentum.
> > >
> > > Thank you for this comment. We would like to clarify a few points, where our wording in our previous response was not precise enough. First of all, you are certainly right that [1] studies the ULPM objective which is not simply a 2-layer neural network, we apologize for the wrong claim in our previous response. We further agree that Figure 2 in [1] indeed shows clear linear convergence on the log-log scale, confirming the theory that they prove on the ULPM.
> > > What we merely wanted to point out is that there still seems to be a disparity between the ULPM and practical models, such as ResNets and VGGs, shown in Fig. 4-8 in [1], which do not have this linear convergence on the log-log scale. This also aligns with the experiments and ablation studies that we have conducted in our work.
> > >
> > > To make our claim on weight decay more precise we have modified our statement in our revision (**line 214-219**) and added a footnote on the ULPM studied in [1]. Please let us know if this resolves your concern or if you would like us to adjust our wording in other parts of the submission.
> > >
> > > ---
> > > We would like to again thank the reviewer for their time and effort that they have devoted to reviewing our work and providing constructive feedback throughout the rebuttal so far.
> > >
> > > If there are any remaining concerns or questions, we would happily address them and continue this constructive discussion. Otherwise, if our rebuttal has meaningfully reduced your earlier concerns, we would sincerely appreciate your consideration of adjusting the score accordingly.
> > >
> > > Thank you again for your thoughtful review.

---

> > > > ### Comment · Reviewer_2Q1N · 2025-11-25
> > > >
> > > > Thank you again for your further efforts to clarify all the points.
> > > >
> > > > - **Regarding Theorem 3.4:** My question was more around why is the equilibrium point balanced (e.g. leads to collapse). Whether this is a coincidence that has to do with the loss function or a general phenomenon. But I think the question is not important enough to overstretch this discussion, since the Theorem is obviously working.
> > > >
> > > > - **Regarding W3:** All right, I think we can settle on this as is. To give you my personal opinion, I think what happens in figures 4-8 in [1] is not an intrinsic limitation, but only a practical one. But I cannot back it up formally so we can leave it here.
> > > >
> > > > Thank you yet again for very constructive discussion and a lot of new ablations that certainly strengthen your point. I will update my score accordingly.

---

> > > > > ### Author Response · Authors · 2025-11-25
> > > > >
> > > > > We are glad that we have resolved your concerns and thank you for updating your score accordingly.
> > > > >
> > > > > We would like to further thank you for this very constructive discussion, which has helped us to strengthen our submission and present it more clearly.

---

> ### Author Response · Authors · 2025-11-23
> **Response to remaining concerns and comments - Part 2**
>
> > **Regarding W4**: Thank you for doing the ablation. However, I think picking the runs that achieved the best NC3, which happen to be the ones where weight decay value is very large, is not a choice that would be expected to show the difference. I am sure the difference will be visible and relevant for the zero weight decay or very small values of weight decay. Alternatively, the normalization is important if comparing one algorithm across multiple weight decay values, such as in Figure 1. An ablation where we redo the Figure 1 and similar ones with the normalized metric, or an alternative to Appendix D.4.4. but with runs with zero or very small values of weight decay would be appreciated.
>
> Thank you for this comment. You are indeed correct that the difference between NC0 and a normalized NC0 would only be visible for small or zero weight decay, and that the figures that we provided in D.4.4. were not sufficient to address your comment.
> To further address your concerns, we have also plotted normalized NC0 for the setting of SGD with zero weight decay discussed in the previous point.
> You can find the results for zero WD in **Figure 21 in Appendix D.4.4.**
> As you have rightfully argued, the dynamics are quite different for NC0 and normalized NC0 and one can observe the monotontic effect of momentum on normalized NC0, but not on NC0.
> However, we would like to point out that in this case normalized NC0 does not correlate with NC1-NC3 anymore. On the contrary, NC1-NC3, while still comparably large, are smaller with less momentum.
> While also NC0 does not necessarily correlate more with NC1-NC3 in this setting, since the values are increasing, at least the ordering of NC0 values for the training runs is more consistent for the first two thirds of training (up to around 120 epochs), i.e. training runs with small momentum have both lower NC0 and NC1-NC3.
> As the dynamics of NC0 and normalized NC0 are either almost the same for large values of WD or normalized NC0 is not or less consistent with NC1-NC3 for zero WD, we remain tentative to conclude that the normalization will not affect the conclusions that we draw in this work.
>
> > **Regarding Q1**: As a follow-up, I have to ask whether this is consistent with your observations in Figure 4 where you show that momentum seems to also have a strong influence on the NC0 and NC3.
>
> Thank you for this follow-up. To best address your question, we have provided more detailed figures in **App. D.4.7** where we plot NC0 and NC3 as a function of WD for varying values of LR and momentum. The results can be found in **Figures 25-27**.
> It can be seen that for the adaptive optimizers (Adam, AdamW, Signum and SignumW) as well as SGDW the variance of NC0 and NC3 for varying values of momentum is comparably small for each fixed learning rate, with the variance generally increasing with larger weight decay.
> For SGD the variance for NC0 is higher for large values of weight decay, consistent with what is shown in Figure 3 (right) and in Figure 4.
> Note again that some points are missing due to the fact that we only included runs, which achieved at least a train accuracy of. 99\%.
>
> > **Regarding Q4**: The NC0 values for SGD in Figure 8 seem weird because the axis range is very very narrow.
>
> Thank you for clarifying your question.
> We have rechecked the figure and corresponding experiment and note that NC0 does not change a lot for SGD as NC0 at initialization is already very small (= 0.001).
>
> The difference between the initial NC0 value of each the optimizers in the original figure was merely due to different plotting settings, as all training runs have the same initialization.
> The NC0 dynamics were previously logged every 10 epochs, with the first logged point corresponding to after 10 epochs of training.
> We have replaced Figure 8 now with **a more detailed plot**, in which we log the NC0 metric after every step of training.
> We hope that this clarifies your question.
>
>
> We hope that we have addressed your remaining concerns and questions and would be happy to clarify any remaining points.

---

### Official Review · Reviewer_nwnq · 2025-10-31

**Soundness:** 3
**Presentation:** 3
**Contribution:** 3
**Rating:** 8
**Confidence:** 3

**Summary:**

The authors provide new insight into Neural Collapse (NC). They present NC0, a new necessary condition for NC2 and NC3, and demonstrate that the choice of optimizer impacts neural collapse, particularly the momentum and weight decay parameters and their specific implementation.
Experimentally, the authors show that non-zero weight decay is necessary for NC0 and NC3 metrics to decrease. They also observe that these NC metrics decrease both by increasing weight decay (for a fixed momentum) and by increasing momentum (for a fixed non-zero weight decay). This behavior is theoretically proven for SGD with momentum and weight decay.
Finally, they show that neural collapse does not emerge with decoupled weight decay for optimizers like Adam and Signum. They prove this for signed SGD with decoupled weight decay, while also showing that NC0 does decrease for signed SGD when using coupled weight decay.

**Strengths:**

- This work investigates Neural Collapse from a novel perspective, supported by both experimentation and theoretical evidence.
- It opens up for further discussion and deeper investigation in the field.
- The work is clearly written.
- The authors critically discuss their findings and limitations.

**Weaknesses:**

- Some plots are difficult to interpret due to overlapping lines and similar colors.

**Questions:**

Given that the occurrence of NC is connected to better generalization, how would the authors interpret their result that decoupled weight decay prevents Neural Collapse, in connection with the fact that AdamW is an industry standard due to its performance gain over coupled weight decay?

---

> ### Author Response · Authors · 2025-11-18
> **Response to reviewers comments**
>
> We would like to thank the reviewer for their feedback and for their positive evaluation of our work.
>
> #### Weaknesses
> > - Some plots are difficult to interpret due to overlapping lines and similar colors.
>
> Thank you for this comment. If you can specify the figures that are difficult to interpret, we are happy to improve them. Generally, we tried to improve readability by using a colorblind friendly palette and markers to distinguish the lines in case the figures are viewed in grayscale.

---

### Official Review · Reviewer_fTjQ · 2025-11-01

**Soundness:** 3
**Presentation:** 3
**Contribution:** 3
**Rating:** 6
**Confidence:** 3

**Summary:**

The paper argues that how weight decay is implemented largely determines whether Neural Collapse (NC) emerges. It introduces NC0 as a necessary diagnostic and proves contrasting NC0 dynamics in a stylized model. The paper also validates the mechanism with large-scale experiments, showing NC metrics improve as the coupled component increases, while accuracy remains about the same.

**Strengths:**

1. An actionable mechanism that practitioners can immediately test and reason about.

2. NC0 provides a tractable and provable indicator, enabling convergence and impossibility statements.

3. Value the importance of optimization algorithm choices in NC.

**Weaknesses:**

1. Modeling gap to real Adam/AdamW. The formal results use SignGD and unconstrained features as a proxy. While the qualitative match to Adam/AdamW is persuasive, the absence of finite-step analysis with (β₁, β₂, ε) leaves open whether corner-casescould break the claimed dynamics.

2. NC0 is necessary, not sufficient. It could happen that NC0→0 but full NC (NC1–NC3) can still fail. Without parallel theory for the other NC metrics, one could over-infer the presence or absence of collapse.

3. External validity across regimes. The empirical case is broad but still focused on common settings. Very large batches, aggressive data augmentation, label smoothing, and heavy regularization may alter gradient noise scales and effective WD, potentially changing the coupling story.

**Questions:**

See weaknesses.

---

> ### Author Response · Authors · 2025-11-18
> **Response to reviewers comments and concerns**
>
> We would like to thank the reviewer for their feedback and for their positive evaluation of our work.
>
> > Modeling gap to real Adam/AdamW. The formal results use SignGD and unconstrained features as a proxy. While the qualitative match to Adam/AdamW is persuasive, the absence of finite-step analysis with (β₁, β₂, ε) leaves open whether corner-casescould break the claimed dynamics.
>
> As the reviewer correctly notes, a full theoretical analysis of Adam/AdamW is notoriously difficult. However, SignGD is a special case of Adam and our experiments show that both Adam and SignGD exhibit similar NC0 dynamics. We therefore believe that combining rigorous theoretical results for SignGD with empirical evidence for Adam provides a meaningful and novel contribution of our work.
>
> > NC0 is necessary, not sufficient. It could happen that NC0→0 but full NC (NC1–NC3) can still fail. Without parallel theory for the other NC metrics, one could over-infer the presence or absence of collapse.
>
> Establishing that NC0 is a necessary condition is precisely what enables our main result, namely demonstrating that AdamW cannot achieve full neural collapse. Developing a complete theory of the sufficient conditions for NC is beyond the scope of this work, but we agree with the reviewer that it is an important direction for future research.
>
> > External validity across regimes. The empirical case is broad but still focused on common settings. Very large batches, aggressive data augmentation, label smoothing, and heavy regularization may alter gradient noise scales and effective WD, potentially changing the coupling story.
>
> We thank the reviewer for these thoughtful suggestions. We note that the weight-decay magnitudes used in our experiments already correspond to strong regularization. The additional ideas - such as more aggressive data augmentation or label smoothing - are indeed promising directions and merit a thorough exploration. While such an exploration lies beyond the scope of the present work, we point out that certain augmentations have already been examined in the context of neural collapse; for example, the impact of Mixup [Zhang et al., 2017] was recently analyzed in Fisher et al. [2024].
>
>
> We hope to have addressed the reviewers concerns and are happy to clarify any further questions.
>
> ---
> - Zhang, Hongyi, et al. "mixup: Beyond empirical risk minimization." arXiv preprint arXiv:1710.09412 (2017).
> - Fisher, Quinn, Haoming Meng, and Vardan Papyan. "Pushing boundaries: Mixup's influence on neural collapse." arXiv preprint arXiv:2402.06171 (2024).

---

### Author Response · Authors · 2025-12-01
**General summary**

We would like to thank the reviewers for providing constructive feedback, especially *Reviewer 2Q1N, who engaged in a very constructive discussion*, which helped us to strengthen our submission through new ablation studies, theoretical results, and further clarifications.

We are glad to have **addressed all concerns and comments of Reviewer 2Q1N** who increased their score to "8: accept, good paper" after our discussion (the scores were rolled back due to the data leak). Both **Reviewer nwnq** and **Reviewer fTjQ** provided an overall positive review, but did not further engage in a discussion during the rebuttal.

After the active discussion with **Reviewer 2Q1N**, we are confident that our additional theoretical results and ablation studies provided during the rebuttal underpin our claim that the choice of optimizer plays a crucial role for NC to occur.

We summarize the rebuttal of our submission below:
### New theoretical results

- **Theorem 3.1**: NC0 vanishes exponentially for SGD with decoupled WD.
- **Refined Theorem 3.3**: SignGD with decoupled WD and *step-wise learning rate decay* also has vanishing NC0.

Please see the following summary table for an overview of our theoretical results.
The complete statements and proofs can be found in our revised paper.


|Index|Optimizers|Model|Convergence to 0?|learning rate|
|-|-|-|-|-|
|Theorem 3.1|SGD with DWD|Any|yes, exponential|constant|
|Theorem 3.2|SGD with CWD|Any|yes, exponential|constant|
|Theorem 3.3|SignGD with DWD|UFM|yes|step-wise decay|
|Theorem 3.4|SignGD with CWD|UFM|no|-|

### Key new empirical results and additional Figures
 - **Training dynamics of minimal NC3 runs (App. D.4.3.)**: We tracked the training dynamics of the runs with minimal NC3  to disentangle the effect of using first-order optimizers (such as SGD and SGDW) vs. second-order like optimizers (such as Adam and AdamW) from the effect of applying coupled vs. decoupled weight decay.
The results, in particular of NC0 and of the smallest singular value of $\mathbf{W}$ indicate that the dynamics of Adam is closer to that of SGD and SGDW.
 - **Ablation study on effect of momentum on NC emergence (App. D.4.5.)**: The purpose of this ablation study was to evaluate whether momentum affects the emergence of NC beyond simply accelerating the speed of convergence. The results in Fig. 22 and 23 suggest to affirm this claim. To the best of our knowledge, connecting the magnitude of momentum to NC is novel and has not been discussed in prior work.
 - **NC emergence under zero weight decay (App. D.4.6.)**: The purpose of this ablation study was to evaluate whether NC can be achieved without weight decay after a pointer to results in [1] by Reviewer 2Q1N. The final NC metrics reached after 2000 epochs of training (10x longer than baseline) are still considerably larger than the best runs with WD after 200 epochs of training (see Tab. 6). While the experiments cannot fully exclude the possibility that NC can be achieved eventually in the asymptotic limit, we argue that WD is essential to observe the emergence of NC in *practical finite-length training settings*.

Regarding comments from **Reviewer nwnq** and **Reviewer fTjQ**:
- *Reviewer fTjQ*'s main critiques were 1) the modeling gap of SignGD to real Adam/AdamW and 2) NC0 only being a necessary condition. We summarize our response below:
    - 1) SignGD is a special case of Adam and our experiments show that both Adam and SignGD exhibit similar NC0 dynamics. We therefore believe that combining rigorous theoretical results for SignGD with empirical evidence for Adam provides a meaningful and novel contribution of our work.
    - 2) Establishing that NC0 is a necessary condition is precisely what enables our main result, namely demonstrating that AdamW cannot achieve full NC. Developing a complete theory of the sufficient conditions for NC is beyond the scope of this work, but we agree with the reviewer that it is an important direction for future research.
- *Reviewer nwnq*'s main comment were that some plots were difficult to interpret:
    - We tried to improve readability by using a colorblind friendly palette and markers to distinguish the lines in case the figures are viewed in grayscale.

**Conclusions.**
 We would like to once again emphasize the *core message of the paper*: the choice of optimizer plays a crucial role for the emergence of NC (as highlighted by the title of our submission), including the form of preconditioning (e.g., SGD vs. Adam), the form of weight decay (coupled vs. decoupled), and the strength of momentum. To the best of our knowledge, this is novel and has not been discussed in prior work.

We have implemented all suggested revisions, highlighting in blue new additions prompted by the reviewers’ questions and in dark green modifications addressing requests for clarification.

---
[1] Ji, Wenlong, et al. "An unconstrained layer-peeled perspective on neural collapse." arXiv preprint arXiv:2110.02796 (2021).

---

### Meta-Review · Area_Chair_ZKA3 · 2026-01-07

**Summary:**

The paper investigates the role of optimization in the emergence of Neural Collapse (NC), highlighting that this phenomenon depends significantly on the choice of optimizer. Specifically, the authors demonstrate that for adaptive optimizers, the emergence of NC is highly sensitive to the implementation of weight decay. Through both empirical and theoretical analysis, the authors show that AdamW and SignumW consistently fail to reach near-zero values for the NC3 metric. In contrast, their coupled counterparts, as well as SGD with momentum (in both coupled and uncoupled forms), converge to significantly lower NC values. The authors complement these primary findings with several related observations and extensive ablation studies.

While many comments were initially raised concerning the presentation and experimental settings, they appear to have been successfully addressed during the rebuttal and revision. Ultimately, all reviewers are convinced by the paper’s primary contribution that the choice of optimizer significantly affects the emergence of NC.

**Reviewer Concerns:**

Several primary concerns were raised by the reviewers: (1) the presentation and interpretation of the results; (2) experimental settings, specifically regarding the fair comparison of weight decay and momentum values across different configurations; (3) the modeling gap between SignGD and actual Adam/AdamW implementations; (4) whether $NC_0$ (convergence of training loss to zero) is merely a necessary condition for Neural Collapse; and (5) general writing and organizational aspects. It appears that the majority of these concerns have been successfully addressed through the rebuttal and revision.

**Reviewer Scores:**

The paper originally received scores of 6, 8, and 2. Both Reviewer nwnq and Reviewer fTjQ provided positive reviews and are likely to maintain their scores. Reviewer 2Q1N (initially gave a score of 2) engaged extensively in the discussion with the authors and indicated an intention to adjust their score and convinced the main claim that the choice of optimizer plays a crucial role in the emergence of NC.

---

### Decision · Program_Chairs · 2026-01-26

Accept (Poster)